# 😳 UNFOLDING SPATIAL COGNITION: EVALUATING MULTIMODAL MODELS ON VISUAL SIMULATIONS

**Linjie Li[1], Mahtab Bigverdi[1], Jiawei Gu[2], Zixian Ma[1], Yinuo Yang[1], Ziang Li[1], Yejin Choi[3], Ranjay Krishna[1]**

[1]University of Washington    [2]National University of Singapore    [3]Stanford University

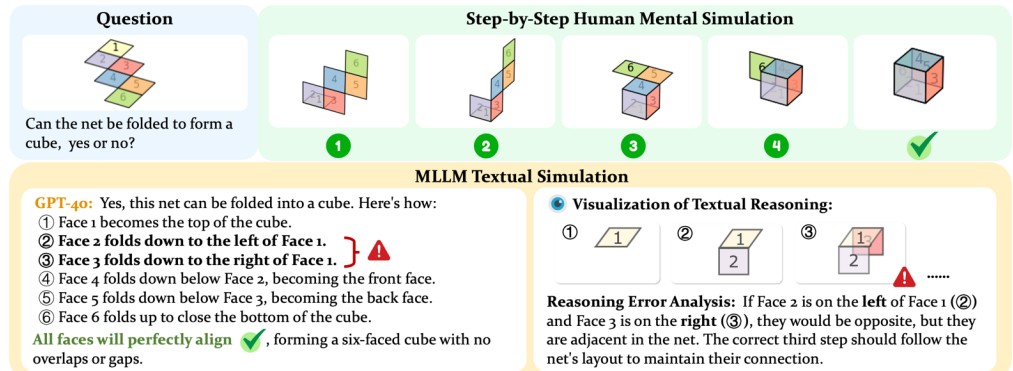

**Figure 1:** Visual simulations play a crucial role in real-world tasks, from assembling complex structures to interpreting mechanical diagrams and predicting spatial interactions. Different from how humans would approach a cube net folding problem, existing multimodal models rely heavily on textual simulation, which is not sufficient for reaching human-level spatial cognition. The above example shows how textual simulations of GPT-4o make obvious errors when we simulate the steps in 3D space.

## ABSTRACT

Spatial cognition is essential for human intelligence, enabling problem-solving through visual simulations rather than relying solely on verbal reasoning. However, existing AI benchmarks primarily assess verbal reasoning, neglecting the complexities of non-verbal, multi-step visual simulation. We introduce **STARE (Spatial Transformations and Reasoning Evaluation)**, a benchmark designed to evaluate multimodal large language models on tasks better solved through multi-step visual simulation. STARE features ∼4K tasks spanning foundational geometric transformations (2D and 3D), integrated spatial reasoning (cube net folding and tangram puzzles), and real-world spatial reasoning (perspective and temporal reasoning), reflecting practical cognitive challenges like object assembly, mechanical diagram interpretation, and everyday spatial navigation. Our evaluations show that models excel at reasoning over simpler 2D transformations, but perform close to random chance on more complex tasks like 3D cube net folding and tangram puzzles that require multi-step visual simulations. Humans achieve near-perfect accuracy but take considerable time (up to 28.0s) on complex tasks, reducing response time by 7.5 seconds on average with intermediate visual simulations. In contrast, models exhibit inconsistent performance gains from visual simulations, improving on most tasks but declining in specific cases like tangram puzzles (GPT-4o, o1) and cube net folding (Claude-3.5, Gemini-2.0 Flash), indicating that models cannot consistently leverage intermediate visual information. Even o3, a strong reasoning model, lags significantly behind human performance across tasks. By evaluating non-verbal visual reasoning beyond conventional text-based benchmarks, STARE highlights critical gaps in current AI spatial capabilities and sets a new standard for assessing spatial intelligence in multimodal models.

# 1 INTRODUCTION

Spatial reasoning is not merely a subset of human cognitive abilities but rather a fundamental underpinning of intellectual processes (Tversky & Suwa, 2009). Reasoning with space enables individuals to solve complex tasks through visually simulating transformations of objects in the mind, anticipating how their actions would physically manipulate other objects. Cognitive psychologists have found ample evidence that humans simulate 2D and 3D transformations to reason about spatial problems (Mitko & Fischer, 2020; Duan et al., 2022; Wai et al., 2009; Battaglia et al.). Shepard & Metzler (1971) found that the time taken by a subject to recognize two perspective drawings as the same 3D shape increases linearly with their angular difference in orientation, suggesting an analog *mental rotation* process. Hegarty (1992) found that humans employ *mental animation*, incrementally simulating the movement of parts to understand mechanical diagrams. Such abilities enable everyday tasks like assembling furniture, reading maps or instructional diagrams, and navigating new environments. They are also strongly correlated with success in STEM disciplines (Judd & Klingberg, 2021; Christensen & Schunn, 2009; Hegarty, 2004b).

Despite their prevalence in real-world applications—from arranging furniture in a house to molecular docking for drug discovery—*dynamic* visual simulations are still under-represented when evaluating multimodal large language models (MLLMs). Existing datasets largely target static recognition or problems that can be re-phrased as linguistic reasoning (Johnson, 2017; Zhang, 2019; Ji, 2022; Duan et al., 2021; Chollet, 2019; Ramakrishnan et al., 2024). In contrast, humans frequently solve spatial challenges—such as folding a 2D net into a 3D object, assembling a tangram, or taking another visual perspective—by running internal, step-wise *visual simulations* (Figure 1), which have a long pedigree in the cognitive-science literature on human spatial reasoning (Huttenlocher & Presson, 1973; Gunalp et al., 2019; Shepard & Feng, 1972; Preuss et al., 2024; Ayaz et al., 2012).

To bridge this gap, we introduce **STARE (Spatial Transformations and Reasoning Evaluation)**, a benchmark focused on spatial reasoning tasks that are better solved through multi-step visual simulations. STARE evaluates whether MLLMs can perform complex spatial reasoning akin to the visual simulations humans perform. It spans a spectrum of spatial cognition challenges (Figure 2), structured in increasing complexity:

- **Foundational geometric transformations**: Tasks involving basic planar (2D) and volumetric (3D) transformations, such as rotations, translations, scaling, and reflections.
- **Integrated spatial reasoning**: Cube net folding, requiring understanding of how 2D patterns fold into 3D objects, and tangram puzzles, which assess sequential assembly and spatial positioning of pieces.
- **Real-world spatial reasoning**: Tasks demanding reasoning about perspective changes and temporal frame sequences, simulating the spatial cognition challenges encountered in everyday scenarios.

In the first two categories, each transformation or operation (e.g., folding a face) can be explicitly visualized step by step, and indeed humans often draw or imagine intermediate states when solving them. The last category demands higher-level visual simulation skills without always having clear intermediate visual cues (e.g., perspective reasoning) (Bass et al., 2022; Chen et al., 2023). We carefully curate ~4K total instances across these categories, controlling difficulty via distractor similarity and number of simulation steps, to push models beyond superficial pattern-matching.

Our experiments show that models find reasoning over simple 2D transformations relatively easy but struggle with 3D cube net folding and tangram puzzles, performing at near random chance due to the need for multi-step simulations. Humans, though nearly perfect in accuracy, took significantly longer—up to 28.0 seconds—to solve some tasks but sped up considerably (down by 7.5 seconds on average) when given intermediate steps. Meanwhile, when models receive intermediate visual steps, their performance varies: *e.g.,* GPT-4o, Gemini-2.0 Flash Thinking and o1 improve while Gemini-2.0 Flash and Claude worsen on cube net folding, suggesting that not all models effectively utilize visual guidance. In general, even o3 lags significantly behind human performance. To better understand these gaps, we conduct detailed error analyses, pinpointing specific reasons for model failures, such as difficulties in accurately interpreting 3D spatial relationships, inability to "imagine in space", and struggles with extended visual contexts even when given explicit visual simulations. Fundamentally, models cannot effectively perform visual simulation.

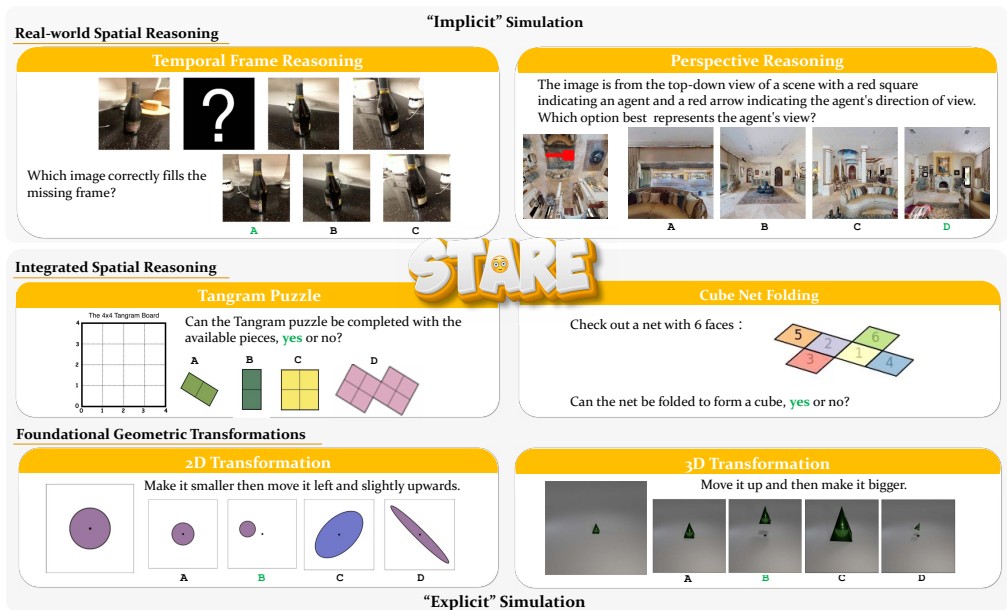

**Figure 2: Overview of STARE.** STARE consists of 3 levels of tasks, 2D Transformation and 3D Transformation for foundational spatial reasoning skills, tangram puzzle and cube net folding for integrated spatial reasoning, temporal frame inference and perspective reasoning to mimic real-world scenarios. The intermediate steps for completing tasks in the first two levels can be explicitly simulated, while the more real-world spatial reasoning tasks require more abstract and implicit mental simulations.

Overall, STARE aims to comprehensively test MLLMs' ability to perform sequential visual simulations as opposed to pure textual reasoning. By evaluating models on tasks grounded in cognitive phenomena like mental imagery, we aim to reveal whether current MLLMs can approach the flexible spatial problem-solving of humans.

## 2  THE STARE BENCHMARK

STARE is designed to evaluate multimodal models' abilities in spatial cognition and visual reasoning, focusing specifically on tasks that humans solve non-linguistically, through visual simulation. Current perception-focused multimodal benchmarks still rely heavily on linguistic reasoning (Fu* et al., 2024; Lu et al., 2021; Li et al., 2024a) or static visual recognition (Tong et al., 2024; Wu & Xie, 2023; Fu et al., 2024), failing to measure models' abilities in sequential visual problem-solving. Parallel work in spatial cognition (Yiu et al., 2024; Zhang, 2019; Hu et al., 2021; Ramakrishnan et al., 2024; Rismanchian et al., 2024) probes analogy making and pattern induction, yet simulation is optional and intermediate visual states are seldom provided because of annotation cost. VSI-Bench (Yang et al., 2024) underscores the role of mental imagery in spatial reasoning, but focuses on spatial memory and estimation from video rather than explicit step-by-step simulation. STARE closes the gap by testing multimodal models across diverse spatial tasks that require step-by-step visual simulations with or without explicit linguistic guidance. We describe the overall design of STARE (§2.1), highlighting key differences compared to existing benchmarks. We then provide detailed descriptions of each task, discussing how the data was curated (§2.2).

### 2.1  OVERVIEW OF STARE

STARE is structured to comprehensively cover spatial reasoning at multiple complexity levels, from basic geometric transformations (2D and 3D) to more integrated tasks (cube net folding and tangram puzzles) and real-world spatial reasoning scenarios (temporal frame and perspective reasoning). Each task is presented as a multiple-choice or yes/no question using carefully designed visual and textual prompts. In total, the dataset contains ∼4K instances across different evaluation setups (Figure 3). Detailed statistics of STARE are provided in Appendix Figure 12.

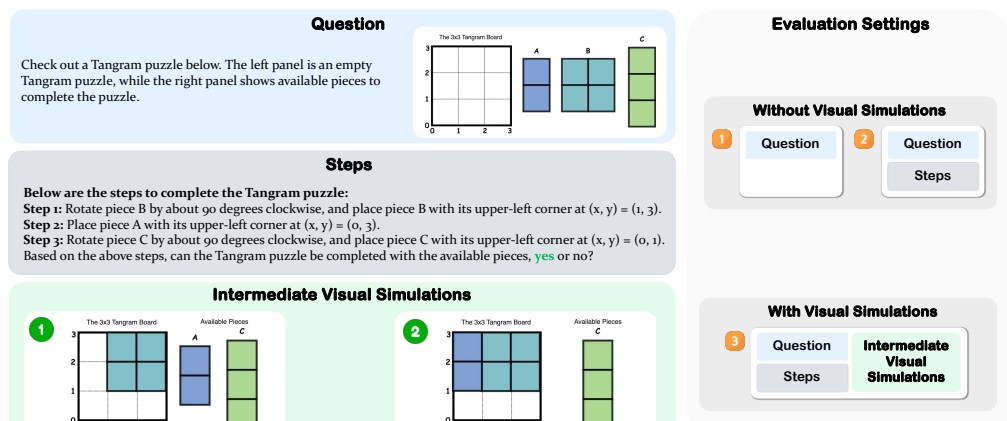

**Figure 3: The different variants in the Tangram Puzzle task.** We provide visualizations of the complete interleaved inputs for all three types in Appendix G.2.

STARE separates tasks that can be visually simulated, i.e., where each transformation step is visually observable, from tasks demanding more abstract and implicit mental simulations, such as perspective reasoning. To support more fine-grained evaluation, we synthesize the tasks that humans can mentally picture or even explicitly draw the intermediate steps, including 2D transformations, 3D transformations, cube net folding and tangram puzzle. Additionally, STARE tasks are intentionally crafted to closely reflect real-world scenarios such as assembling objects (e.g., tangram puzzles), interpreting mechanical diagrams (e.g., cube net folding) and navigating environments (e.g., perspective reasoning). These scenarios can potentially shed lights on models' abilities in practical, everyday spatial cognition, providing meaningful assessments aligned with common human challenges. A detailed discussion of related works in human visual reasoning and multimodal LLM benchmarks is provided in Appendix D.

## 2.2 Data curation

**2D transformations:** We design two types of tasks assessing spatial reasoning through two-dimensional shape transformations: visual analogy, and instruction-based tasks. In visual analogy tasks, a shape $A$ is shown to transform visually into shape $A'$, after which a shape $B$ is provided with candidate shapes for applying the same transformation sequence to $B$. Instruction-based tasks explicitly describe transformations (e.g., "Rotate 90 degrees clockwise, then make it bigger") and require selecting the correctly transformed shape from 4 answer choices. Transformations include rotations, translations, uniform scaling, reflection and shearing, with clearly defined parameters. Each task is created with three difficulty levels: easy (with two distractors out of three clearly different in appearance), medium (one obvious distractor), and hard (all distractors visually similar, forcing the model to pay attention to the transformation itself). In addition, we synthesize samples with 1/2/3 transformation steps to facilitate evaluations in multi-turn visual transformations. We programmatically generate all shapes and their transformed versions using Matplotlib (Matplotlib, 2012). Visualizations of different 2D transformation variants are provided in Appendix Figures 13 and 14.

We develop two experimental setups: (1) **question + transformation steps**, where the transformation steps are shown either verbally (for instruction-based tasks) or visually (for visual analogy tasks); and (2) **question + transformation steps + intermediate visual simulations**, showing all intermediate visualizations of shape $B$, excluding the final step. We synthesize a total of ∼1000 instances, ∼600 of which are without intermediate visual simulations.

**3D transformations:** We extend the 2D transformation tasks to three dimensions, creating similar tasks using 3D shapes. Reflection is omitted in 3D because the mirror plane isn't obviously recognizable to human evaluators. The transformations include rotations around arbitrary axes, translations in 3D space, scaling, and shearing. Tasks, difficulty levels, and experimental setups mirror those of the 2D tasks, with a total of ∼1000 instances. Following (Johnson et al., 2017), we create abstract 3D shapes as detailed meshes and use Blender (Blender) to render realistic and consistent visuals.

**Tangram puzzles:** Tangram puzzles test spatial reasoning about how individual pieces fit together to form a complete shape. Each puzzle provides a target grid and pieces, and the task is to determine whether the pieces can exactly fill the grid. Valid puzzles were generated by randomly dividing small grids (3x3 or 4x4) into rectangular or square shapes, then randomly rotated. Irregular variants were also created by merging adjacent rectangles. Invalid puzzles were constructed by adding or removing pieces, altering piece sizes, or giving incorrect placement instructions.

We create three setups for evaluation: (1) **question-only**, which presents the initial puzzle configuration with a query about solvability; (2) **question + assembly steps**, adding descriptive instructions of each assembly step without visual aids; and (3) **question + assembly steps + intermediate visual simulations**, providing both descriptive annotations and intermediate visualizations of the assembly process, excluding the final visualization indicating success or failure. This task comprises ∼800 puzzles, evenly divided into solvable and unsolvable instances.

**Cube net folding:** This task evaluates the model's capacity to mentally fold flat 2D patterns into 3D cubes. We provide examples comprising both valid nets (correctly folding into a cube) and invalid nets (leading to overlapping or disconnected faces). Each cube net has explicitly labeled faces. To generate these examples, we implement a step-by-step algorithm that simulates the folding process by designating a stationary base face and sequentially folding the connected faces. During each folding step, we detect and annotate errors, such as overlaps or disconnected faces, and generate corresponding visualizations using Matplotlib, clearly delineating face boundaries. Similar to tangram puzzles, we evaluate models in three setups, including (1) question-only, (2) question + folding steps, and (3) question + folding steps + intermediate visual simulations. The final cube net folding task contains ∼320 samples, balanced between valid and invalid configurations.

**Temporal frame reasoning:** This task evaluates a model's ability to infer missing sequential visual information. Each example consists of four consecutive frames from a video, with one frame hidden. The model must identify the missing frame from a set of three options, relying on temporal consistency and logical scene progression.

We construct 471 examples from the Objectron (Ahmadyan et al., 2021) dataset, which contains short, object-centric videos with camera pose annotations. To create meaningful sequences, we extract the longest continuous segment where the camera moves only in one direction (left or right), divide it into four equal intervals, and select a frame from the central portion of each interval. One of these frames is hidden, and the model must identify it from three choices: the correct missing frame and two distractor frames sampled from different, non-overlapping parts of the video.

**Perspective reasoning:** This task assesses a model's ability to understand how scenes appear from different viewpoints. Each example consists of a top-down map that indicates an agent's position and orientation, represented by an arrow showing the agent's viewing direction. The model must then select the correct first-person view from four choices, emphasizing spatial perspective reasoning and spatial relationships in various indoor environments.

We construct 250 samples using the HM3D dataset (Ramakrishnan et al., 2021), a large collection of 3D indoor spaces derived from real-world environments. To generate each example, we use the Habitat simulator (Savva et al., 2019; Szot et al., 2021; Puig et al., 2023) to place an agent at a random position on the floor while ensuring the surrounding scene contains enough visual cues, such as objects and structures, rather than just walls. A top-down view of the agent's position is then captured, and a random viewing direction is assigned (forward, right, left, or backward). The four answer choices correspond to these fixed 90-degree viewpoints, ensuring clear distinctions between them. To improve dataset quality, we manually remove ambiguous cases and low-resolution images.

## 3 EXPERIMENTS

In this section, we describe our experimental setup in detail, present comprehensive results, and provide an in-depth analysis of common model errors and limitations.

### 3.1 EXPERIMENTAL SETUP

For synthetic tasks involving explicit simulations (2D transformations, 3D transformations, cube net folding, tangram puzzles), we explore two evaluation settings:

| Model | 2D Trans. | | 3D Trans. | | Cube Net | | Tangram | | Temp-oral | Pers-pective | Overall |
|---|---|---|---|---|---|---|---|---|---|---|---|
| | ✗VSim | ✓VSim | ✗VSim | ✓VSim | ✗VSim | ✓VSim | ✗VSim | ✓VSim | | | |
| Random | 25.0 | 25.0 | 25.0 | 25.0 | 50.0 | 50.0 | 50.0 | 50.0 | 33.3 | 25.0 | 34.8 |
| *Closed-source Models* | | | | | | | | | | | |
| GPT-4o | 71.2 | 82.7 (↑ 11.5) | 65.5 | 68.4 (↑ 2.9) | 50.3 | 52.2 (↑ 1.9) | 52.5 | 51.5 (↓ 1.0) | 39.0 | 38.7 | 53.9 |
| Claude-3.5 Sonnet | 65.9 | 71.4 (↑ 5.5) | 51.5 | 57.8 (↑ 6.3) | 52.3 | 51.6 (↓ 0.7) | 59.0 | 67.6 (↑ 8.6) | **54.0** | 26.1 | 53.1 |
| Gemini-2.0 Flash | 69.5 | 75.2 (↑ 5.7) | 56.1 | 59.3 (↑ 1.6) | 37.7 | 35.6 (↓ 2.1) | 65.0 | 65.5 (↑ 0.5) | 38.6 | 37.2 | 51.3 |
| Gemini-2.0 Flash Think | 60.6 | 62.8 (↑ 2.2) | 49.5 | 56.1 (↑ 6.6) | 48.3 | 50.7 (↑ 2.4) | 39.8 | 62.8 (↑ 23.0) | 45.0 | 32.7 | 48.8 |
| o1 | 81.8 | 87.7 (↑ 5.9) | 67.9 | 71.6 (↑ 3.7) | 51.3 | 53.4 (↑ 2.1) | 55.3 | 53.2 (↓ 2.1) | 45.0 | 36.8 | 57.2 |
| o3 | **87.5** | **89.3** (↑ 1.8) | **75.2** | **78.4** (↑ 3.2) | **68.4** | **79.4** (↑ 11.0) | **68.6** | **82.1** (↑ 13.5) | 51.4 | **42.8** | **68.1** |
| *Open-source Models* | | | | | | | | | | | |
| LLaVA-OneVision-72B | 32.9 | 32.2 (↓ 0.7) | 27.0 | 30.6 (↑ 3.6) | 28.5 | 34.2 (↑ 3.7) | 30.3 | 39.8 (↑ 9.5) | 35.7 | 24.8 | 31.4 |
| InternVL2.5-78B | 47.5 | 50.1 (↑ 2.6) | 38.1 | 36.5 (↓ 1.6) | 37.1 | 37.3 (↑ 0.2) | 60.7 | 48.2 (↓ 12.5) | 31.4 | 26.0 | 39.2 |
| Qwen2.5-VL-3B | 16.6 | 20.0 (↑ 3.4) | 29.1 | 31.4 (↑ 2.3) | 43.5 | 41.0 (↓ 2.5) | 50.1 | 42.7 (↓ 7.4) | 33.3 | 23.3 | 32.3 |
| Qwen2.5-VL-7B | 35.4 | 32.4 (↓ 3.0) | 28.8 | 31.7 (↑ 2.9) | 40.7 | 44.9 (↑ 4.2) | 54.5 | 52.9 (↓ 1.6) | 36.5 | 23.2 | 36.7 |
| Qwen2.5-VL-72B | 45.2 | 48.5 (↑ 3.2) | 43.0 | 49.1 (↑ 6.1) | 35.2 | 53.4 (↑ 18.2) | 61.2 | 56.9 (↓ 4.3) | 31.4 | 26.0 | 42.3 |
| *Human Performance* | | | | | | | | | | | |
| Accuracy | 96.8 | 98.6 (↑ 1.8) | 94.6 | 96.9 (↑ 2.3) | 98.3 | 98.9 (↑ 0.6) | 91.5 | 95.8 (↑ 4.3) | 99.0 | 98.1 | 97.1 |
| Response Time (s) | 14.2 | 11.0 (↓ 3.2) | 17.1 | 12.5 (↓ 4.6) | 13.7 | 5.2 (↓ 8.5) | 28.0 | 10.1 (↓ 17.9) | 16.2 | 18.4 | - |
| Δ(Best Model, Human) | -9.3 | -9.3 | -19.4 | -18.5 | -29.9 | -19.5 | -22.9 | -13.7 | -45.0 | -55.3 | -29.0 |

**Table 1:** Model Performance With or Without Visual Simulation (VSim) Across Tasks in STARE. Even the top performer, o1, achieves just under 60% accuracy. Humans, in contrast, get near perfect scores. Green (Red) arrows indicate performance improvements (degradations) with visual simulation.

- *Without Visual Simulations:* Models receive only an initial image with or without step-by-step textual instructions and had to mentally infer the subsequent transformations without visual guidance, thereby testing their internal mental simulation capabilities.

- *With Visual Simulations:* Models were provided with step-by-step visualizations clearly illustrating each transformation step before the final result, enabling explicit visual reasoning. Instead of collating the complex step-by-step visualizations into a single image, we provide the model with interleaved image and text query for evaluation.

For real-world reasoning tasks, including temporal frame and perspective reasoning, we evaluate models under the standard single image setting without providing explicit intermediate visual steps.

**Evaluation Metrics.** We report accuracy for multiple-choice questions in 2D/3D transformations, temporal frame, and perspective reasoning tasks. For cube net folding and tangram puzzles, which involve binary yes/no questions, we report the F1 score. We report macro-average performance across tasks as the overall evaluation metric.

**Models.** We consider the following models: **(1) Closed-source models**: GPT-4o (OpenAI), Claude-3.5 Sonnet (Anthropic), Gemini2.0 Flash (Deepmind, a), and the reasoning-focused Gemini2.0 Flash Thinking (Deepmind, b), o1 (OpenAI et al., 2024) and o3 (OpenAI, 2025). **(2) Open-source models**: InternVL2.5-78B (Chen et al., 2024), LLaVA-OneVision-72B (Li et al., 2024b), Qwen2.5-VL-3B, Qwen2.5-VL-7B, and Qwen2.5-VL-72B (Bai et al., 2025).

Additionally, we invite 5 undergraduate students to complete the same tasks as the models. The averaged performance and response time are recorded to benchmark model capabilities against human-level spatial reasoning, with additional human evaluation results provided in Appendix H.

## 3.2 MAIN RESULTS

The results present in Table 1 show notable variations in model performance across different spatial reasoning tasks in the STARE benchmark. Models achieve the highest accuracy (up to 89.3%) on simpler 2D transformation tasks, significantly surpassing random chance (25%). Accuracy decreases by roughly 5% on average for more complex 3D transformations. Tasks involving intricate multi-step reasoning, such as cube net folding and tangram puzzles, resulted in even worse model performance without visual simulation. Additionally, temporal frame reasoning and perspective reasoning, which require interpreting sequential visual contexts and viewpoint changes, posed considerable difficulties, with most models performing similarly to random chance.

The use of visual simulations (VisSim) enhances model performance in most cases, but not all. GPT-4o exhibits a notable improvement of 11.5% accuracy on 2D transformations with visual simulations, and Claude-3.5 Sonnet shows significant gains (+8.6%) on tangram puzzles. However, visual simulations did not uniformly benefit model performance; certain models like Gemini-2.0 Flash experienced slight performance declines (e.g., a 2.1% decrease on F1 for cube net tasks), indicating that models can not always effectively leverage intermediate visual information. The latest reasoning-focused o3 model outperforms all other models with visual simulations. Overall, it improves over GPT-4o by 14.2% on average, but still lag behind human performance. Notably, o3 seems to be better at leveraging visual simulations. However, humans show relatively small performance gaps between conditions with and without visual simulation, indicating they can mentally simulate transformations effectively even without explicit visual aids.

Open-source models generally exhibit lower accuracy compared to closed-source counterparts, highlighting a significant performance gap. Larger models like InternVL2.5-78B and Qwen2.5-VL-72B performe relatively better, suggesting benefits from scale, but their results with visual simulations also varied. For instance, InternVL2.5-78B's performance decreases significantly in tangram tasks (-12.5%), whereas Qwen2.5-VL-72B improves notably (+18.2%) in cube net folding.

Human performance consistently surpasses that of models, achieving high accuracy across all STARE tasks, and further improved by intermediate visual simulations. However, these tasks were cognitively demanding even for humans, reflected by relatively long response times (e.g., 28.0 seconds on tangram puzzles without visual simulations). Although intermediate visual simulations significantly reduces cognitive load and response time, humans still require more than 5 seconds to mentally manipulate and reason through these problems and complete the last step. Thus, STARE tasks clearly involve complex, multi-step spatial reasoning beyond simple recognition tasks solvable at a glance (Fu et al., 2024). These findings underscore humans' superior spatial reasoning capabilities, particularly when aided by visual simulations.

Moreover, to study whether gains on abstract, synthetic spatial tasks translate to real-world tasks, we computed model-level correlations between the two domains. Concretely, for each model, we average its performance across with or without visual simulation on the 4 synthetic tasks and contrast that with its mean accuracy on the two real-world tasks. This yields a strong overall Pearson correlation ($r \approx 0.88$, $p \approx 5e^{-4}$) across all 11 models. Including human performance further increases the correlation to $r \approx 0.97$ ($p \approx 1e^{-7}$).

### 3.3 DETAILED ANALYSIS

To gain deeper insights into model limitations and identify specific reasoning challenges, we structure our detailed analysis around several targeted questions. We focus on GPT-4o, which achieves the best overall performance among non-reasoning models, with extended analysis on other models in Appendix H.

**Q1: How well do models understand individual transformation types in 2D and 3D?** We evaluate model accuracy on individual transformation operations—rotation, translation, scaling, reflection, and shearing—for both 2D and 3D tasks, comparing performance with and without visual simulation (Figure 4). For 2D tasks, scaling achieves the highest accuracy (approximately 90% without visual simulation), improving further with visual simulation. Shearing was the most challenging in 2D (around 54%), showing minimal improvement from visual aids. Reflection, rotation, and translation significantly benefits from visual simulation, improving roughly 10 percentage points each. In 3D tasks, translation had the highest accuracy (about 76% without visual simulation), although it slightly declines with visual simulation. However, shearing, scaling, and rotation notably improve with visual simulation by about 3–8 percentage points. Overall, visual simulation substantially enhances performance for complex transformations, especially in 2D, though the added complexity of 3D transformations continues to present significant challenges.

**Q2: How does model accuracy change as task complexity increases?** **(1) Performance vs. Difficulty-level**: The left sub-figure in Figure 6 shows model accuracy decreased as tasks became harder. For 2D tasks, GPT-4o performed best on easy tasks (∼86% with visual simulation), with accuracy declining notably for medium and hard tasks, especially without visual simulation (dropping to ∼66% for hard tasks). For 3D tasks, overall accuracy was lower, decreasing from easy tasks (∼72% without visual simulation) to hard tasks (∼60% without). Visual simulation generally im-

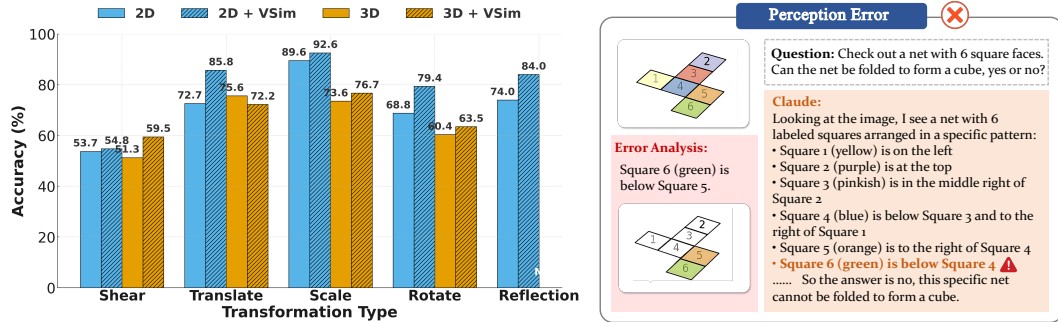

**Figure 4:** GPT-4o performance on individual 2D/3D transformation types, with and without Visual Simulation (VSim).

**Figure 5:** A perception error from Claude-3.5 Sonnet. See Appendix G.5 for more case study.

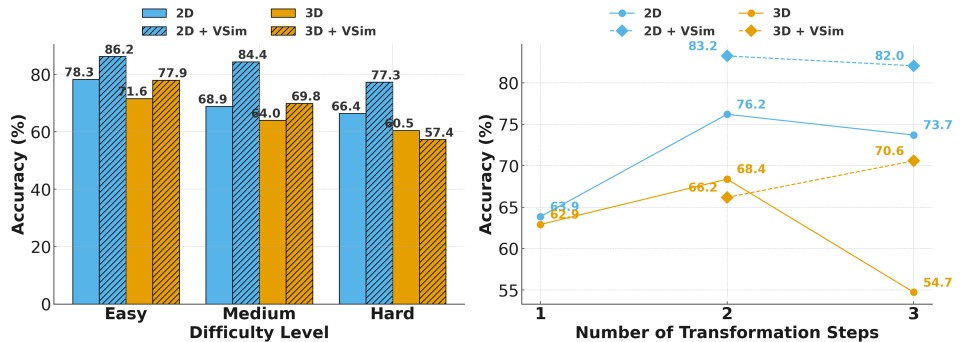

**Figure 6:** GPT-4o performance vs. task complexity (left: difficulty levels and right: number of transformation steps) with or without Visual Simulation (VSim).

proved accuracy but was less effective or even slightly detrimental for the hardest 3D tasks (60.5% without, 57.4% with). **(2) Performance vs. Number of Turns**: The right sub-figure in Figure 6 shows that how model performance varies with the number of transformation steps ($N$ = 1, 2, 3). Without visual simulation, accuracy for both 2D and 3D tasks initially increases from $N$ = 1 to $N$ = 2, and then decreases at $N$ = 3. The observed peak at $N$ = 2 likely occurs because two-step transformations combine simpler transformations (e.g., scaling) with more challenging ones (e.g., shearing), allowing models to leverage the simpler transformations to determine the correct answer. In contrast, one-step transformations are evenly distributed across all transformation types, while at $N$ = 3, the increased complexity from multiple transformations compounds cognitive demands, reducing overall model accuracy. With visual simulation, accuracy remains consistently high across 2 and 3 steps in 2D tasks and shows stable or slightly improved performance at $N$ = 3 in 3D tasks (performance at $N$ = 1 with visual simulation is omitted as there is no intermediate step).

**Q3: Do model failures originate from basic visual perception errors?** To determine if model failures originate from fundamental visual perception rather than higher-level reasoning limitations, we design a straightforward probing experiment. Specifically, we simplify the task by directly presenting the model with the final, fully simulated outcomes, reducing the problem to visually matching these outcomes to the correct candidate answers. Under these conditions, accuracy improves by 4.2% (from 82.7% to 86.9%) on 2D transformations and 2.8% (from 68.4% to 71.2%) on 3D transformations, indicating only a modest improvement when eliminating intermediate steps. However, for more structured tasks like cube net folding and tangram puzzles, providing the fully completed final form drastically raises accuracy to 100% and 91.6%, respectively, highlighting that models can solve these tasks when the perceptual complexity is minimized. To further isolate the nature of perceptual errors in cube net folding, we create targeted tasks to test both 2D perception (color recognition and face connectivity) and 3D perception (identifying if a face has been folded). Results from these tasks (Table 2) reveal perfect color recognition but a notable decrease in accuracy for face connectivity (94.1%) and particularly low accuracy in correctly identifying folded faces (57.4%). Figure 5 illustrates an example of perception error on connectivity misalignments from Claude-3.5 Sonnet. Moreover, these specific perceptual errors in folding explain the limited benefits from visual simulations observed in Table 1 for GPT-4o. Overall, while some errors indeed stem

| Model | 2D Perception | | 3D Perception |
|---|---|---|---|
| | Color | Connectivity | Folded? |
| GPT-4o | 100.0 | 94.1 | 57.4 |

**Table 2:** 2D and 3D perception accuracy in cube-net folding.

| Input | 2D Trans. | 3D Trans. | Cube Nets | Tangram |
|---|---|---|---|---|
| Text-only | 87.5 | 64.7 | 57.0 | **72.6** |
| Image-only | 75.1 | 67.7 | 56.0 | 62.5 |
| Image+Text | **90.8** | **70.0** | **62.1** | – |

**Table 3:** GPT-4o performance without visual simulation under different input representations.

| Input | Cube Nets | Tangram |
|---|---|---|
| Question-only | 50.2 | **62.4** |
| Question+Steps | **50.4** | 34.7 |

**Table 4:** GPT-4o performance with question-only vs. explicit reasoning steps.

| Simulation State | 2D Trans. | 3D Trans. | Cube Nets | Tangram |
|---|---|---|---|---|
| Partial | 86.8 | **72.1** | 51.3 | 43.5 |
| All | 82.7 | 68.4 | **52.2** | **51.5** |
| Last | **89.4** | 68.4 | 35.2 | 43.4 |

**Table 5:** GPT-4o performance with different intermediate visual-simulation states.

from basic visual perception deficits, particularly in more complex 3D scenarios, the results suggest higher-level reasoning also plays a large role in overall model failures. We further extend the perception probing to 2D/3D transformations and tangram puzzles (Appendix H); models score well above chance on most basic perceptual subtasks, yet still fail on the full reasoning tasks, confirming that multi-step spatial simulation, rather than low-level perception, is the core bottleneck. See Appendix E and G.5 for additional quantitative and qualitative error analysis.

**Q4: How well do models reason spatially in text?** To evaluate how well models reason spatially from text alone, we translate each visual task into clear, concise descriptions. For *2D and 3D transformation tasks*, each object is described by stating its shape, color, position, size and etc.—for instance, *"a red square at position (3,4) with size 2"*. In the *cube-net folding task*, the unfolded cube is represented by numbering each face and arranging these numbers in a grid matching the cube net's visual layout. For example, "123456" represents all six faces in a single row. Lastly, for the *tangram puzzle task*, each piece is labeled (e.g., "Piece A") and represented by a compact grid indicating occupied cells marked by "1". For instance, a square piece might be shown as two rows of "11". Examples of the text representations for each task are shown in Appendix G.4.

As shown in Table 3, providing the model with a text representation removes much of the perception challenge, yet accuracy remains well below human performance—about 57% on cube-net folding, 65% on 3D transformations, and roughly 73% on tangram puzzles, suggesting that the model still lacks the ability to mentally simulate the steps to solve each task. Text helps most on 2D spatial reasoning: accuracy on 2D transformations rises from 75% with images alone to 87% with text, and tangram performance climb from 63% to 73%. For tasks involving 3D spatial reasoning, however, text gives little benefit, partly because the simple text description about shape, color, material, center, and size, cannot capture all the depth and adjacency cues in 3D spatial reasoning. We observe similar trends with o3 (Appendix Table 22), confirming that while text helps on language-friendly tasks, strong 3D spatial performance still depends critically on visual perception and cannot be substituted by textual descriptions alone.

**Q5: How well do models verbally simulate without visual simulation?** We evaluate how effectively models verbally simulate spatial reasoning without intermediate visual simulations by comparing performance when provided only the question (Question-only) versus explicit verbal reasoning steps (Question+Steps). Table 4 shows minimal improvement in cube net folding (50.2% to 50.4%), indicating limited benefit from verbal reasoning alone. Conversely, tangram performance notably decreases (62.4% to 34.7%), suggesting models adopt shortcuts like summing piece areas rather than genuine spatial simulation. This result partially reflects a bias in our question-only set: models can achieve ∼75% accuracy by simply checking total areas of available pieces rather than reasoning about spatial arrangement.

**Q6: How well do models integrate textual context with isolated visual simulations?** We compared accuracy when presenting models with complete visual sequences versus only the final or most relevant visual state (Table 5). Easier tasks like 2D and 3D transformations showed improved or comparable accuracy when presented only the final state (e.g., 82.7% for complete vs. 89.4% for last), suggesting that for these tasks, the final visual state closely resembles the initial state, reducing cognitive load. However, in complex tasks such as cube net folding (52.2% complete vs. 35.2% last) and tangram puzzles (51.5% complete vs. 43.4% last), the final state becomes more disconnected from the initial configuration, requiring deeper understanding of preceding verbal steps.

This disconnection introduces significant challenges for models, aligning with earlier findings (Q4) and underscoring their difficulties in integrating complex visual sequences during multi-step reasoning. Beyond synthetic tasks, we explored extending visual simulations to real-world settings (Appendix E): perspective reasoning benefits strongly from intermediate views (GPT-4o: 38.7% $\rightarrow$ 76.7%), while temporal reasoning shows negligible improvement. A pilot study on ALFRED robot planning tasks (Appendix E) further shows that visual simulations substantially improve task planning accuracy, suggesting practical value for embodied AI applications. See Appendix E for additional analysis on the impact of visual simulation granularity.

## 4 CONCLUSION

We introduced STARE, a benchmark of $\sim$4K instances that evaluates multimodal models on spatial cognition tasks requiring step-by-step visual simulation. Our experiments across 11 models show that visual simulations substantially improve performance on structured tasks but yield inconsistent gains on complex multi-step reasoning. Even o3, the strong reasoning model, lags behind humans and relies on externally provided simulations rather than internal spatial reasoning. Our analysis identifies multi-step spatial simulation as the core bottleneck: models handle isolated perceptual subtasks well but struggle to chain multiple visual transformations into coherent spatial reasoning.

**Limitations and Future Work.** STARE's synthetic tasks enable controlled evaluation but may not capture the full complexity of real-world spatial reasoning. Pilot studies on robot planning and mechanical diagram interpretation (Appendix E) suggest that visual simulation benefits extend to practical domains. We also find that current interleaved generation models show limited ability to self-generate task-aligned visual simulations (Appendix E), though this may improve with more capable unified models. Future work should extend the benchmark to more diverse real-world domains and investigate whether explicitly training models to perform visual simulation can effectively close the gap with human-level spatial cognition.

## 5 ETHICS STATEMENT

STARE provides a standardized way to measure AI capabilities in spatial reasoning tasks, potentially guiding research toward AI systems that can better support robotics, autonomous driving, augmented reality, and educational applications. However, improved spatial reasoning could also lead to negative societal impacts if misused, such as enhanced surveillance capabilities or military applications. Additionally, the synthetic nature of STARE may introduce biases toward simplified or artificial scenarios, limiting direct applicability to diverse real-world conditions. Future versions should aim to include more realistic and diverse datasets and consider ethical guidelines to minimize potential risks and ensure fair, positive societal outcomes.

## 6 REPRODUCIBILITY STATEMENT

We have taken substantial steps to ensure the reproducibility of our results. All experimental settings are described in detail in Appendix G. We provide complete documentation of the statistics of STARE and the design spaces for all synthetic tasks, including 2D transformations, 3D transformations, cube net folding, and tangram puzzles in Appendix F. Data curation and evaluation code is included in the supplementary material to facilitate verification and reuse. Together, these resources enable the community to reproduce our experiments and extend our findings.

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

## A    OVERVIEW OF THE APPENDIX

This Appendix is organized as follows:

- Section B discusses the use of LLMs.
- Section C discusses the limitations of STARE.
- Section D presents an extended discussion about related works.
- Section F details the statistics of STARE and the design spaces for all synthetic tasks, including 2D transformations, 3D transformations, cube net folding, and tangram puzzles.
- Section E provides additional analysis complementary to Section 3, including detailed error analysis, impact of # visual simulations and reasoning efforts, additional results on human evaluation, and new experiments on open-ended answering, interleaved image generation, visual simulation for real-world tasks, downstream robot planning, and mechanical diagram interpretation.
- Section G describes the experimental setup, covering the prompts used, model configurations, hyperparameter settings, and presents full visualizations of different experimental settings (e.g., evaluation settings with or without visual simulations, perception probing questions).
- Section H provides results on additional models for analysis conducted in Section 3, including extended perception probing results and o3 results on text representations.

## B    THE USE OF LARGE LANGUAGE MODELS

We used large language models (LLMs) as auxiliary tools during manuscript preparation, but only for surface-level editing such as grammar correction, minor rephrasing, and stylistic refinements to improve readability. AI-assisted coding was employed in curating synthetic data, but under strict human supervision and review. In addition, LLMs served as judges in our detailed error analysis; however, we manually reviewed a subset of their outputs to verify accuracy. All research ideas, methodologies, experiments, and conclusions were conceived and executed exclusively by the authors.

## C    LIMITATIONS

Although STARE provides valuable insights, it still has several limitations. First, it uses simplified synthetic images that do not fully represent real-world complexity; future versions could include realistic or dynamic scenes with clutter and occlusion. Second, it focuses only on rigid shape transformations; adding tasks involving flexible shapes, articulated objects, or additional sensory cues (such as audio or depth) would cover a wider range of spatial reasoning skills. Lastly, multiple-choice scoring hides intermediate reasoning steps; extending evaluations with explanations, step-by-step checks, or open-ended responses would give more detailed insights, which we briefly explore in Appendix E.

Still, STARE 's current design has clear strengths. The simplified images isolate spatial reasoning from general object recognition tasks. Its structured variety of tasks helps pinpoint specific model difficulties. Automatic scoring ensures consistent and easy-to-scale evaluations. Modular task presentations (image-only, text-only, image+text prompts) let researchers analyze individual modality contributions. Additionally, synthetic data makes STARE easily reproducible, accessible, and extensible. Overall, STARE is a strong first step toward measuring multimodal spatial reasoning, with clear pathways toward more realistic and comprehensive future benchmarks.

## D    RELATED WORK

**Human visual reasoning.** Human visual reasoning relies on two complementary faculties: *relational analogy*—mapping abstract structures across scenes—and *mental simulation*—predicting future states through incremental transformations. Structure–Mapping Theory (Gentner, 1983) and

analyses of Raven's Progressive Matrices (Carpenter et al., 1990) first showed that success in visual problem-solving hinges on aligning relations rather than surface features. Computational accounts echo this claim: explicit relational models reproduce human-like performance (Lovett & Forbus, 2017), whereas modern deep networks still struggle with visual analogy tasks (Ichien et al., 2021; Webb et al., 2022; Ichien et al., 2023).

Mental simulation complements analogy-making. Classic work on mental rotation (Shepard & Metzler, 1971) and mechanical reasoning (Hegarty, 2004a) demonstrates that people mentally "run" transformations, consistent with grounded-cognition theories (Barsalou, 2008). Intuitive-physics studies cast the mind as a noisy physics engine that combines object-centric structure with probabilistic dynamics (Battaglia et al., 2013; Tenenbaum et al., 2006; Ullman et al., 2017). Object-based predictive-coding models such as PLATO extend these ideas, achieving human-like physical prediction and developmental trajectories (Yang et al., 2023; Piloto et al., 2022). Simulations are also *selective*: people allocate attention "just in time," focusing on the most diagnostic elements instead of exhaustively modeling the entire scene (Bass et al., 2022; Bear et al., 2022; Chen et al., 2023).

Together, these findings suggest that effective problem-solving hinges on the ability to carry out step-by-step visual simulations; our benchmark therefore probes whether multimodal models can effectively leverage or even produce such simulations and exhibit *human-like visual reasoning* on sequential, compositional tasks.

**Multimodal evaluation benchmarks.** Recent advances in evaluating multimodal large language models have led to the development of benchmarks targeting diverse aspects of visual reasoning. Early benchmarks such as VQA (Antol et al., 2015) and CLEVR (Johnson, 2017) focus on compositional reasoning and general visual question answering. However, more challenging benchmarks, such as MMMU (Yue et al., 2023) and Humanity's Last Exam (HLE) (Phan et al., 2025), assess expert-level, domain-specific reasoning using complex multimodal inputs, where state-of-the-art models achieve only around 60% on MMMU-pro (Yue et al., 2024) and below 20% on HLE.

In response to the growing demand for robust evaluation, several new benchmarks (Fu* et al., 2024; Lu et al., 2021; Li et al., 2024a; Tong et al., 2024; Wu & Xie, 2023) have been introduced. For example, M3Exam repurposes multilingual professional-license questions (Zhang et al., 2023). MME (Fu et al., 2023) and MMBench (Liu et al., 2024) separate low-level perception from higher-level cognition. BLINK (Fu et al., 2024) departs from pure linguistic reasoning tasks to include tasks grounded in core computer vision capabilities, including relative depth estimation, semantic correspondence, visual similarity assessment, inpainting, etc. Improvements on BLINK require the use of perception tokens (Bigverdi et al., 2024), which generate latent intrinsic images to reason, demonstrating for the first time, that reasoning doesn't have to be linguistic. In this work, we build upon this finding, targeting primarily visual reasoning that can be better solved with visual cues.

The most relevant benchmarks to ours are KiVA (Yiu et al., 2024), RAVEN/I-RAVEN (Zhang, 2019; Hu et al., 2021), SPACE (Ramakrishnan et al., 2024), and TurtleBench (Rismanchian et al., 2024), which primarily evaluate static analogy or pattern induction, where intermediate visual simulations are optional and often infeasible to curate. VSI-Bench (Yang et al., 2024) emphasizes mental imagery in spatial reasoning but centers on spatial memory and distance estimation from video. Other recent efforts—such as Forgotten Polygons (Rudman et al., 2025), GeoGramBench (Luo et al., 2025), and VisuLogic (Xu et al., 2025)—target more isolated failures in visual reasoning, including shape recognition, symbolic geometry, or visual logic puzzles. In contrast, STARE introduces programmatically generated puzzles—2D/3D transformations, cube-net folding, and tangram assembly—that isolate a model's capacity to benefit from *explicit* visual simulations, and further extends to perspective-taking and temporal reasoning tasks that mirror real-world scenarios.

Table 6 compares STARE with other spatial reasoning datasets: RoboSpatial (Song et al., 2025) provides large-scale real 2D/3D images with annotated spatial relations; MSR3D (MSQA) (Linghu et al., 2024) and SQA3D (Ma et al., 2022) support situated QA in 3D scenes but focus on single-step queries; the Visual Spatial Reasoning benchmark (Liu et al., 2023) probes basic positional relations; SPARE3D (Han et al., 2020) presents synthetic 2D to 3D consistency puzzles; and DetermiNet (Lee et al., 2023) emphasizes logical spatial tasks without multi-step simulation. As shown, STARE is the only benchmark that offers a **diverse suite of multi-step visual simulation tasks** across both **2D and 3D domains**, uniquely combining **procedural puzzles**, **geometric transformations**, and **realistic inference** (perspective and temporal reasoning). Its synthetic design allows fine-grained

| Dataset | VSim | 2D/3D | Synth/Real | Multi-step | Train/Eval | Size | Focus |
|---|---|---|---|---|---|---|---|
| **STARE** | ✔ | 2D & 3D | Both | ✔ | Eval | ∼4K | Multi-step spatial simulations |
| VSI-Bench | ✘ | 3D (video) | Real | ✘ | Eval | 5K | Spatial memory & layout recall from egocentric videos |
| KiVA | ✘ | 2D | Synthetic | ✔ | Eval | 4.3K | Visual analogical reasoning inspired by child cognition |
| TurtleBench | ✘ | 2D | Synthetic | ✔ | Eval | 260 | Reproduce geometric programs via turtle graphics |
| SPARE3D | ✘ | 2D & 3D | Synthetic | ✔ | Both | 220K+ | Spatial reasoning with 3-view CAD drawings |
| VSR | ✘ | 2D | Real | ✘ | Eval | 10K | Spatial relation understanding in caption-image pairs |
| DetermiNet | ✘ | 2D | Synthetic | ✘ | Both | 250K | Referring expression understanding with quantifiers and determiners |
| Forgotten Polygons | ✘ | 2D | Synthetic | ✘ | Eval | 2K | Shape identification and counting under visual ambiguity |
| GeoGramBench | ✘ | 2D | Synthetic | ✔ | Eval | 500 | Symbolic geometric reasoning from procedural programs |
| VisuLogic | ✘ | 2D | Synthetic | ✘ | Eval | 1K | General visual logic across diverse reasoning types |

**Table 6:** Comparison of existing visual/spatial reasoning datasets versus STARE.

control over step difficulty and granularity, enabling analyses of visual reasoning beyond what existing datasets support.

# E   ADDITIONAL EXPERIMENTAL RESULTS

**Fine-grained Reasoning Evaluation** Because each synthetic task in STARE includes ground-truth metadata for every intermediate simulation step, we can scrutinize a model's entire reasoning chain—something impossible on benchmarks that provide only final answers.

We have conducted a deeper error analysis of GPT-4o and Claude-3.5 predictions on all synthetic tasks.

- We first examined representative case studies (Appendix G.5) and identified four recurring categories of reasoning failure:

| | Error Category | Description | Representative Example |
|---|---|---|---|
| **A** | Misperception | The model misreads shapes, color, adjacency, or face layout. | Misidentifies cube-net face positions (Figure 25) |
| **B** | Flawed Spatial Simulation | The model forms an incorrect mental model of how shapes transform. | Claims rotated hexagon is unchanged after 30° rotation (Figure 23-right) |
| **C** | Heuristic Over-Use | The model falls back on shallow heuristics (e.g., area counting). | Sums tangram piece areas and misjudges solvability (Figure 26-left) |
| **D** | Logic Inconsistency | The reasoning process contradicts itself or the final answer. | Correctly identifies two valid answers but chooses the wrong one arbitrarily (Figure 24-right) |

**Table 7:** Representative error categories (A–D) observed in model predictions, with descriptions and examples.

- We implemented an automated LLM-based judgment pipeline. Given a model's chain-of-thought, its prediction, and full ground-truth metadata (e.g., initial shape, transformation sequence, correct answer, and options), the judge model categorizes each incorrect prediction into one of the four failure types above. Here, we use o3 as the judge model.

The table below summarizes the distribution of error types across 100 randomly sampled incorrect predictions per task, per model:

| Model | Task | A | B | C | D |
|---|---|---|---|---|---|
| GPT-4o | 2D | 75.8% | 21.2% | 0.0% | 3.0% |
| | 3D | 31.3% | 67.7% | 0.0% | 1.0% |
| | Cube Net | 12.7% | 87.3% | 0.0% | 0.0% |
| | Tangram | 68.1% | 12.1% | 6.6% | 13.2% |
| Claude-3.5 Sonnet | 2D Transform | 60.0% | 35.8% | 0.8% | 3.3% |
| | 3D | 34.0% | 64.1% | 0.0% | 2.0% |
| | Cube Net | 22.2% | 74.7% | 3.0% | 0.0% |
| | Tangram | 65.9% | 17.1% | 9.8% | 7.3% |

**Table 8:** Distribution of error categories (A–D) across tasks for GPT-4o and Claude-3.5 Sonnet.

Different tasks show distinct failure modes: 2D and tangram errors stem from misperception, 3D and cube nets from simulation gaps, with occasional heuristic over-use and logic inconsistency in chain-of-thought.

- We use an LLM-as-judge with a structured rubric to score how well model reasoning traces align with ground-truth metadata (e.g., shape interpretation, transformation accuracy). We then sorted model responses into quartiles based on alignment score and evaluated accuracy within each group.

| Model | Task | Q1 (alignment) | Q2 (alignment) | Q3 (alignment) | Q4 (alignment) |
|---|---|---|---|---|---|
| GPT-4o | 2D | 61.1% (0.664) | 100.0% (0.917) | 100.0% (0.950) | 100.0% (0.952) |
| | 3D | 33.3% (0.325) | 100.0% (0.875) | 100.0% (0.927) | 100.0% (0.973) |
| | Cube Net | 0.0% (0.004) | 0.0% (0.096) | 0.0% (0.118) | 100.0% (0.606) |
| | Tangram | 0.0% (0.000) | 50.0% (0.228) | 100.0% (0.819) | 100.0% (0.998) |
| Claude-3.5 Sonnet | 2D | 0.0% (0.214) | 83.3% (0.768) | 100.0% (0.931) | 100.0% (0.959) |
| | 3D | 0.0% (0.168) | 90.9% (0.727) | 100.0% (0.909) | 100.0% (0.981) |
| | Cube Net | 0.0% (0.041) | 0.0% (0.100) | 0.0% (0.150) | 100.0% (0.714) |
| | Tangram | 0.0% (0.000) | 42.9% (0.161) | 100.0% (0.725) | 100.0% (0.991) |

**Table 9:** Task accuracy by alignment quartile for GPT-4o and Claude-3.5 Sonnet.

This analysis reveals a strong correlation between reasoning quality and final task accuracy. The prompt we used to scoring model reasoning is shown below.

---

**Reasoning Error Analysis Rubrics**

**You are an expert spatial-reasoning judge.**
Given the information blocks below, score the model's reasoning using the rubric.

**Question:** {question_text}

**Ground Truth**

- Start shape: {initial_shape_desc}
- {transformation_step_desc} {transformation_outcome_desc}
- {choice_shape_desc}
- Correct answer: {gt_answer}

**Model Response:** {model_response}
**Model Final Answer:** {model_pred}

**Rubric — assign 0 or 1 to each item**

1. Shape interpretation
2. Transformation comprehension
3. Spatial simulation accuracy

---

> 4. Answer justification (choice evaluation)
>
> 5. Logical consistency
>
> **Return JSON**
>
> ```
> {
>   "shape_interpretation": {"score": 0|1, "details": ""},
>   "transformation_comprehension": {"score": 0|1, "details": ""},
>   "spatial_simulation_accuracy": {"score": 0|1, "details": ""},
>   "answer_justification": {"score": 0|1, "details": ""},
>   "logical_consistency": {"score": 0|1, "details": ""},
>   "primary_errors": [],
>   "overall_assessment": ""
> }
> ```

**Number of visual simulations vs. Performance,** We investigated this question briefly in Table 5. The key insight is that the benefit of visual simulations depends on task complexity and where in the sequence the decisive information appears. We also conducted a stricter ablation that incrementally reveals 0 to 3 simulation frames. The takeaway is straightforward: visual simulations help only when the model can integrate them; otherwise they add noise.

Because examples have different total chain lengths, the same column mixes complete simulations for short chains (e.g., all 2–step tasks are already complete at # simulations = 1) and partial simulations for longer ones (e.g., only one of two frames visible at # simulations = 1 for 3–step tasks). If a model could perfectly integrate every extra frame, scores would rise monotonically. However, the results above suggest otherwise, especially for more complex tasks. These results again highlight a core limitation: models lack the capacity to mentally simulate and reason over visual sequences, a skill that humans perform reliably.

| Model | Task | # of simulations = 0 | 1 | 2 | 3 |
|-------|------|------|------|------|------|
| **GPT-4o** | 2D | 71.2 | 78.0 | **85.6** | — |
| | 3D | 65.5 | 67.2 | **69.6** | — |
| | Cube Net | 50.3 | **50.5** | 49.2 | 49.1 |
| | Tangram | 52.5 | 51.7 | 46.6 | **54.7** |
| **Claude-3.5** | 2D | 65.9 | 67.7 | **75.3** | — |
| | 3D | 51.5 | **60.8** | 53.9 | — |
| | Cube Net | **52.3** | 51.3 | 51.7 | 50.0 |
| | Tangram | 59.0 | 60.1 | 62.2 | **69.0** |

**Table 10:** Ablation with 0–3 intermediate simulations. Bold indicates best performance per row.

**Reasoning Efforts vs. Performance.** In addition to the results of o3 reported in Table 1, we further report o3 performance across different reasoning efforts (low, auto, high), as it is specifically optimized to "think with images" and perform extended reasoning.

| Task (w/o vs. w/ VSim) | o3-Low | o3-Auto | o3-High | Human |
|-------|------|------|------|------|
| 2D | 88.1 / **92.3** | 87.5 / 89.3 | **89.7** / 89.5 | 96.8 / 98.6 |
| 3D | 73.7 / 75.3 | **75.2** / **78.4** | 73.9 / 76.5 | 94.6 / 97.0 |
| Cube Net | 65.3 / 71.7 | **68.4** / **72.5** | 66.3 / 71.1 | 98.3 / 99.0 |
| Tangram | **68.6** / 76.5 | **68.6** / 82.1 | 66.4 / **82.8** | 91.5 / 95.8 |
| Video-temporal | **55.8** | 51.4 | 54.3 | 99.0 |
| Perspective | 43.6 | 42.8 | **44.0** | 98.1 |
| **Overall** | **67.5** | **67.5** | 68.8 | 97.1 |

**Table 11:** Performance of o3 model across different reasoning efforts (Low, Auto, High) compared to human annotators on STARE tasks. For tasks with visual simulation (VSim), accuracy is reported as "w/o / w/ VSim".

While o3 outperforms earlier models such as o1 and GPT-4o, its performance on STARE remains significantly below human-level. Notably, o3 leverages visual simulations more effectively than prior models. However, humans show only small performance gaps between conditions with and

without visual simulation, reflecting their ability to mentally simulate transformations. In contrast, o3 exhibits substantial drops without external visuals, highlighting a key limitation: the inability to perform internal, structured visual simulation–a core component of human spatial reasoning.

In addition, while o3 has reported to benefit from extended reasoning, simply increasing the reasoning efforts from low to high does not guarantee better spatial reasoning.

**Additional Results on Human Evaluation.** In Table 1, we reported the average human performance and average response time for each task across 5 participants. In Table 12, wer further report the standard deviation. Mean accuracy across five annotators is 97.07% with a standard deviation of 0.47, indicating that the questions in STARE are well-defined.

| Task | Accuracy (%) | Time (s) |
|---|---|---|
| 2D | $96.75 \pm 1.30$ | $14.23 \pm 1.99$ |
| 2D + VSim | $98.56 \pm 1.10$ | $10.95 \pm 1.51$ |
| 3D | $94.61 \pm 1.44$ | $17.06 \pm 5.08$ |
| 3D + VSim | $96.98 \pm 0.80$ | $12.53 \pm 0.65$ |
| Cube Net | $98.29 \pm 1.65$ | $13.67 \pm 3.66$ |
| Cube Net + VSim | $98.86 \pm 1.46$ | $5.16 \pm 0.49$ |
| Tangram | $91.53 \pm 2.56$ | $27.98 \pm 5.85$ |
| Tangram + VSim | $95.78 \pm 1.07$ | $10.08 \pm 4.20$ |
| Temporal | $99.03 \pm 0.98$ | $16.19 \pm 3.96$ |
| Perspective | $98.10 \pm 0.17$ | $18.04 \pm 4.46$ |
| **Overall** | $\mathbf{97.07 \pm 0.47}$ | – |

**Table 12:** Task accuracy and response time across STARE benchmark tasks, with and without visual simulation (VSim).

**Correlation Analysis between Synthetic tasks and Real tasks.** In Section 3.2, we briefly discussed the correlation between averaged model performance on synthetic tasks (including 2D transformation, 3D transformation, cube net folding and tangram puzzle) and that on real-world tasks (including temporal frame reasoning and perspective reasoning). Figure 7 shows the averaged model performance on synthetic and real-world tasks across 11 models and the fitted line with correlation coefficient $r \approx 0.88$.

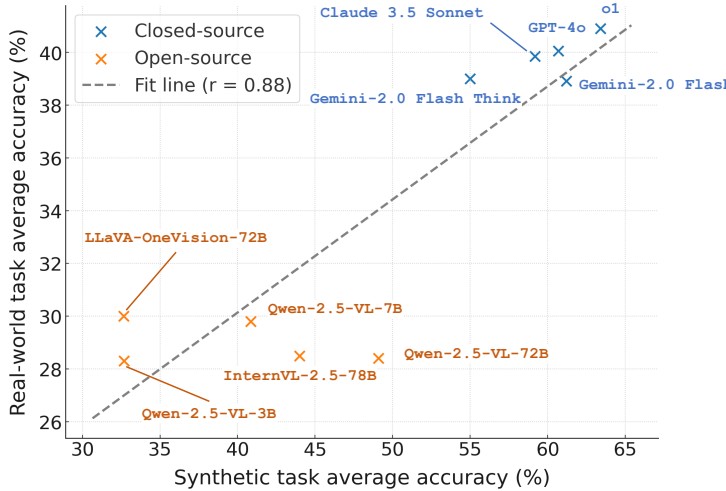

**Figure 7:** Correlation between model performance on synthetic tasks and that on real-world tasks.

Note that for open-source models, the real-world task performance is close to random guessing (29%). Removing the open-source models, the correlation coefficient decreased to $r \approx 0.58$, still showing a weak but positive correlation between synthetic task performance and real-world task performance.

**Visual Simulation for Real-World Tasks.** While visual simulations improve performance on synthetic spatial tasks, we investigate whether this benefit extends to real-world tasks. We create three

evaluation settings for perspective and temporal reasoning (see Figures 8 and 9 for examples): (1) **w/o VSim**: the standard setting; (2) **w/ partial VSim**: 1 intermediate frame for perspective (sampled along the camera trajectory) and 1 extra context frame before and after the missing frame for temporal; (3) **w/ all VSim**: 3 intermediate frames for perspective and 2 extra context frames on each side for temporal.

As shown in Table 13, the two tasks exhibit strikingly different patterns. For **perspective reasoning**, accuracy increases sharply as more intermediate views are provided: GPT-4o improves from 38.7% to 76.7%, and o3 from 42.8% to 84.8%. This confirms that perspective tasks support a meaningful with/without simulation framing, and models benefit substantially from seeing the viewpoint transition. For **temporal reasoning**, however, adding partial or full context yields only modest or no gains (GPT-4o: 39.0% → 37.5%; o3: 51.4% → 50.3%), suggesting that current models struggle to infer coherent motion trajectories even when more evidence is available. Notably, humans already reach 99.0% (temporal) and 98.1% (perspective) without any extra simulated frames, highlighting the large gap between human and model spatial reasoning.

| Model | Task | w/o VSim | Partial VSim | All VSim |
|---|---|---|---|---|
| GPT-4o | Temporal | **39.0** | 37.4 | 37.5 |
| GPT-4o | Perspective | 38.7 | 55.0 | **76.7** |
| o3 | Temporal | **51.4** | 50.7 | 50.3 |
| o3 | Perspective | 42.8 | 55.2 | **84.8** |

**Table 13:** Effect of visual simulations on real-world tasks. Perspective reasoning benefits strongly from intermediate views, while temporal reasoning shows negligible improvement.

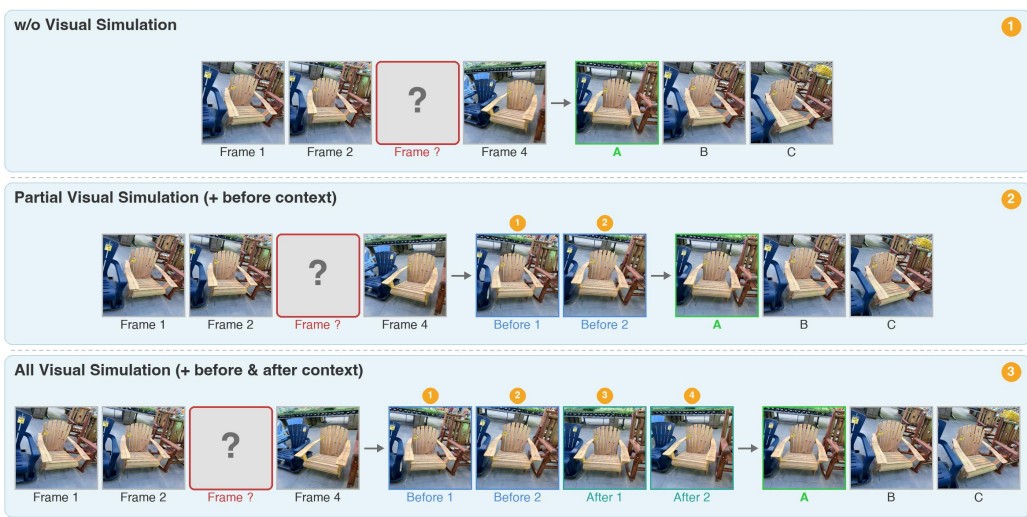

**Figure 8:** Example from the temporal frame reasoning task with visual simulation. **Top:** without visual simulation, the model sees 4 frames with one missing and 3 candidate options. **Middle:** partial visual simulation adds 2 context frames before the missing frame. **Bottom:** full visual simulation adds context frames both before and after. Green border indicates the correct answer.

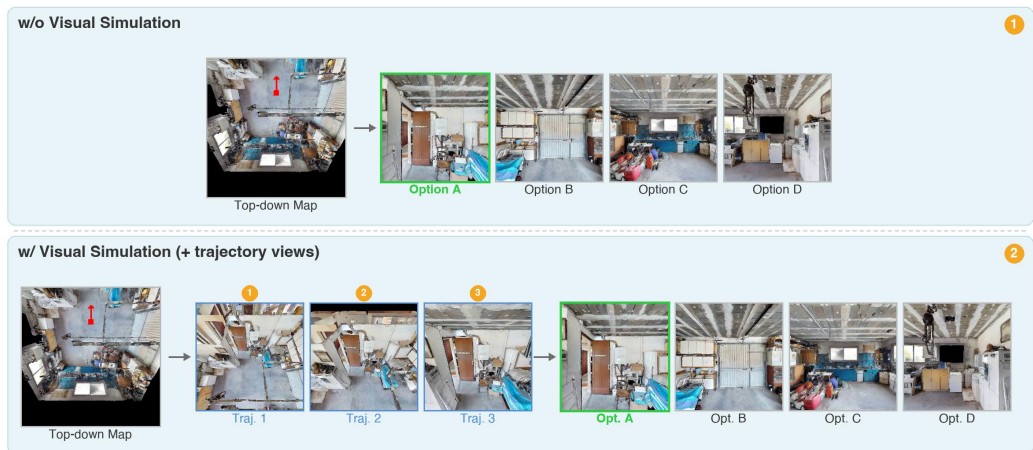

**Figure 9:** Example from the perspective reasoning task with visual simulation. **Top:** without visual simulation, the model sees a top-down map with the agent's position/orientation and selects a first-person view from 4 options. **Bottom:** with visual simulation, 3 intermediate trajectory views along the agent's path are provided as additional context. Green border indicates the correct answer.

**Downstream Application: Robot Planning.** To explore the practical value of visual simulation beyond benchmark evaluation, we conduct a pilot study on the ALFRED dataset (Shridhar et al., 2020), which involves household robot planning tasks. The model sees a current state image, a goal state image, and several candidate plans (sequences of high-level actions), and must choose which plan reaches the goal (see Figure 10 for examples). For each sample, we also have intermediate state images along the execution of the correct plan (average 5.73 steps). We evaluate models with incremental visual simulations (25%, 50%, 75%, and 100% of the intermediate frames) on 200 multiple-choice samples. As shown in Table 14, task planning accuracy improves substantially as more simulation frames are provided. GPT-4o improves from 39.5% without simulation to 79.5% with all frames, and o3 improves from 73.5% to 93.0%. This strong benefit from visual simulations in a procedural, embodied setting complements our findings on synthetic tasks, though we note that verbal reasoning alone can partially solve these tasks (e.g., matching goal descriptions to plan text), which is why we keep STARE focused on tightly controlled, non-verbal spatial reasoning.

| Model | w/o VSim | 25% | 50% | 75% | All VSim |
|---|---|---|---|---|---|
| GPT-4o | 39.5 | 60.5 | 66.5 | 68.0 | **79.5** |
| o3 | 73.5 | 71.0 | 78.5 | 87.0 | **93.0** |

**Table 14:** Pilot study on ALFRED robot planning tasks (200 samples). Accuracy improves as more visual simulation frames are provided.

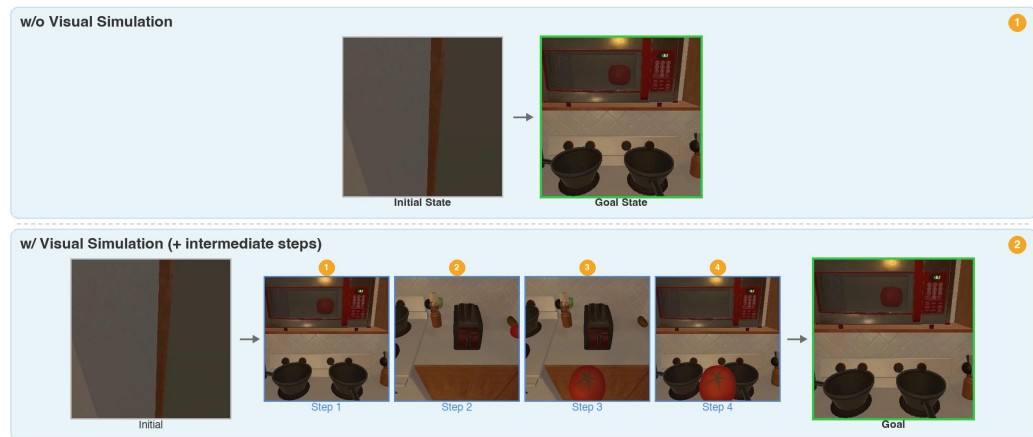

**Figure 10:** Examples from the ALFRED robot planning task. **Top:** without visual simulation, the model sees only the initial state and goal state. **Bottom:** with visual simulation, intermediate state images showing the progression of the correct plan are provided. The model must select the correct plan from multiple candidates.

**Mechanical Diagram Interpretation.** Motivated by the potential applicability of spatial reasoning to engineering domains, we conduct a pilot study on mechanical diagram interpretation. We curate 28 questions involving gear trains and pulley systems (see Figure 11 for examples), requiring models to reason about rotational direction, mechanical advantage, and motion transmission.

As shown in Table 15, o3 (67.86%) significantly outperforms GPT-4o (39.29%), consistent with its stronger spatial reasoning observed on synthetic tasks. The large gap suggests that mechanical diagram interpretation requires the same kind of structured spatial reasoning that o3 excels at. This pilot demonstrates that our benchmark's spatial reasoning constructs extend to practical engineering domains, and we leave a comprehensive evaluation to future work.

| Model | Accuracy (%) |
|-------|-------------|
| GPT-4o | 39.29 |
| o3 | 67.86 |

**Table 15:** Pilot study on mechanical diagram interpretation (28 questions on gear/pulley systems).

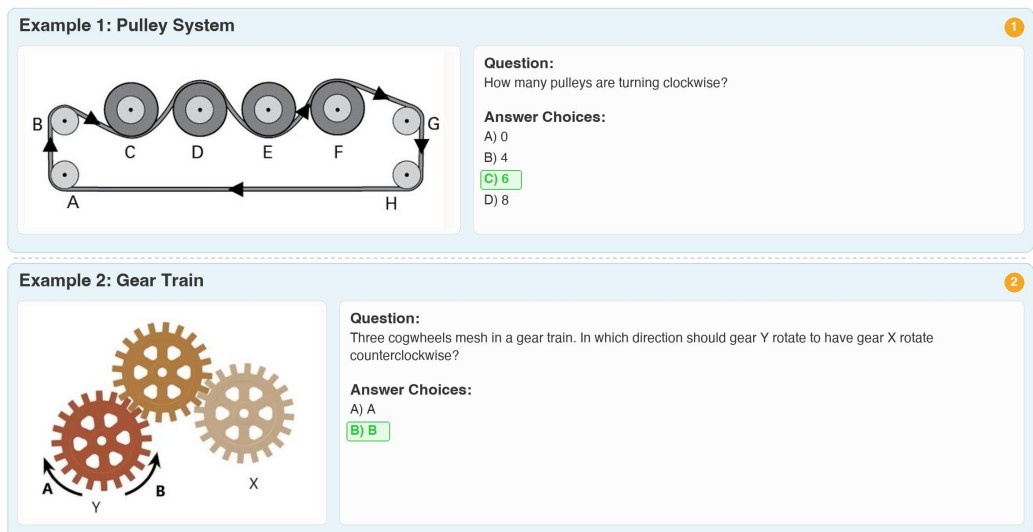

**Figure 11:** Examples from the mechanical diagram interpretation pilot. **Top:** a pulley system question requiring counting and spatial reasoning. **Bottom:** a gear train question requiring models to reason about rotational direction propagation.

**Open-Ended Answering.** To verify that the multiple-choice (MC) format does not artificially inflate model performance, we evaluate GPT-4o on cube net folding and tangram puzzle tasks using free-form, open-ended answering where the model must generate the answer without seeing candidate options.

As shown in Table 16, MC and open-ended scores are comparable: cube net folding achieves 50.3% with MC and 50.9% with open-ended answering, while tangram puzzles show 52.5% with MC and 48.3% with open-ended answering. The small differences ($<$5%) in both directions confirm that the MC format does not systematically inflate or deflate performance, validating our evaluation protocol.

| Task | MC | Open-Ended |
|------|------|------------|
| Cube Net | 50.3 | 50.9 |
| Tangram | 52.5 | 48.3 |

**Table 16:** Comparison of multiple-choice (MC) vs. open-ended answering for GPT-4o. Scores are comparable, confirming the MC format does not inflate performance.

**Interleaved Image Generation.** To investigate whether models capable of interleaved text and image generation can leverage self-generated visual intermediates for spatial reasoning, we evaluate Bagel-7B, an open-source model with interleaved generation capabilities. We modified the decoding logic to force interleaved generation: after the model produces a text "thought," we switch to image generation mode, then back to text generation with all previous context (both image and text). Since the model often does not strictly follow the requested format, we use o3 as a judge model to extract answer choices.

As shown in Table 17, interleaved generation helps on some tasks (3D: 24.0% $\rightarrow$ 48.0%; cube nets: 50.8% $\rightarrow$ 76.5%; tangram: 48.5% $\rightarrow$ 64.8%), but performance remains low overall and even drops on temporal (22.5% $\rightarrow$ 16.0%) and perspective (6.0% $\rightarrow$ 4.0%). These results show that simply allowing interleaved image–text generation does not automatically produce meaningful, task-aligned visual simulations. The bottleneck lies in performing the correct spatial transformations and using visual evidence to reach the right answer.

| Setting | 2D | 3D | Cube Nets | Tangram | Temporal | Perspective |
|---------|------|------|-----------|---------|----------|-------------|
| Text-only | 44.9 | 24.0 | 50.8 | 48.5 | 22.5 | 6.0 |
| Interleaved Gen. | 46.0 | 48.0 | 76.5 | 64.8 | 16.0 | 4.0 |

**Table 17:** Bagel-7B performance (w/o VSim) with text-only generation vs. interleaved image generation. Interleaved generation helps on some synthetic tasks but hurts on real-world tasks.

# F  DATA CURATION DETAILS

Figure 12 presents the overall composition of STARE. Table 18 details the number of instances for each task in STARE, further broken down by whether the input contains an explicit intermediate visual simulations.

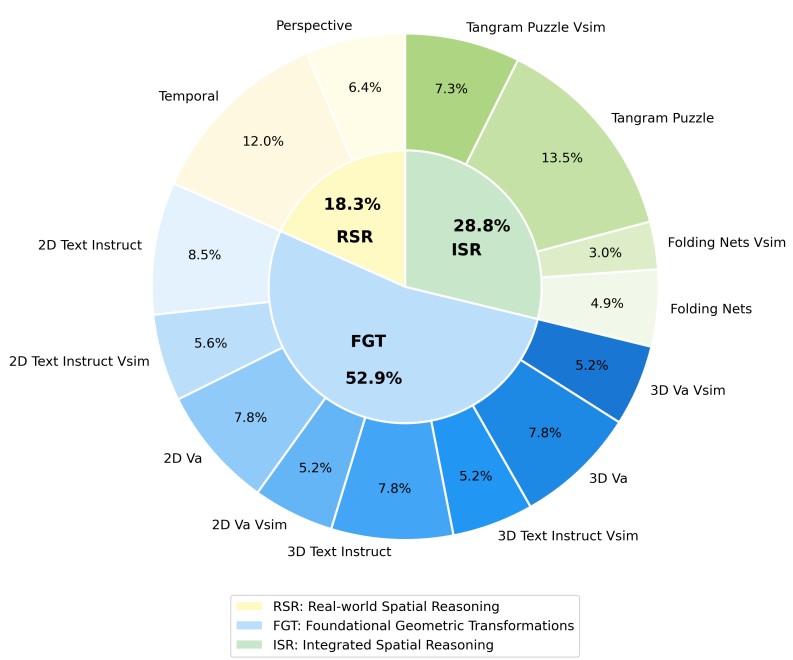

**Figure 12: Data Statistics of STARE.**

| Task category | Without visual simulation | With visual simulation | Total |
|---|---:|---:|---:|
| *Foundational Geometric Transformations* | | | |
| 2D transformations | 639 | 423 | 1,062 |
| 3D transformations | 612 | 408 | 1,020 |
| *Integrated Spatial Reasoning* | | | |
| Cube net folding | 193 | 120 | 313 |
| Tangram puzzle | 532 | 289 | 821 |
| *Real-world Spatial Reasoning* | | | |
| Perspective reasoning | 250 | – | 250 |
| Temporal frame reasoning | 471 | – | 471 |
| **Total** | 2,697 | 1,240 | 3,937 |

**Table 18:** Dataset statistics grouped by task category and by the presence of full intermediate visual simulation.

Below, we summarize the design space of data curation for synthetic tasks, including (1) 2D Transformations (§F.1); (2) 3D Transformations (§F.2); (3) Cube Net Folding (§F.3); and (4) Tangram Puzzles (§F.4);

## F.1 2D TRANSFORMATIONS

**Shape generation.** Shapes are selected from a fixed set and assigned properties as follows:

- **Types:** Circle, Square, Rectangle, Triangle, Ellipse, Hexagon, Pentagon.
- **Colors:** Face color is a random RGB tuple ($r, g, b \in [0, 1]$); edge color is fixed (black).
- **Center & Size:** All shapes are centered at $(0, 0)$. For circles, squares, triangles, hexagons, and pentagons, size is a scalar drawn from $[30, 35]$; for rectangles and ellipses, size is a tuple (width in $[30, 35]$, height in $[20, 25]$).

**Transformations.** A sequence of randomly sampled operations is applied to the shapes:

- **Rotate:**
    - *Squares:* $\pm 30°$, $\pm 60°$ (avoiding $90°$).

- *Hexagons:* $\pm 30°$, $\pm 90°$.
- *Others:* $\pm 30°$, $\pm 60°$, or $\pm 90°$.

Rotation is applied w.r.t the shape's center.

- **Flip:** Horizontal (about $y = 0$) or vertical (about $x = 0$); not applied when the shape is centered at $(0, 0)$ for symmetric shapes such as square, circle and etc.

- **Translate:** $(dx, dy)$ with $dx, dy \in \{-30, -10, 0, 10, 30\}$ with constraints to ensure a nonzero translation.

- **Scale:** Factors chosen from $\{0.5, 2.0\}$, ensuring the resultant size is within roughly $[10, 40]$.

- **Shear:** Parameters $(\text{shear}_x, \text{shear}_y)$ are drawn from approximately $[-1, 1]$, with constraints to ensure a perceptible skew. Shear is excluded for 2D text instructed transformation tasks, as human participants find it hard to describe the degree of shear such that they can differentiate among the answer candidates.

**Number of Transformation Steps.** The final dataset contains instances with 1, 2, or 3 transformation steps.

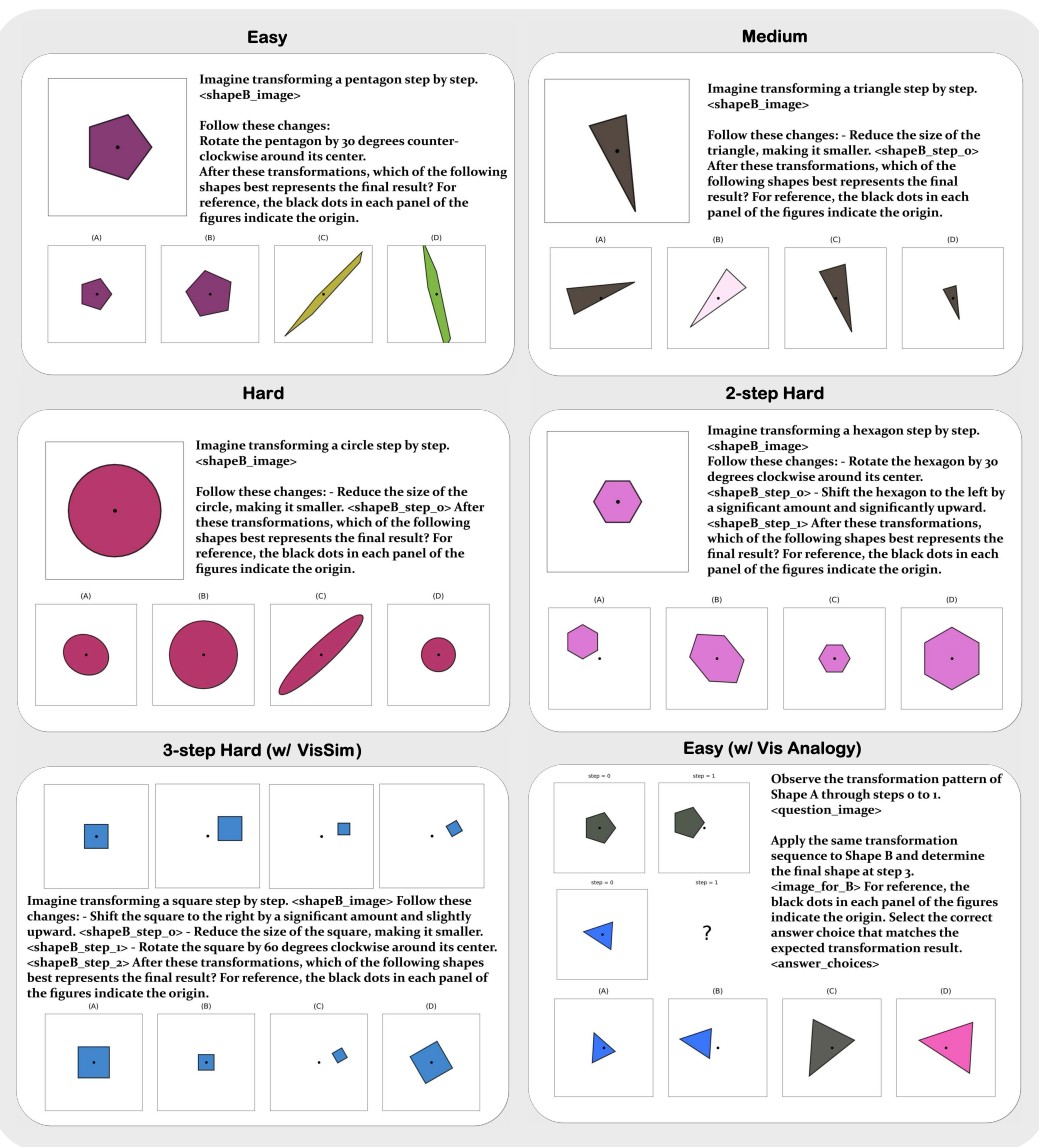

**Figure 13: Design space of 2D Transformations (1).**

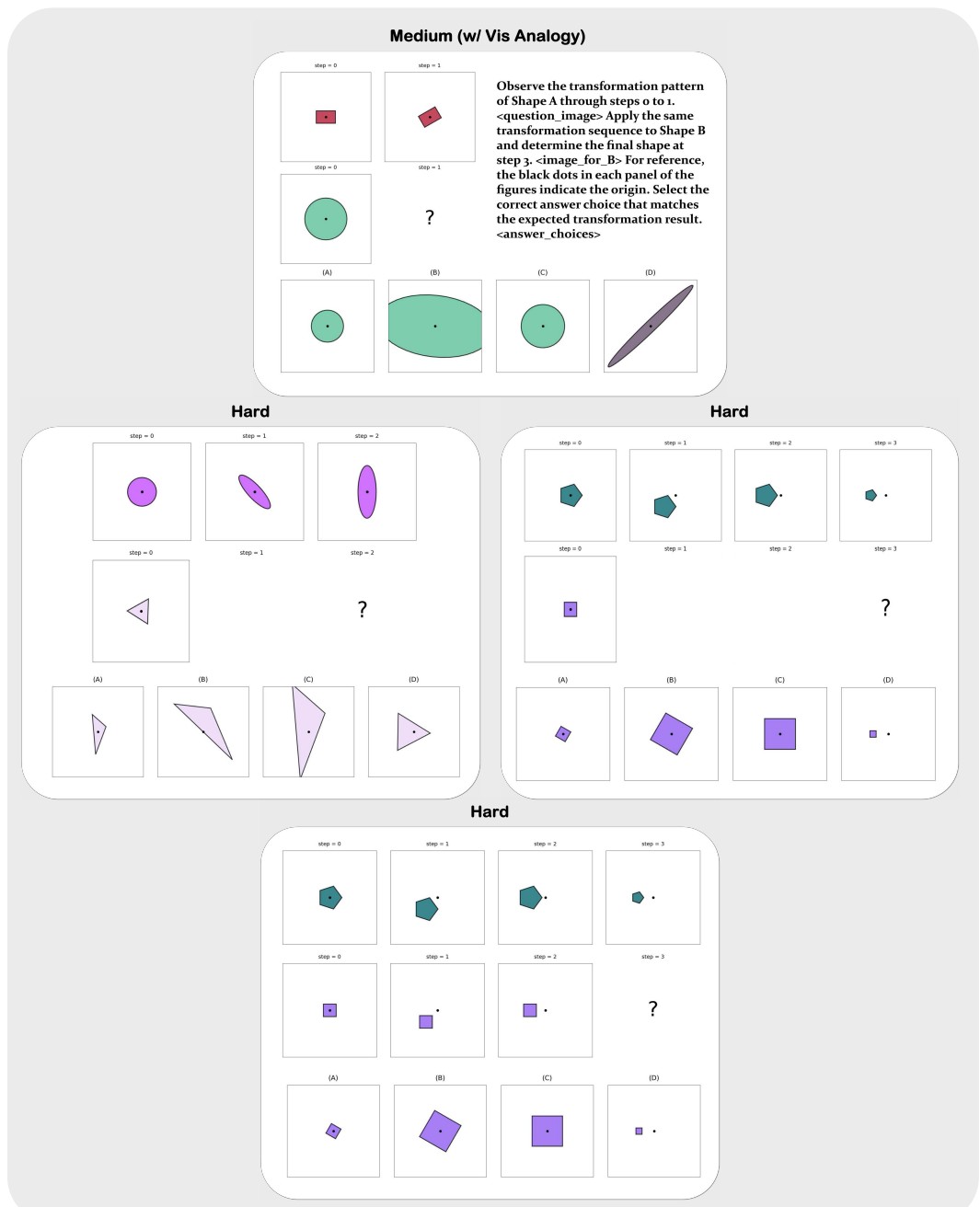

**Figure 14: Design space of 2D Transformations (2).**

## F.2 3D TRANSFORMATIONS

**Shape generation.** 3D objects are loaded from external blend files and instantiated with random properties defined in a JSON file. Their attributes include:

- **Types:** Various 3D models such as cube, sphere, cone, cylinder, torus, pyramid, etc.
- **Colors & Materials:** Colors are sampled from a predefined set, and materials are selected from external files.
- **Size & Location:** Objects are assigned a size scalar (from the JSON-specified values) and an initial 3D location (typically near the origin), with adjustments to ensure they remain above the ground plane.

**Transformations.** A sequence of randomly sampled operations is applied to the objects in 3D space:

- **Translate:**
  - *Axis selection:* Randomly choose one or more axes from *x*, *y*, and *z* (e.g., "x", "xy", "xz", "yz").
  - *Displacement:* Translations are applied with discrete displacements: along *x* and *y* by $\pm 2$ units and along *z* by $\pm 1$ unit, with constraints to keep the object above the ground ($z \geq 0$).

- **Rotate:**
  - *Axis:* A single rotation axis is chosen randomly from *x*, *y*, or *z*.
  - *Angle:* The rotation angle is drawn from a discrete set (typically $\pm 30°$, $\pm 60°$, or $\pm 90°$), with the range sometimes adjusted for specific shapes (e.g., cubes or pyramids).
  - Rotation is applied about the object's center.

- **Shear:**
  - *Plane:* The shear operation is applied along one of three directional pairs: $x_y$, $x_z$, or $y_z$.
  - *Factors:* Two shear factors are sampled uniformly from the interval $[0.2, 1.0]$, with an enforced minimum difference (approximately 0.4) to ensure a perceptible skew.

- **Scale:**
  - *Factor:* A uniform scaling factor is chosen from $0.5, 2.0$, either reducing or enlarging the object while keeping its final size within acceptable bounds.

- **Flip:**
  - *Direction:* The object is reflected along a principal axis—flipped horizontally (reflection across the *x*-axis) or vertically (reflection across the *y*-axis).

All transformation operations are applied sequentially, updating the object's 3D coordinates (including its bounding box and center) to reflect the cumulative effects.

**Number of Transformation Steps.** Instances are generated with transformation sequences comprising 1, 2, or 3 steps, where each step randomly selects one of the available operations. This multi-step approach enables a diverse design space of 3D transformations, as the operations can compound in various orders and combinations.

F.3 CUBE NET FOLDING

**Net Representation.** Cube nets are represented as collections of faces, where each face is defined by its vertices in 3D space. Additional attributes include:

- **Face Geometry:** Each face is a polygon (typically a quadrilateral) with vertex coordinates stored as NumPy arrays.
- **Connectivity:** A mapping of face connections identifies which faces share common edges, serving as potential hinges.
- **Visual Attributes:** Faces are rendered with colors (sampled from a colormap) and labeled with their keys for easy identification.

**Folding Operations.** The folding process simulates converting a 2D cube net into a 3D cube via a sequence of rotation operations:

- **Shared Edge Detection:** The algorithm locates the common edge between a candidate face and an already folded face. A tolerance is used to robustly identify two shared vertices.
- **Rotation Calculation:** Using the shared edge as a hinge, a rotation is computed with a fixed magnitude of $90°$ (i.e. $\pm \pi/2$ radians). The sign of the angle is chosen by comparing the candidate face's center (projected onto the hinge's perpendicular plane) with the desired direction toward the cube's center, which is derived from the base face.

- **Recursive Propagation:** The rotation is applied not only to the candidate face but also recursively to all connected faces that have not been folded yet, ensuring that the entire net adjusts consistently.

**Folding Sequence and Visualization.** The design space supports iterative, step-by-step folding, with each step comprising:

- **Candidate Selection:** Among the faces not yet folded, the algorithm picks one that is connected to an already folded face.
- **Folding Parameters:** It computes the rotation axis (the shared edge) and the appropriate 90° rotation (with correct sign) to fold the face into its 3D position.
- **Instruction Generation:** Each fold is described in natural language (e.g., "Fold face 2 upwards towards face 3") based on changes in the face's center relative to the cube's base.
- **3D Rendering:** After each step, the current state of the net is visualized using a 3D plot (with Poly3DCollection) and saved as an image.

**Perturbation and Validity.** To enrich the design space and introduce challenge:

- **Perturbations:** Selected folding steps can be intentionally altered by inverting the rotation angle or modifying the rotation axis. This simulates errors or variations, yielding nets that might fold incorrectly.
- **Validity Checks:** Functions are provided to verify that folded faces do not overlap, that shared edges are consistently maintained, and that face connections remain intact. These checks ensure that the final folded cube is geometrically valid.

**Dataset Generation and Perception Tasks.** Beyond simulating the folding process, the design space incorporates mechanisms to create annotated datasets:

- **Instructional Sequences:** Detailed, step-by-step folding instructions (with corresponding images) are generated, supporting tasks that require understanding the folding procedure.
- **Perception Variants:** Additional tasks query the observer's perception—such as verifying if a particular face has been folded or determining the connectivity between faces—using intermediate folding images.

**Randomness and Parameter Control.** Stochastic elements pervade the folding simulation:

- Random seeds govern the selection of candidate faces, the decision to perturb a folding step, and the choice of rotation adjustments.
- This randomness ensures that a diverse range of cube nets and folding sequences are produced, which is crucial for generating robust datasets and for studying perception and reasoning in 3D folding tasks.

### F.4  TANGRAM PUZZLE

**Segmentation.** The puzzle begins with an iterative segmentation algorithm that splits a full rectangular board into smaller pieces. The process is governed by a minimum piece size and a maximum number of pieces. At each segmentation step, the algorithm:

- Selects a splittable rectangle based on its area.
- Chooses a split direction (horizontal if the height is greater or vertical otherwise) and a split line ensuring both resulting pieces exceed the minimum size.
- Records each split as an action with details (original rectangle, split line, and direction) that form the basis for later textual instructions.

**Piece Generation & Attributes.** Each tangram piece is defined by its board coordinates (e.g., `(r0, r1, c0, c1)`) and derived properties such as area and dimensions. Additionally:

- **Colors:** Pieces are assigned unique, randomly generated colors.

- **Visualization:** Grid lines and labels are overlaid on each piece to indicate its boundaries and area, facilitating clear visualization during reassembly.

**Scrambling and Transformation.** Once segmented, pieces are scrambled to increase puzzle complexity. This involves applying a series of random transformation operations:

- **Rotation:** Each piece is rotated by a discrete angle chosen from $0°, 30°, 60°, 90°$.
- **Translation:** Pieces are repositioned into non-overlapping cells on a larger canvas.
- **Flip:** In some reassembly variants, horizontal or vertical flips are applied to further randomize the piece orientations.

## G  EXPERIMENTAL DETAILS

### G.1  MODELS AND SETTINGS

To expedite response generation, we use the vLLM (Kwon et al., 2023) library, an open-source tool for fast LLM inference and serving. For all other cases, we load models directly using the Transformers (Wolf et al., 2020) library. All model sources are official and listed in Table 19. When evaluating different models, we use default hyperparameter values unless otherwise specified, with detailed parameter settings provided in Table 19. For all models, we explicitly prompt it with `Think step-by-step, and then put your final answer in \"\\boxed{}\".` to encourage chain-of-thought reasoning and for easier answer parsing.

| Model | Parameter Setting | Source | URL |
|---|---|---|---|
| GPT-4o | temperature = 0.0 | chatgpt-4o-latest | `https://platform.openai.com` |
| Claude 3.5 Sonnet | temperature = 0.0 | claude-3-5-sonnet | `https://www.anthropic.com/` |
| Gemini 2.0 Flash | temperature = 0.0 | gemini-2.0-flash-exp | `https://ai.google.dev/` |
| Gemini 2.0 Flash Thinking | temperature = 0.0 | gemini-2.0-flash-thinking-exp-1219 | `https://ai.google.dev/` |
| OpenAI o1 | temperature = 0.0 | o1-2024-12-17 | `https://platform.openai.com` |
| OpenAI o3 | reasoning-efforts=auto | o3-2025-04-16 | `https://platform.openai.com` |
| Qwen2.5-VL-3B | do_sample=True, temperature = 0.7 | local checkpoint | `https://huggingface.co/Qwen/Qwen2.5-VL-3B-Instruct` |
| Qwen2.5-VL-7B | do_sample=True, temperature = 0.7 | local checkpoint | `https://huggingface.co/Qwen/Qwen2.5-VL-7B-Instruct` |
| Qwen2.5-VL-72B | do_sample=True, temperature = 0.7 | local checkpoint | `https://huggingface.co/Qwen/Qwen2.5-VL-72B-Instruct` |
| LLaVA-Onevision-72B | do_sample=True, temperature = 0.7 | local checkpoint | `https://huggingface.co/llava-hf/llava-onevision-qwen2-72b-ov-hf` |
| InternVL2.5-78B | do_sample=True, temperature = 0.7 | local checkpoint | `https://huggingface.co/OpenGVLab/InternVL2_5-78B` |

**Table 19:** The sources of models used in the experiments and the hyperparameters configuration.

### G.2  VISUALIZATION OF EVALUATION SETTINGS

Figures 15–16 provide full visualizations of evaluation settings illustrated in Figure 3. In addition, we show an example of how real-world spatial reasoning task – temporal frame reasoning is evaluated without visual simulation in Figure 17.

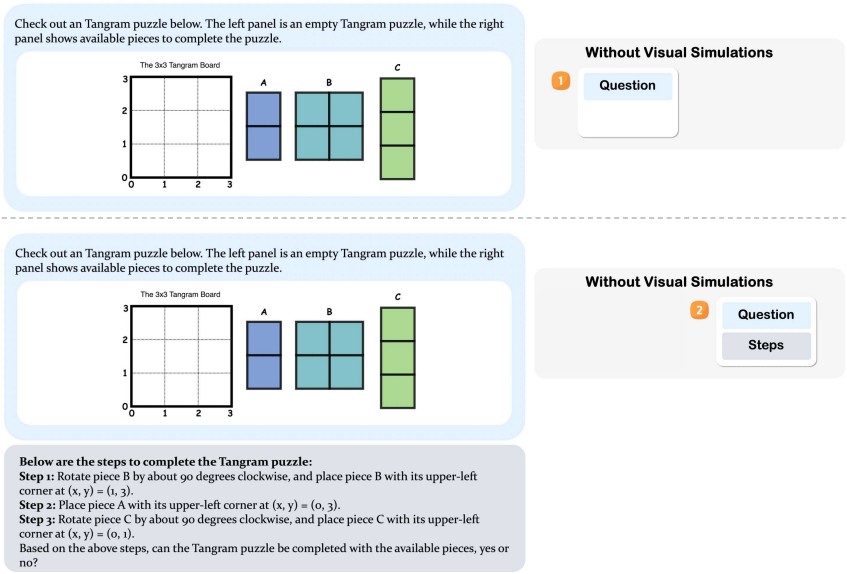

**Figure 15:** Examples of Tangram Puzzle under "without Visual Simulations" Evaluation Setting (top: question-only, bottom: question+assembly steps).

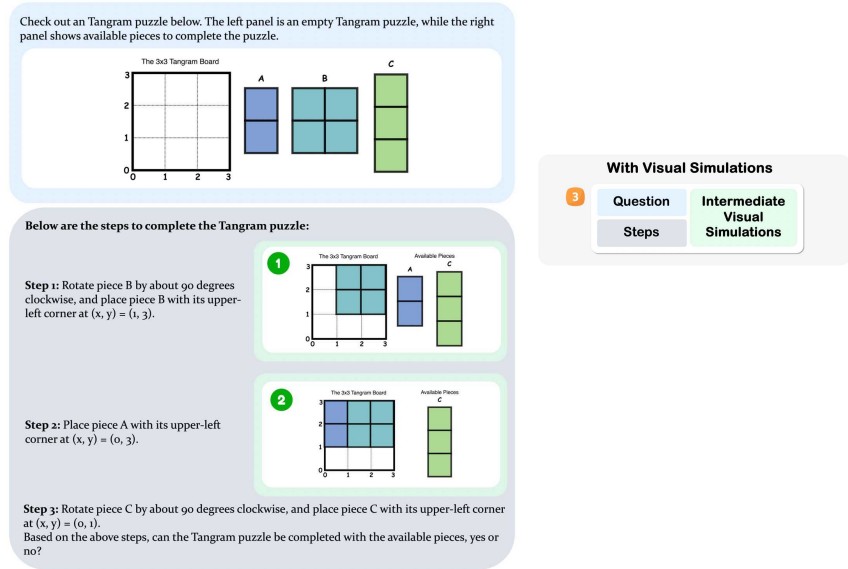

**Figure 16:** Example of Tangram Puzzle under "with Visual Simulations" Evaluation Setting.

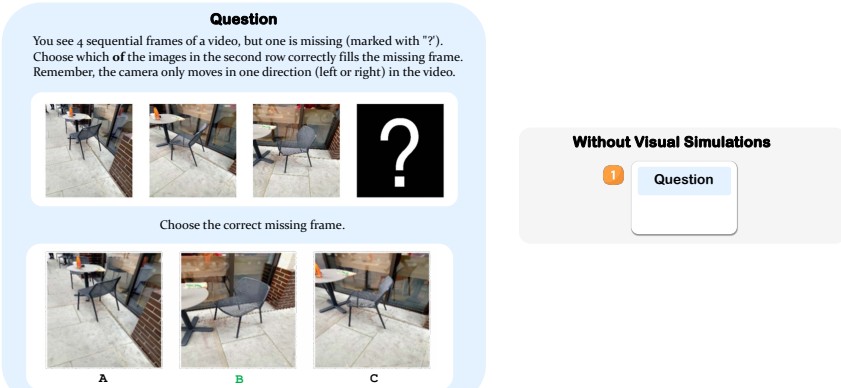

**Figure 17:** Examples of Temporal Frame Reasoning under "without Visual Simulations" Evaluation Setting.

### G.3 VISUALIZATIONS OF PERCEPTION PROBING QUESTIONS

In Figure 5, Claude demonstrates a perceptual error: while it correctly identifies all face colors, it incorrectly perceives face 6 to be positioned beneath face 4, when it is actually located beneath face 5. Such errors prompt an important question regarding task performance: for challenging tasks like cube net folding, to what extent does the low performance stem from perceptual inaccuracies rather than deficiencies in simulation capabilities or an inability to correctly interpret simulation outcomes? We design probing questions to evaluate model performance 2D and 3D perception on cube nets (Figure 18), which reveals that model fail substantially on 3D perception (Table 2), which may be the main bottleneck in understanding intermediate visualizations in cube net folding (Table 1).

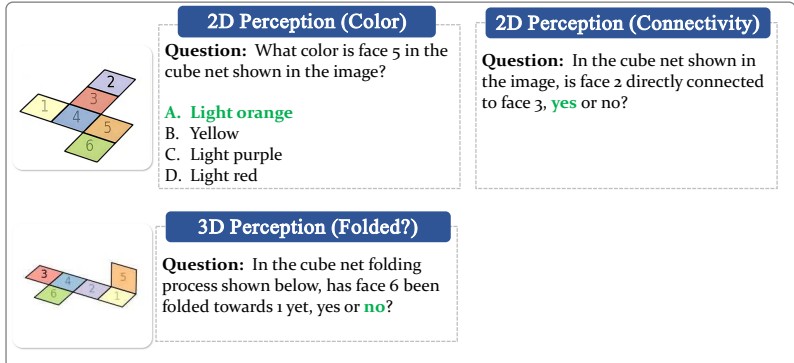

**Figure 18:** Exemplary questions on cube nets to probe model performance on 2D and 3D perception.

### G.4 VISUALIZATIONS OF STARE TASK IN DIFFERENT REPRESENTATIONS

Figures 19–22 provide concrete examples of the input modalities evaluated in STARE. For every task family we visualize the image-only variant (the original format in STARE), the text-only variant (compact symbolic description that can be consumed without vision), and—where applicable—the combined image+text variant that concatenates the two.

- **2D and 3D transformations**. In the text-only panels, each object is serialized as `<shape>, <color>, <x,y>, <size>`, with attributes separated by commas (e.g., "`square, red, (3, 4), 2`"). The image+text panels place the same textual description beneath the image, so that language and vision can be attended to jointly.
- **Cube-net folding**. We flatten the cube into a 2D grid and enumerate its faces from 1 to 6. The text-only representation thus becomes a short digit string (e.g., "`123456`") or a block array that mirrors the spatial arrangement of the net.
- **Tangram puzzle**. Because rotations in the image cannot be expressed succinctly in the image+text setting, we show only image-only and text-only variants. Each piece is labeled

alphabetically and encoded by a binary occupancy grid—rows of "`1`" indicate filled cells, yielding a representation that is both human-readable and unambiguous for MLLMs.

Together, these examples clarify the correspondence between the natural visual stimuli and the stripped-down symbolic forms used in our text-only experiments, as introduced in Section 3.3.

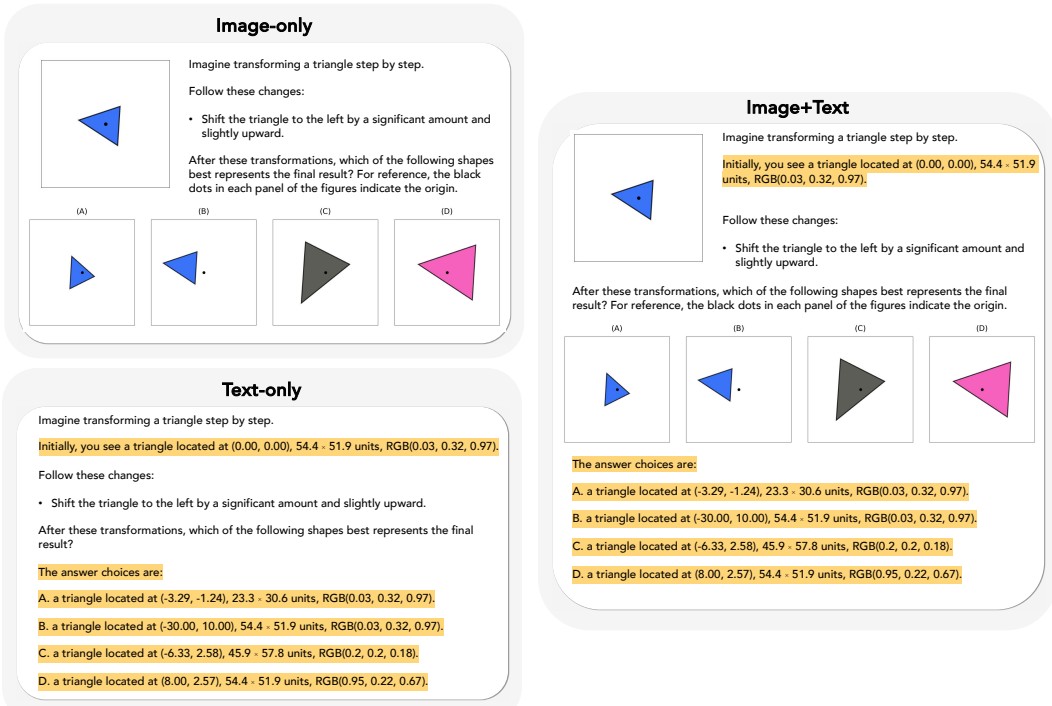

**Figure 19:** Visualizations of 2D transformations (w/ text instructions) in different representations (upper left: image-only, lower left: text-only, right: image+text).

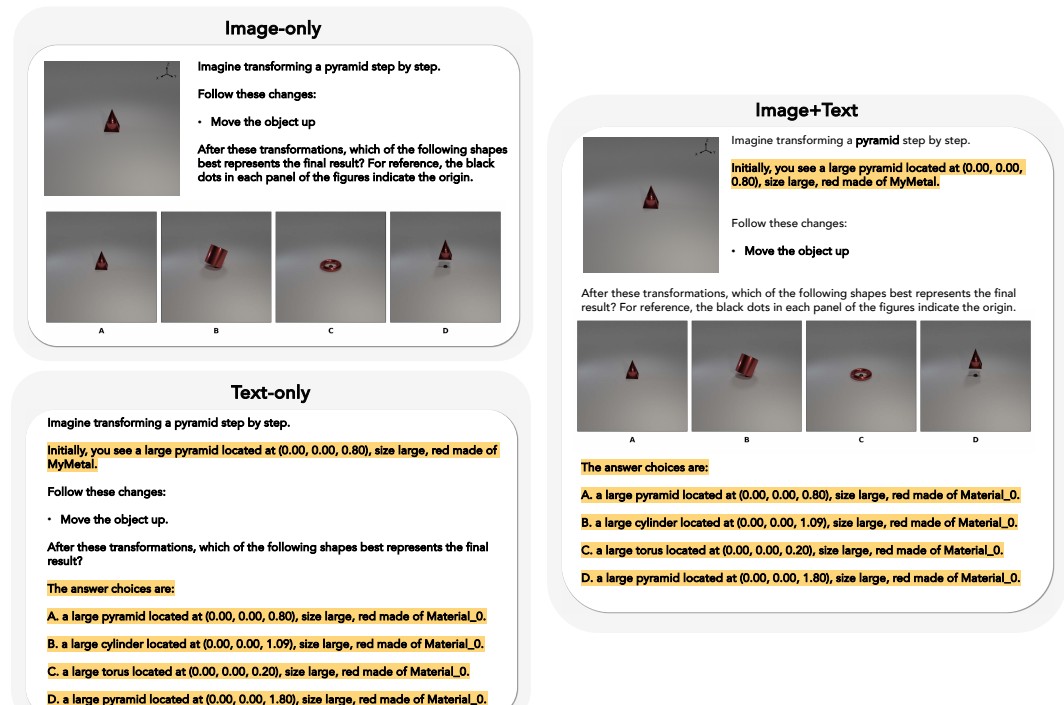

**Figure 20:** Visualizations of 3D transformations (w/ text instructions) in different representations (upper left: image-only, lower left: text-only, right: image+text).

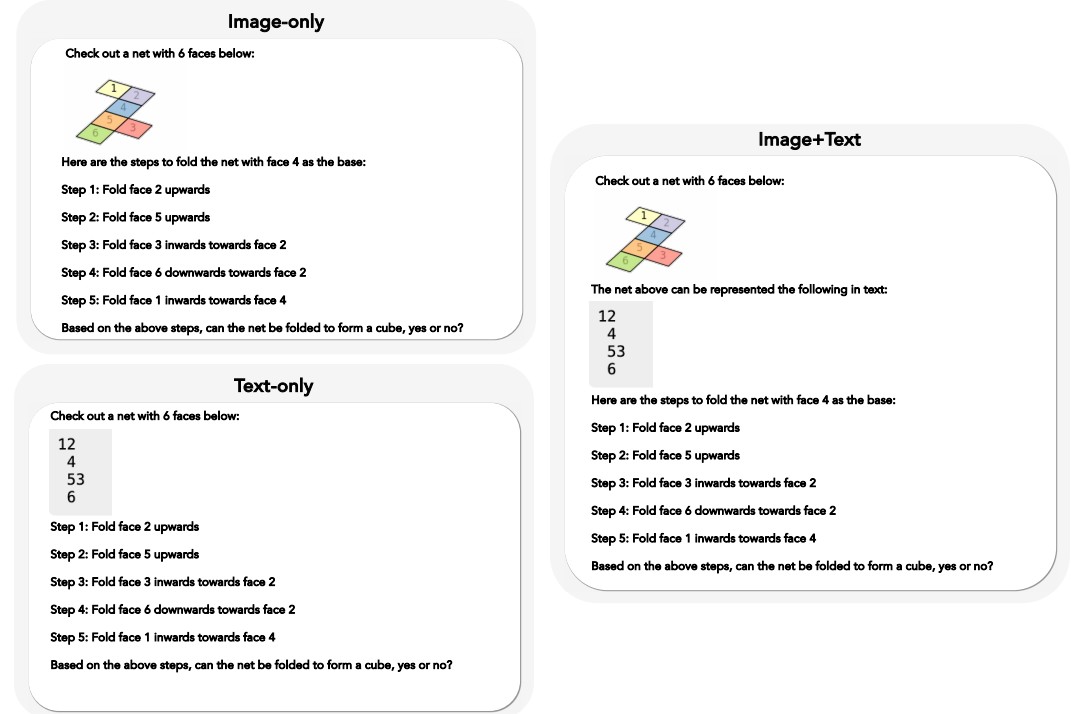

**Figure 21:** Visualizations of cube net folding in different representations (upper left: image-only, lower left: text-only, right: image+text).

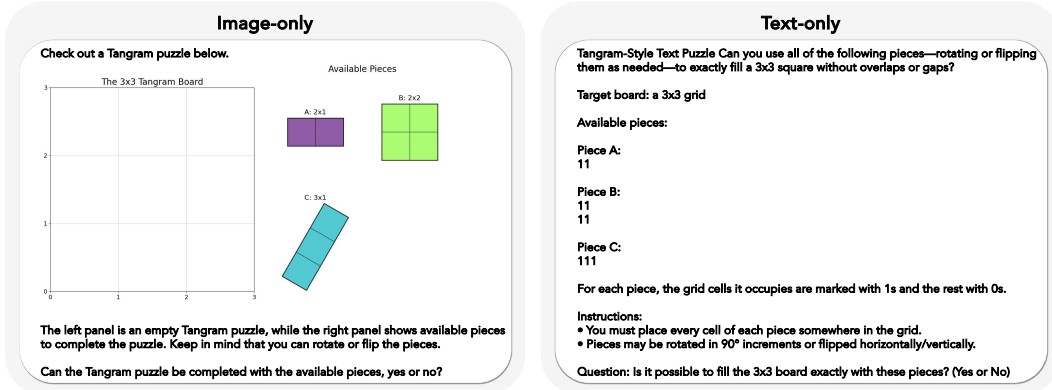

**Figure 22:** Visualizations of tangram in different representations (left: image-only, right: text-only).

## G.5    CASE STUDY

Figure 23 presents error cases on 2D transformation tasks from o1 and GPT-4o. The o1 explanation correctly identifies that a regular hexagon rotated 30° counter-clockwise will have a vertex pointing straight up, so it chooses panel B, but its justification overlooks potential differences in size, color, or shape variants among the answer choices. GPT-4o, meanwhile, mistakenly claims the hexagon looks "exactly the same" after a 30° rotation (confusing 60° symmetry with 30°) and vacillates between panels A and C, revealing it doesn't fully grasp the visual outcome of the specified rotation.

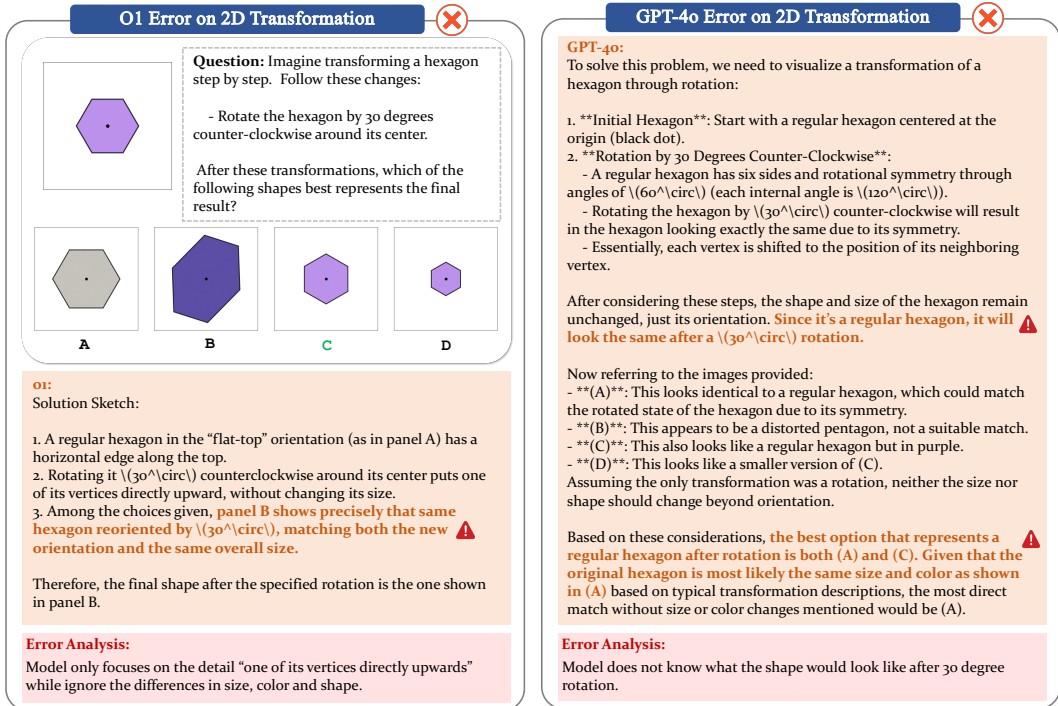

**Figure 23:** Error cases of o1 and GPT-4o on 2D transformation.

Figure 24 presents error cases on 3D transformation tasks from o1 and GPT-4o. The o1 response accurately places the torus up and left of the origin and selects panel A, but it overlooks that material/appearance differences could also rule out that choice. GPT-4o correctly sees that both panels A and D share the required displacement, yet it arbitrarily favors D, giving a justification ("clearer displacement") unrelated to the stated transformation criteria, so its final selection is wrong despite partially sound reasoning.

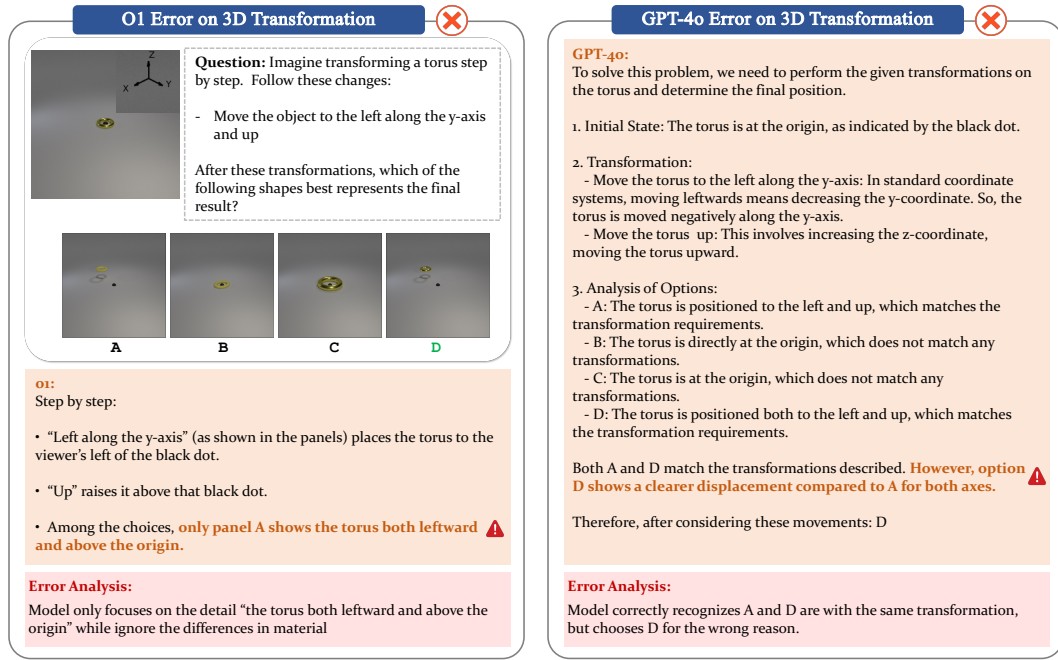

**Figure 24:** Error cases of o1 and GPT-4o on 3D transformation.

On cube net folding task, besides the perception error from Claude in Figure 5 and the text simulation error from GPT-4o in Figure 1, when provided with intermediate visual simulation, models like GPT-4o still struggles with understanding the intermediate visual cues. In Figure 25, GPT-4o asserts that face 1 becomes the cube's top surface, whereas face 1 is actually enclosed between faces 6 and 3. This misrepresentation of face adjacency highlights GPT-4o's difficulty in maintaining accurate 3D spactial cognition, which led to the wrong final result. This observation aligns with the conclusion from the perception probing test on cube nets in Section 3.3, where GPT-4o fails substantially on 3D perception than 2D perception.

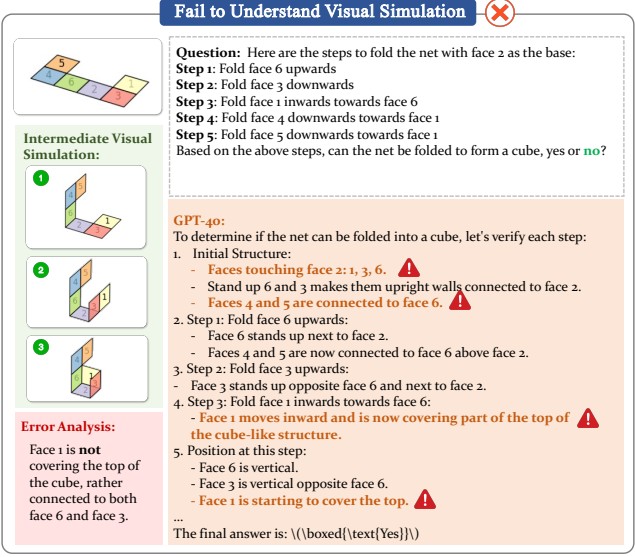

**Figure 25:** A error case from GPT-4o on cube net folding, where the model fails to understand the intermediate visual simulation.

For tangram puzzles, the left example of Figure 26 shows how Claude takes a shortcut by conducting an area-based feasibility check, correctly noting that the $4 \times 4$ board contains 16 unit squares, but it

erroneously counts the L-shaped piece C as 5 rather than its actual 7 squares. This miscalculation reduces the summed piece area to 14 instead of the correct 16, leading to conclude—incorrectly—that the puzzle cannot be completed. The right example of Figure 26 shows how Claude fails to reason about the piece positions when given clear step-by-step instructions. Claude correctly interprets the coordinate instructions and board dimensions, yet its spatial reasoning falters when projecting how the rotated pieces occupy the grid. It underestimates the extents of pieces B and D, inventing overlaps and gaps that never arise, and therefore erroneously concludes the puzzle cannot be completed. This error stems from an inaccurate mental simulation of spatial relationships.

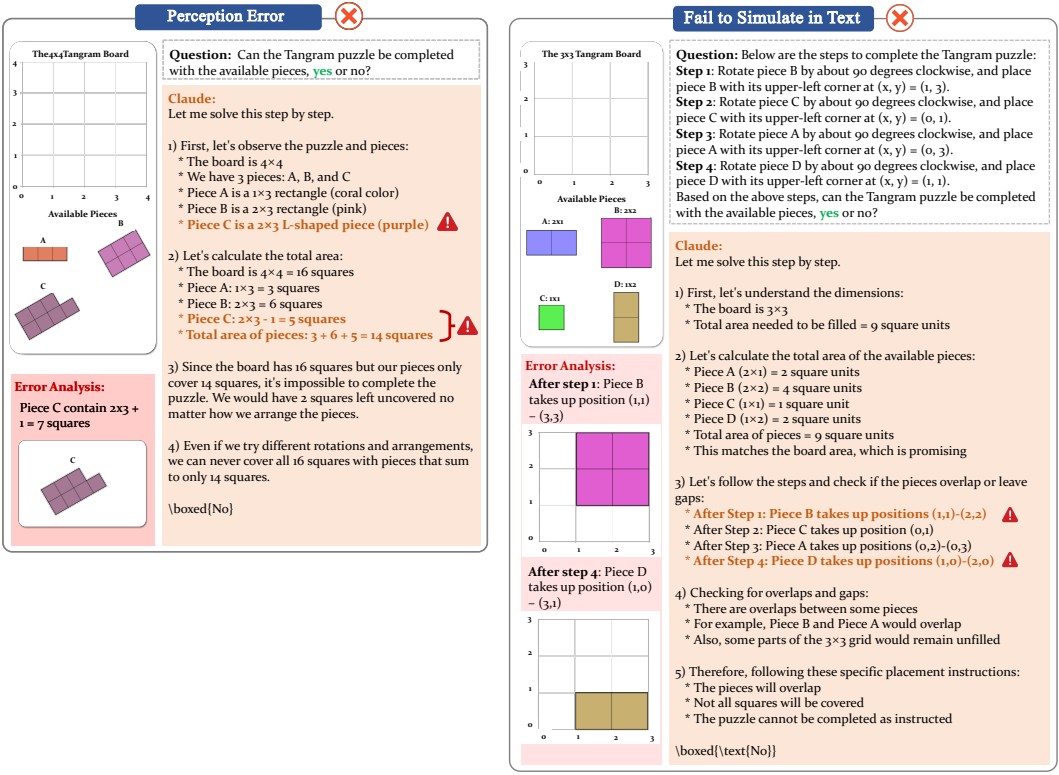

**Figure 26:** Left: A perception error case from Claude on tangram puzzle. Right: An error case from Claude on tangram puzzle, which failed to simulate the intermediate steps even when step-by-step instructions are given.

Figure 27 presents two error cases from Claude on temporal frame reasoning. In the left example, Claude correctly inferred the camera's left-to-right movement across the given frames, yet it mis-evaluated the viewpoints depicted in the answer choices and consequently selected the wrong completion frame. In the right example, the model erred even earlier, misconstruing the direction of camera motion itself; this foundational misinterpretation then led to an incorrect choice despite a seemingly systematic rationale.

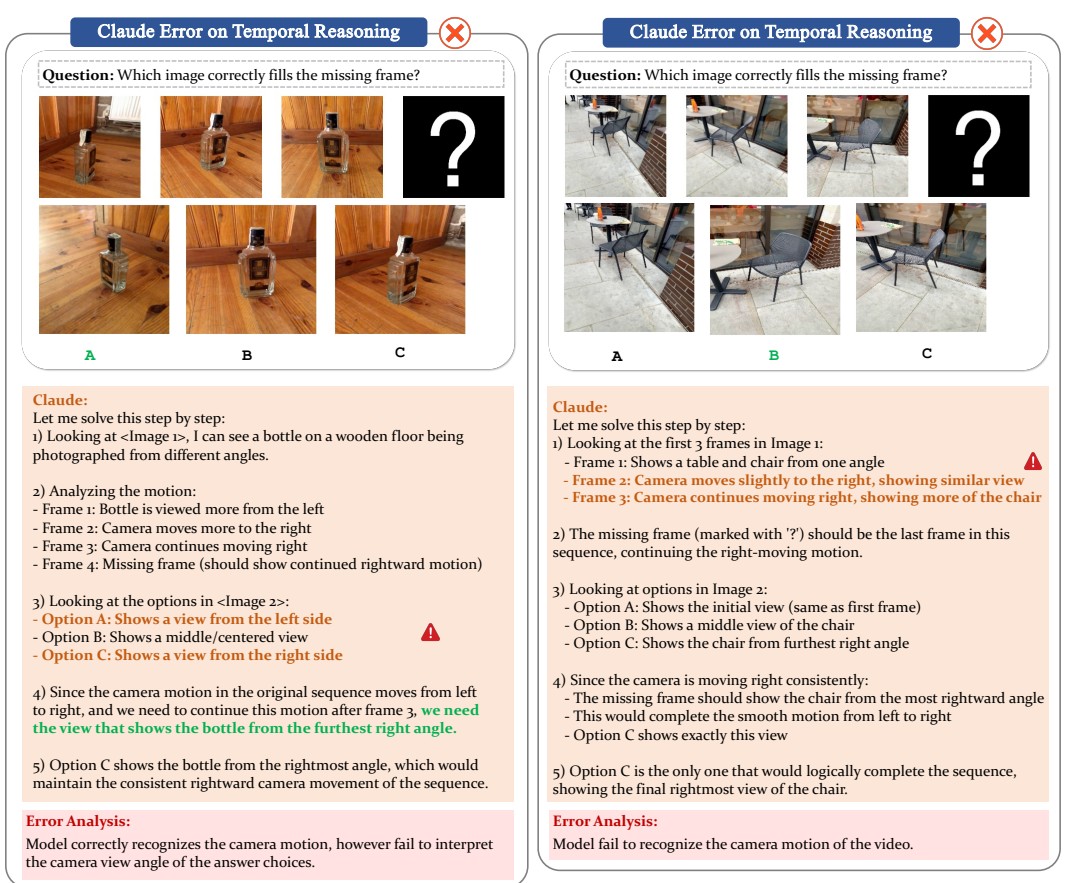

**Figure 27:** Error case on temporal frame reasoning.

Figure 28 presents an error for perspective reasoning from GPT-4o. GPT-4o misprojects the agent's top-down pose into egocentric space: it assumes the arrowed direction is pointing towards a curved seating area and therefore selects option A. A correct geometric mapping should conclude that the agent would instead stand next to the curved seating area. This error underscores the model's difficulty in reason about perspective changing in 3D space.

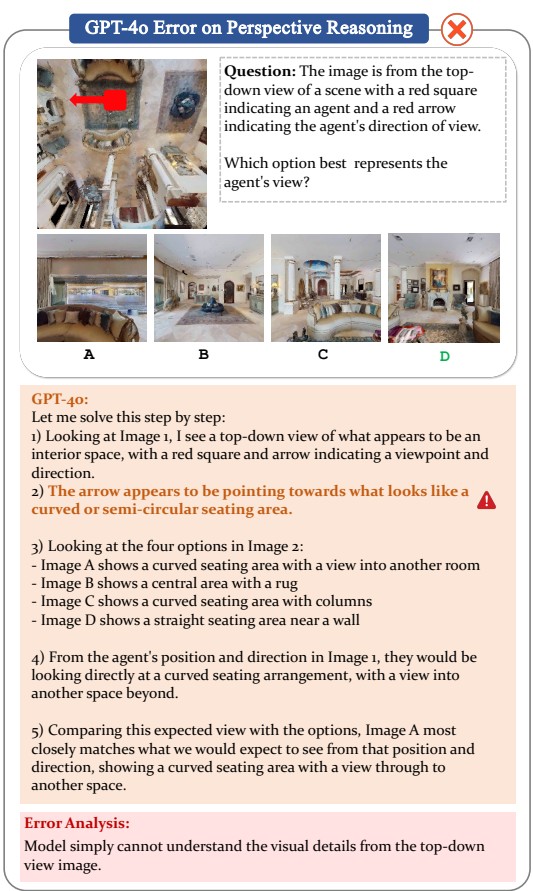

**Figure 28:** Error case on perspective reasoning.

Figure 29-31 presents error cases from Gemini-2.0-Flash(-Thinking), o1 and GPT-4o. on temporal frame reasoning.

Figure 34-33 presents errors for perspective reasoning from Gemini-2.0-Flash(-Thinking), o1 and Claude.

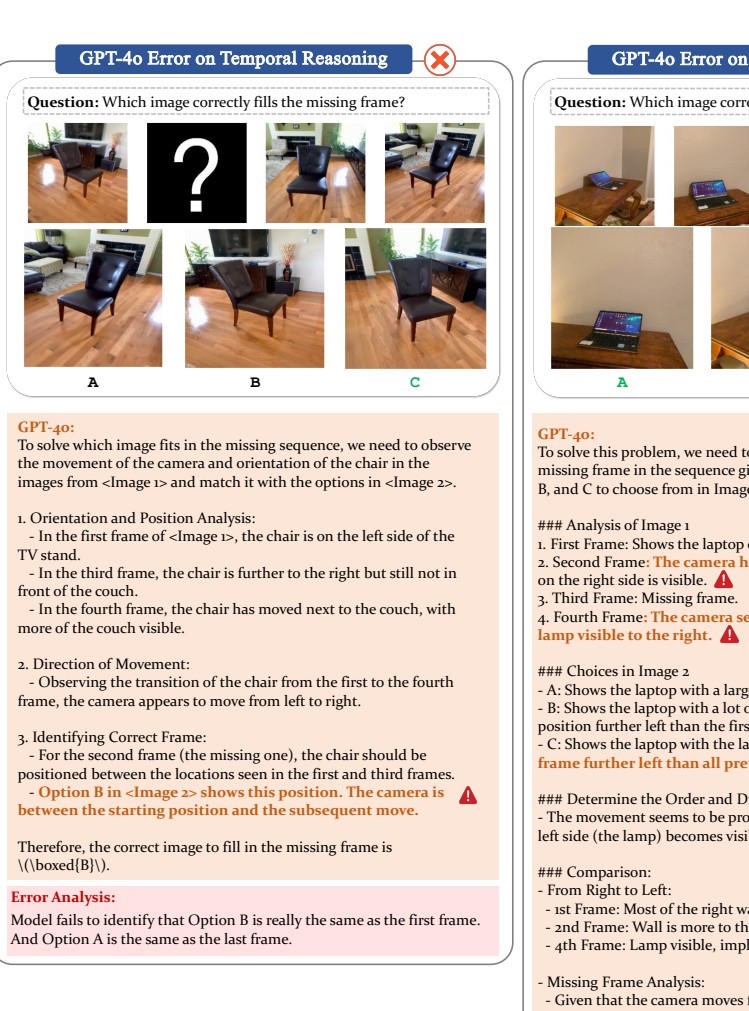

**GPT-4o Error on Temporal Reasoning** ✖

**Question:** Which image correctly fills the missing frame?

**GPT-4o:**
To solve which image fits in the missing sequence, we need to observe the movement of the camera and orientation of the chair in the images from <Image 1> and match it with the options in <Image 2>.

1. Orientation and Position Analysis:
   - In the first frame of <Image 1>, the chair is on the left side of the TV stand.
   - In the third frame, the chair is further to the right but still not in front of the couch.
   - In the fourth frame, the chair has moved next to the couch, with more of the couch visible.

2. Direction of Movement:
   - Observing the transition of the chair from the first to the fourth frame, the camera appears to move from left to right.

3. Identifying Correct Frame:
   - For the second frame (the missing one), the chair should be positioned between the locations seen in the first and third frames.
   - **Option B in <Image 2> shows this position. The camera is between the starting position and the subsequent move.** ⚠

Therefore, the correct image to fill in the missing frame is \(\boxed{B}\).

**Error Analysis:**
Model fails to identify that Option B is really the same as the first frame. And Option A is the same as the last frame.

**GPT-4o Error on Temporal Reasoning** ✖

**Question:** Which image correctly fills the missing frame?

**GPT-4o:**
To solve this problem, we need to determine which image correctly fills the missing frame in the sequence given in Image 1. We have images labeled A, B, and C to choose from in Image 2.

### Analysis of Image 1
1. First Frame: Shows the laptop on a desk with the wall visible to the right.
2. Second Frame: **The camera has moved to the left** as more of the wall on the right side is visible. ⚠
3. Third Frame: Missing frame.
4. Fourth Frame: **The camera seems to have moved further left, with a lamp visible to the right.** ⚠

### Choices in Image 2
- A: Shows the laptop with a large portion of the right wall visible.
- B: Shows the laptop with a lot of wall visible behind it, similar to a position further left than the first frame.
- C: Shows the laptop with the lamp close to it, **which suggests it's a frame further left than all previous.** ⚠

### Determine the Order and Direction:
- The movement seems to be progressing from right to left as more of the left side (the lamp) becomes visible in subsequent frames.

### Comparison:
- From Right to Left:
  - 1st Frame: Most of the right wall visible.
  - 2nd Frame: Wall is more to the left, but space still visible on the right.
  - 4th Frame: Lamp visible, implying maximum left position.

- Missing Frame Analysis:
  - Given that the camera moves from right to left:
    - A: Seems to show more of the right wall (first frame position).
    - B: Fits between the first two frames—it's between mostly right wall visibility and the lamp.
    - C: Shows the lamp, indicating a further left position than the final frame.

The best fit for the missing frame, which should logically be between the current second and fourth frames (moving left), is B.

**Error Analysis:**
Model fail to recognize the camera motion of the video, and the relative camera angle change between the answer choices and the provided frames.

**Figure 29:** Error cases from GPT-4o on temporal frame reasoning.

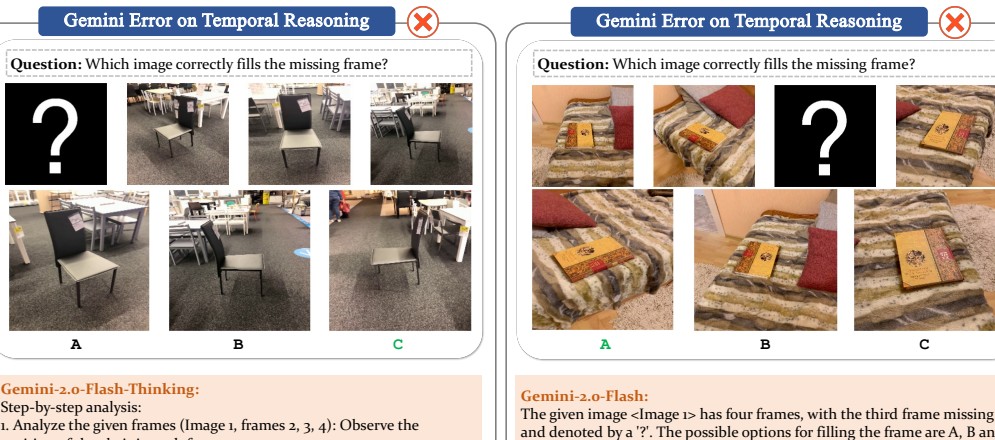

**Figure 30:** Error cases from Gemini models on temporal frame reasoning.

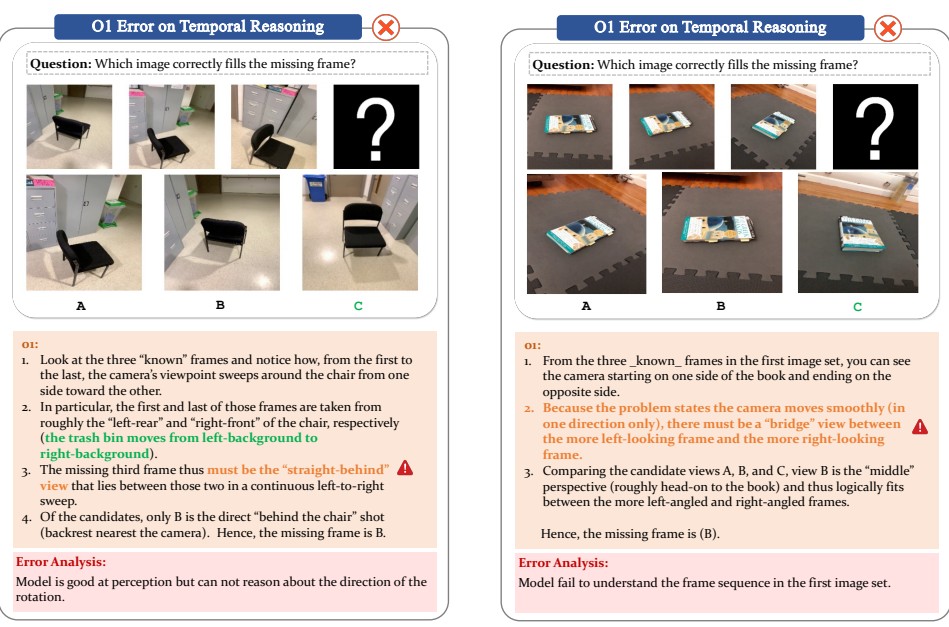

**Figure 31:** Error cases from o1 on temporal frame reasoning.

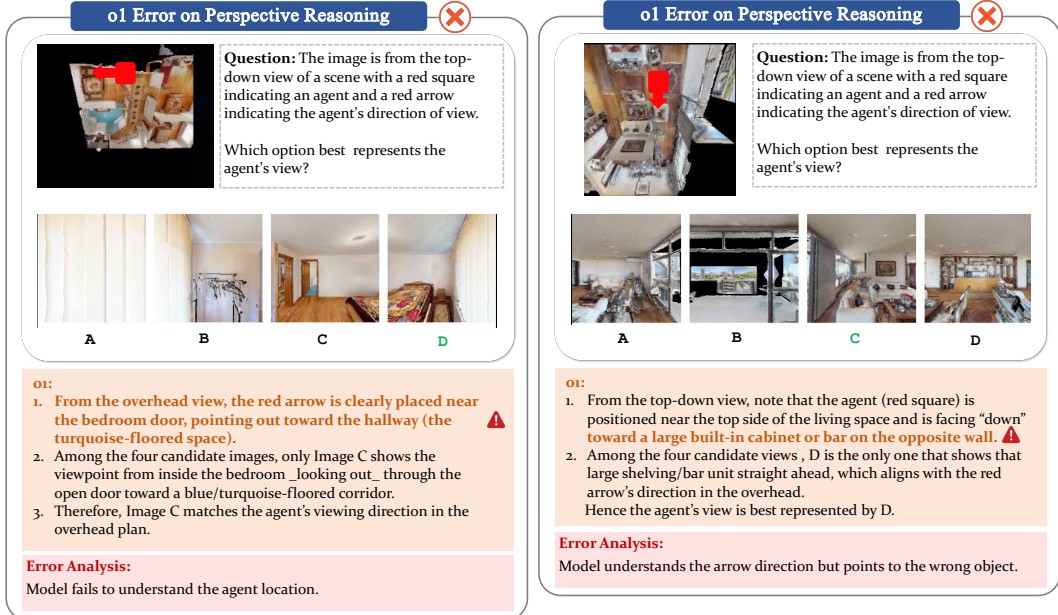

**Figure 32:** Error case from o1 on perspective reasoning.

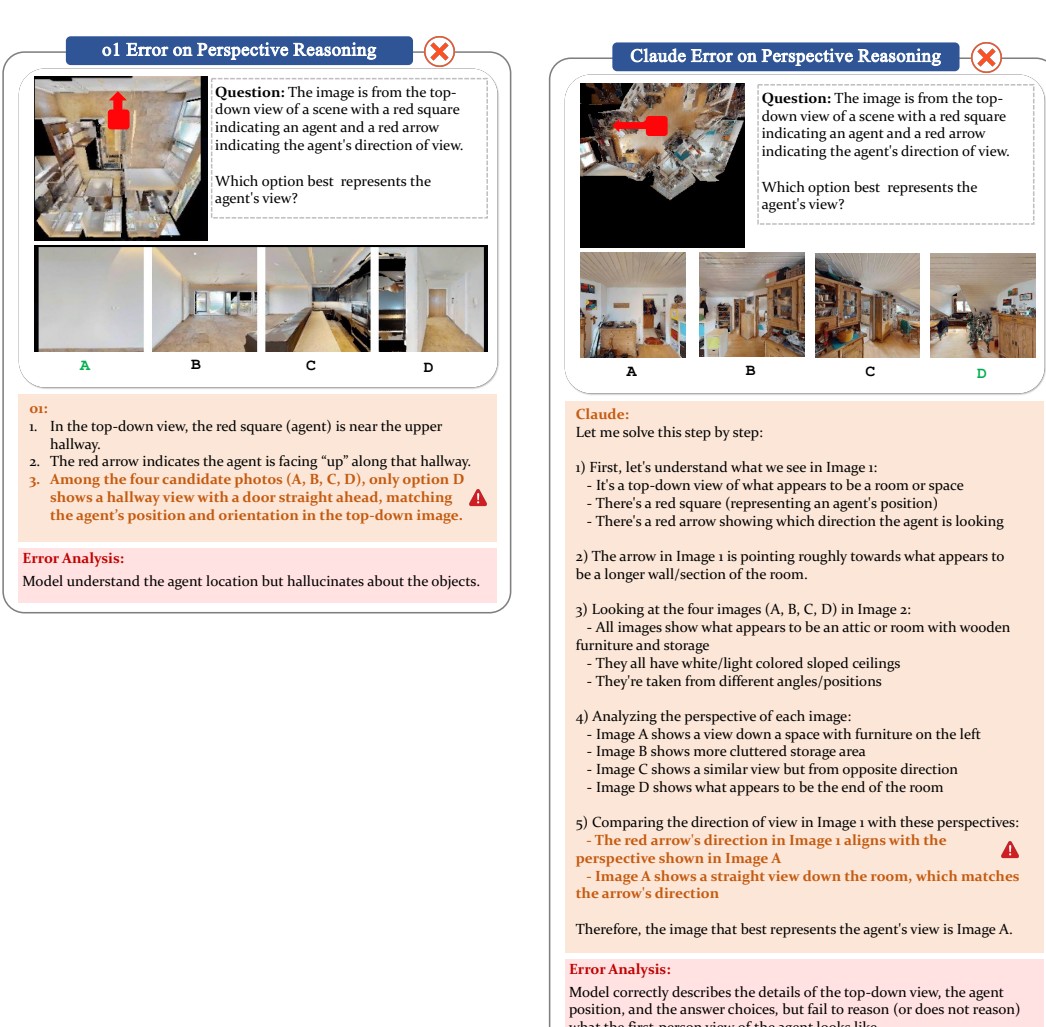

**Figure 33:** Error case from o1 and Claude on perspective reasoning.

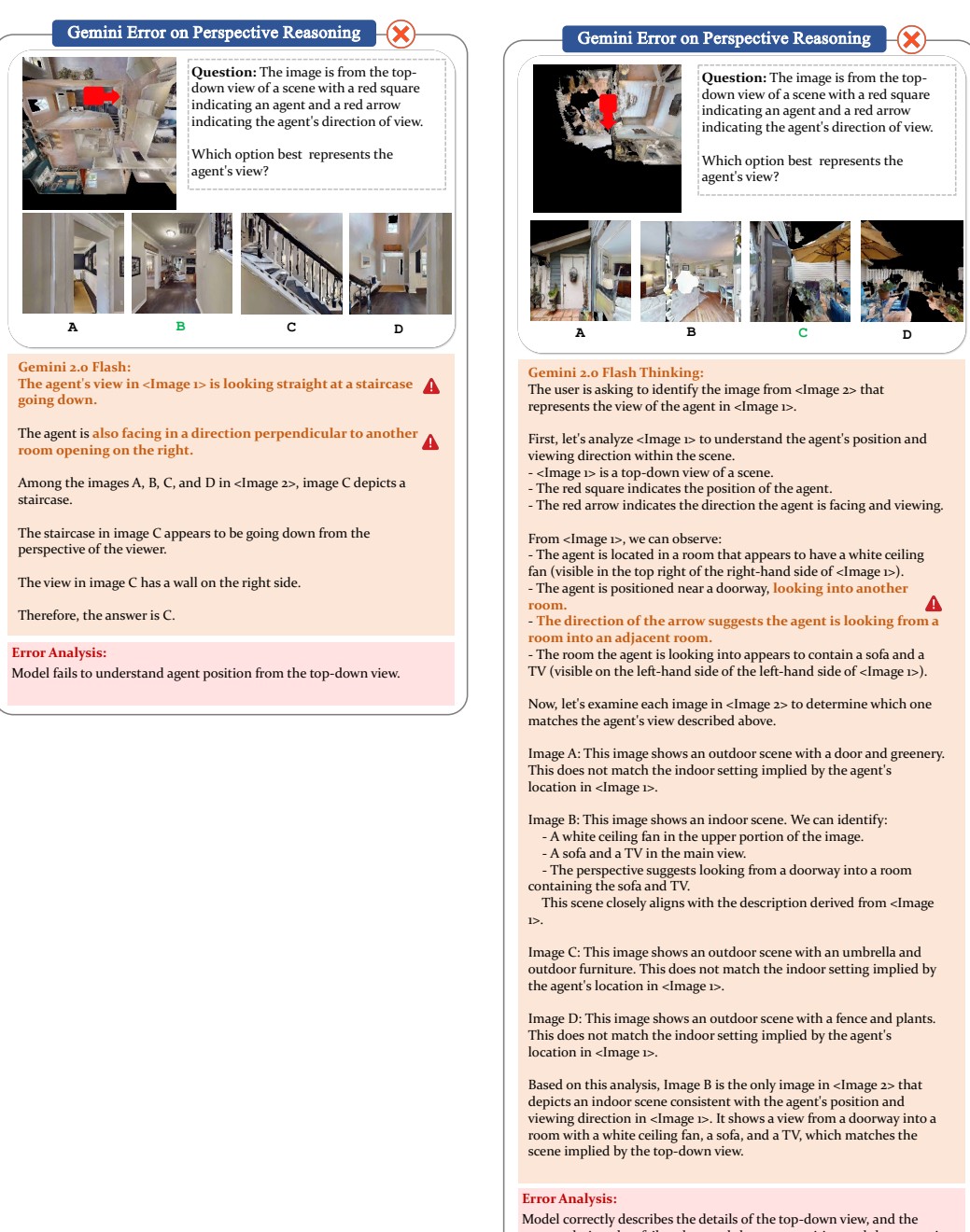

**Figure 34:** Error case from Gemini models on perspective reasoning.

## H  COMPLETE ANALYSIS RESULTS ON OTHER MODELS

**Model Performance on 2D/3D Individual Transformation Types.** Table 20 presents model accuracy across 2D visual analogy and text instruction tasks. Across the nine subtasks, adding visual simulation lifted accuracy for every model except in a few narrow cases, and the size of the gain correlates strongly with baseline capability. Closed-source leaders that were already solid on the raw pixel tasks—o1 ($\sim$ +3 points overall) and GPT-4o ($\sim$ +8 points)—were pushed into the mid-80 s and low-90 s, effectively reaching ceiling on the text-instruction variants, where gains were biggest (e.g., GPT-4o jumps +25 points on "Reflection" and +18 points on both "Rotation" and "Translation"). Mid-tier proprietary models such as Gemini 2.0 Flash ($\sim$ +5 points) and its "Flash Thinking" mode ($\sim$ +5.5 points) benefited even more on instructions than on analogies, narrowing the gap to

GPT-4-class systems. Open-source vision-language models lag a full generation behind—the best of them (InternVL 2.5-78B) still sits below 55% on average after simulation—but they, too, record healthy boosts of 6–12 points, chiefly on the analogy side. The lone regression is GPT-4o's –5 pt dip on "Reflection" analogies, suggesting that simulation may occasionally overwrite a correct latent heuristic. Overall, the pattern indicates that visual simulation chiefly helps models convert verbal transformation instructions into precise spatial operations, while stronger base perception/reasoning models harvest the largest absolute improvements and approach human-like proficiency.

Table 21 presents model accuracy across 3D visual analogy and text instruction tasks. Visual simulation gives 3D spatial reasoning a measurable—but more uneven—boost than in 2D: averaged over all eight subtasks, every proprietary model gains between ~+2 points (GPT-4o, o1) and +6 points (Claude-3.5 Sonnet, Gemini-Flash Thinking), while the open-source field improves by ~ +4–7 points—except InternVL, which slips a point. Gains concentrate in the conceptually harder operations: across models, Shearing (both analogy +6.6 points and instruction +6.6 points) and Rotation-instruction ( +6.4 points) see the largest lifts, whereas Translation under visual analogy actually falls slightly (–0.9 points), echoing a smaller 2D reflection dip. Even after simulation, closed-source leaders plateau in the high-60s to mid-70s on most 3D subtasks—roughly 15 points below their 2D ceilings—indicating that depth-aware transformations remain a major bottleneck. Open-source VL models still trail a full generation ($\leq$45% average), but their sharper relative gains suggest they, too, leverage synthetic roll-outs to bridge language and geometry.

| Model | 2D Transformations w/ Visual Analogy | | | | | 2D Transformations w/ Text Instruction | | | |
|---|---|---|---|---|---|---|---|---|---|
| | Reflection | Rotation | Shearing | Scaling | Translation | Reflection | Rotation | Scaling | Translation |
| *Without Visual Simulation* | | | | | | | | | |
| GPT-4o | 82.1 | 69.8 | 53.7 | 88.5 | 72.0 | 65.8 | 67.8 | 90.6 | 73.3 |
| Claude-3.5 Sonnet | 75.0 | 60.9 | 55.8 | 87.4 | 71.2 | 63.8 | 58.9 | 85.9 | 66.5 |
| Gemini2.0 Flash | 85.7 | 63.8 | 51.0 | 84.4 | 71.4 | 65.8 | 62.3 | 88.4 | 70.3 |
| Gemini2.0 Flash Thinking | 52.4 | 48.9 | 46.9 | 71.9 | 55.1 | 63.2 | 60.6 | 83.0 | 67.8 |
| o1 | 92.9 | 70.7 | 59.2 | 83.3 | 84.0 | 89.5 | 78.1 | 92.2 | 92.2 |
| LLaVA-OneVision | 7.1 | 25.9 | 32.7 | 24.4 | 25.4 | 31.7 | 33.1 | 51.0 | 34.6 |
| Qwen2.5-VL-72B | 57.1 | 38.8 | 34.7 | 64.4 | 42.3 | 29.3 | 49.6 | 62.5 | 38.8 |
| InternVL2.5-78B | 35.7 | 41.4 | 34.7 | 45.6 | 36.6 | 41.5 | 51.1 | 75.0 | 51.9 |
| *With Visual Simulation* | | | | | | | | | |
| GPT-4o | 76.9 | 72.8 | 54.8 | 91.9 | 80.0 | 91.2 | 86.0 | 93.2 | 91.5 |
| Claude-3.5 Sonnet | 73.1 | 70.9 | 50.0 | 85.5 | 73.9 | 55.9 | 72.9 | 83.8 | 73.9 |
| Gemini2.0 Flash | 73.1 | 70.9 | 59.5 | 85.5 | 74.5 | 79.4 | 74.8 | 90.5 | 78.2 |
| Gemini2.0 Flash Thinking | 61.5 | 68.2 | 40.5 | 71.0 | 56.4 | 70.6 | 68.2 | 89.2 | 73.9 |
| o1 | 80.8 | 80.6 | 54.8 | 87.1 | 84.2 | 100 | 93.5 | 94.6 | 97.6 |
| LLaVA-OneVision | 15.4 | 30.1 | 31.0 | 30.6 | 24.8 | 20.6 | 41.1 | 48.6 | 33.9 |
| Qwen2.5-VL-72B | 65.4 | 56.3 | 35.7 | 71.0 | 57.0 | 41.2 | 40.2 | 60.8 | 39.4 |
| InternVL2.5-78B | 69.2 | 43.7 | 33.3 | 59.7 | 47.3 | 50.0 | 53.3 | 73.0 | 53.9 |

**Table 20:** Model Performance With or Without Visual Simulation across 2D Transformation types in Visual Analogy and Text Instruction Tasks.

| Model | 3D Transformations w/ Visual Analogy | | | | 3D Transformations w/ Text Instruction | | | |
|---|---|---|---|---|---|---|---|---|
| | Rotation | Shearing | Scaling | Translation | Rotation | Shearing | Scaling | Translation |
| *Without Visual Simulation* | | | | | | | | |
| GPT-4o | 60.7 | 55.7 | 76.0 | 80.1 | 60.1 | 46.9 | 71.1 | 71.2 |
| Claude-3.5 Sonnet | 50.0 | 46.2 | 63.3 | 62.6 | 45.9 | 40.4 | 55.6 | 53.4 |
| Gemini2.0 Flash | 54.2 | 53.9 | 63.3 | 73.0 | 55.86 | 44.44 | 61.90 | 51.63 |
| Gemini2.0 Flash Thinking | 42.4 | 43.6 | 61.5 | 63.8 | 37.8 | 32.7 | 52.5 | 55.7 |
| o1 | 65.6 | 58.1 | 76.7 | 85.6 | 61.3 | 46.3 | 70.5 | 73.9 |
| LLaVA-OneVision | 18.8 | 29.1 | 28.9 | 25.3 | 27.0 | 19.4 | 41.0 | 30.7 |
| Qwen2.5-VL-72B | 36.5 | 40.2 | 61.1 | 46.6 | 36.9 | 33.3 | 47.6 | 45.1 |
| InternVL2.5-78B | 31.2 | 30.8 | 51.1 | 37.4 | 37.8 | 32.4 | 60.0 | 40.5 |
| *With Visual Simulation* | | | | | | | | |
| GPT-4o | 64.3 | 64.3 | 78.2 | 76.0 | 62.6 | 54.7 | 75.3 | 68.5 |
| Claude-3.5 Sonnet | 51.2 | 59.5 | 69.2 | 59.7 | 55.6 | 48.0 | 64.5 | 59.3 |
| Gemini2.0 Flash | 46.4 | 64.3 | 62.8 | 68.2 | 60.9 | 49.5 | 64.9 | 56.1 |
| Gemini2.0 Flash Thinking | 50.0 | 47.6 | 60.3 | 66.7 | 48.5 | 46.7 | 59.1 | 59.3 |
| o1 | 63.1 | 63.1 | 76.9 | 79.8 | 69.7 | 50.7 | 79.6 | 74.1 |
| LLaVA-OneVision | 27.4 | 28.6 | 32.1 | 29.5 | 27.3 | 26.7 | 45.2 | 35.2 |
| Qwen2.5-VL-72B | 46.4 | 54.8 | 69.2 | 55.0 | 45.5 | 34.7 | 48.4 | 46.3 |
| InternVL2.5-78B | 31.0 | 28.6 | 43.6 | 32.6 | 43.4 | 37.3 | 57.0 | 40.7 |

**Table 21:** Model Performance With or Without Visual Simulation across 3D Transformation types in Visual Analogy and Text Instruction Tasks.

**o3 Performance across Input Representations.** In Table 3, we reported GPT-4o performance under different input representations (text-only, image-only, image+text). Table 22 extends this analysis to o3. For 2D transformations and cube nets, the image+text combination achieves the best performance (98.5% and 69.0%, respectively). For tangram, text-only is strongest (87.7%), consistent with strong language priors about piece counts and simple configurations. For 3D transformations, image-only performs best (73.5%), and image+text does not recover the full image-only advantage (69.6%). These results verify Table 3's trends with a reasoning model: while text can help on more language-friendly tasks, strong performance on 3D and other spatial tasks still critically depends on visual perception and non-verbal reasoning.

| Input | 2D Trans. | 3D Trans. | Cube Nets | Tangram |
|---|---|---|---|---|
| Text-only | 92.8 | 71.0 | 66.1 | **87.7** |
| Image-only | 95.2 | **73.5** | 64.3 | 78.2 |
| Image+Text | **98.5** | 69.6 | **69.0** | – |

**Table 22:** o3 performance without visual simulation under different input representations, extending Table 3 (GPT-4o).

**Task complexity vs. performance.** Table 24 presents model performance across different task difficulties for 2D and 3D transformations. Adding visual simulation helps most when tasks get tougher, but the effect differs by setting. For 2D text instructions tasks, we observe big boost – closed-source models jump about 10-20 points on medium and hard tasks, often hitting 90%+. For 2D visual analogy tasks, we observe smaller lift—several points on easy, up to 1̃0 points on medium/hard. For 3D tasks, only a few-point gain, and some models slip on the hardest visual analogy tasks, showing 3D reasoning is still hard. Open-source MLLMs stay well behind; their scores move up and down unpredictably, meaning they haven't yet learned to use the simulated views well.

Table 23 presents model performance across different number of transformation steps for 2D and 3D transformations. Models struggle more as the number of transformation steps grows, and visual simulation mainly fixes that. Without simulation, accuracy often peaks at one or two steps and drops at three—especially in 3D visual-analogy, where GPT-4o falls from 73% (N = 2) to 49% (N = 3). When simulation is added, scores for the multi-step cases (N = 2–3) jump 10–15 points for the top proprietary systems and a few points for open-source ones, erasing most of the earlier decline in 2D tasks and cutting the 3D drop roughly in half. Single-step problems were already easy for the best

models and see little change. Overall, simulation is most useful for longer, instruction-driven chains of transforms, while depth-heavy 3D sequences remain the hardest setting.

| Model | 2D Visual Analogy | | | 2D Text Instruction | | | 3D Visual Analogy | | | 3D Text Instruction | | |
|---|---|---|---|---|---|---|---|---|---|---|---|---|
| | N=1 | N=2 | N=3 | N=1 | N=2 | N=3 | N=1 | N=2 | N=3 | N=1 | N=2 | N=3 |
| *Without Visual Simulation* | | | | | | | | | | | | |
| GPT-4o | 60.46 | 74.84 | 73.86 | 67.27 | 77.56 | 73.55 | 62.75 | 73.37 | 48.69 | 63.07 | 63.40 | 60.78 |
| Claude-3.5 Sonnet | 63.73 | 75.82 | 65.69 | 65.17 | 65.02 | 60.61 | 45.10 | 57.35 | 57.35 | 50.98 | 55.23 | 45.75 |
| Gemini2.0 Flash | 64.71 | 73.53 | 68.53 | 63.96 | 76.24 | 70.25 | 61.76 | 60.78 | 63.73 | 46.08 | 56.86 | 56.86 |
| Gemini2.0 Flash Thinking | 54.58 | 52.94 | 55.56 | 61.71 | 67.33 | 71.07 | 47.71 | 53.92 | 57.19 | 45.59 | 50.00 | 20.59 |
| o1 | 66.7 | 81.4 | 82.4 | 82.0 | 89.1 | 89.3 | 66.67 | 72.55 | 77.45 | 61.76 | 66.67 | 62.75 |
| LLaVA-OneVision | 30.39 | 26.47 | 24.51 | 49.57 | 33.70 | 31.53 | 25.49 | 28.43 | 22.55 | 30.39 | 30.39 | 24.51 |
| InternVL2.5-78B | 43.14 | 34.31 | 42.16 | 61.74 | 52.17 | 50.45 | 40.2 | 29.41 | 36.27 | 34.31 | 48.04 | 40.2 |
| Qwen2.5-VL-72B | 50.00 | 45.10 | 41.18 | 55.65 | 36.96 | 40.54 | 48.04 | 42.16 | 45.10 | 38.24 | 43.14 | 41.18 |
| *With Visual Simulation* | | | | | | | | | | | | |
| GPT-4o | - | 78.43 | 73.53 | - | 88.04 | 90.57 | - | 70.59 | 72.55 | - | 61.76 | 68.63 |
| Claude-3.5 Sonnet | - | 70.59 | 70.59 | - | 71.74 | 72.64 | - | 56.86 | 57.84 | - | 65.69 | 50.98 |
| Gemini2.0 Flash | - | 69.6 | 73.5 | - | 80.43 | 77.40 | - | 61.76 | 59.80 | - | 61.76 | 53.92 |
| Gemini2.0 Flash Thinking | - | 46.08 | 58.82 | - | 79.35 | 67.92 | - | 55.88 | 60.78 | - | 53.92 | 53.92 |
| o1 | - | 73.4 | 85.3 | - | 94.6 | 97.2 | - | 70.6 | 75.5 | - | 70.6 | 69.6 |
| LLaVA-OneVision | - | 30.39 | 25.49 | - | 38.04 | 34.91 | - | 28.43 | 28.43 | - | 36.27 | 29.41 |
| InternVL2.5-78B | - | 39.22 | 51.96 | - | 56.52 | 52.83 | - | 25.49 | 35.29 | - | 46.08 | 39.22 |
| Qwen2.5-VL-72B | - | 51.96 | 58.82 | - | 43.48 | 41.51 | - | 49.02 | 58.82 | - | 47.06 | 43.14 |

**Table 23:** Model Performance With or Without Visual Simulation across number of transformation steps (N) in 2D/3D Visual Analogy and Text Instruction Tasks.

| Model | 2D Visual Analogy | | | 2D Text Instruction | | | 3D Visual Analogy | | | 3D Text Instruction | | |
|---|---|---|---|---|---|---|---|---|---|---|---|---|
| | easy | medium | hard | easy | medium | hard | easy | medium | hard | easy | medium | hard |
| *Without Visual Simulation* | | | | | | | | | | | | |
| GPT-4o | 80.4 | 67.3 | 61.4 | 76.2 | 70.4 | 71.3 | 74.2 | 64.9 | 65.7 | 69.0 | 63.1 | 55.2 |
| Claude-3.5 Sonnet | 76.5 | 66.7 | 62.1 | 68.7 | 61.8 | 59.4 | 54.9 | 54.4 | 50.5 | 55.6 | 52.0 | 44.4 |
| Gemini 2.0 Flash | 78.4 | 63.7 | 64.7 | 75.0 | 67.2 | 67.3 | 67.6 | 59.8 | 58.8 | 56.9 | 52.9 | 50.0 |
| Gemini 2.0 Flash Think | 66.3 | 52.3 | 44.4 | 65.5 | 69.4 | 65.4 | 54.6 | 53.9 | 50.3 | 48.0 | 44.7 | 46.1 |
| o1 | 83.3 | 77.5 | 69.6 | 90.6 | 81.1 | 89.1 | 78.4 | 70.6 | 67.7 | 69.6 | 64.7 | 56.9 |
| LLaVA-OneVision | 22.6 | 32.4 | 26.5 | 39.5 | 46.3 | 29.2 | 25.5 | 20.6 | 30.4 | 31.4 | 28.4 | 25.5 |
| InternVL 2.5-78B | 45.1 | 40.2 | 34.3 | 63.2 | 50.9 | 50.0 | 32.4 | 34.3 | 39.2 | 48.0 | 37.3 | 37.3 |
| Qwen 2.5-VL-72B | 57.8 | 40.2 | 38.2 | 50.9 | 41.7 | 41.7 | 55.9 | 40.2 | 39.2 | 42.2 | 38.2 | 42.2 |
| *With Visual Simulation* | | | | | | | | | | | | |
| GPT-4o | 80.9 | 79.4 | 67.7 | 91.6 | 89.4 | 86.9 | 80.9 | 75.0 | 58.8 | 75.0 | 64.7 | 55.9 |
| Claude-3.5 Sonnet | 76.5 | 72.1 | 63.2 | 78.9 | 65.2 | 72.1 | 67.7 | 52.9 | 51.5 | 66.2 | 57.4 | 51.5 |
| Gemini 2.0 Flash | 79.4 | 72.1 | 63.2 | 81.7 | 86.4 | 67.2 | 64.7 | 58.8 | 58.8 | 57.4 | 55.9 | 60.3 |
| Gemini 2.0 Flash Think | 54.4 | 55.9 | 47.1 | 76.1 | 74.2 | 68.9 | 72.1 | 54.4 | 48.5 | 63.2 | 48.5 | 50.0 |
| o1 | 80.9 | 82.4 | 75.0 | 94.4 | 98.5 | 95.1 | 85.3 | 69.1 | 64.7 | 73.5 | 75.0 | 61.8 |
| LLaVA-OneVision | 36.8 | 19.1 | 27.9 | 39.4 | 34.9 | 34.4 | 26.5 | 20.6 | 38.2 | 45.6 | 25.0 | 27.9 |
| InternVL 2.5-78B | 57.4 | 44.1 | 35.3 | 64.8 | 48.5 | 49.2 | 23.5 | 27.9 | 39.7 | 55.9 | 27.9 | 44.1 |
| Qwen 2.5-VL-72B | 72.1 | 50.0 | 44.1 | 59.2 | 30.3 | 36.1 | 63.2 | 50.0 | 48.5 | 47.1 | 44.1 | 44.1 |

**Table 24:** Model Performance With or Without Visual Simulation across different difficulty levels in 2D/3D Visual Analogy and Text Instruction Tasks.

**2D and 3D Perception Probing with Cube Nets.** Table 25 presents model performance on 2D and 3D perception probing questions about cube nets, in comparison to the success rate on cube net folding task. The results show that success on cube-net folding is driven by a model's 3D perception, not its 2D eyesight. All closed-source systems (and several open-source ones) already read colors and 2D face connectivity at or near ceiling, yet their cube-net scores diverge sharply.

When we compare cube accuracy ( ✗VSim column) with each perceptual measure, the strongest linear relationship is with the 3D "Folded?" test (Pearson r ≈ 0.89), while 2D connectivity (r ≈ 0.68) and color (r ≈ 0.72) are weaker. Gemini Flash illustrates the pattern: it pairs the top 3D perception score (69%) with the best cube-net performance (65%), whereas GPT-4o and InternVL match its 2D vision but lag 10-20 points on both 3D perception and cube folding. In short, being able to judge how faces come together in depth—rather than recognizing colors or flat adjacencies—largely determines how well a model can reason about folded cubes.

| Model | 2D Perception | | 3D Perception | Cube Net Performance | |
|---|---|---|---|---|---|
| | Color | Connectivity | Folded? | ✗VSim | ✓Vsim |
| Random | 25.0 | 50.0 | 50.0 | 50.5 | 50.5 |
| *Closed-source Models* | | | | | |
| GPT-4o | **100.0** | **94.1** | 57.4 | 52.5 | 49.1 |
| Gemini-2.0-Flash | **100.0** | 84.9 | 68.8 | 65.0 | **65.5** |
| Gemini-2.0-Flash-Thinking | 99.0 | 49.4 | 54.3 | 39.8 | 62.8 |
| *Open-source Models* | | | | | |
| LLaVA-OneVision | 88.0 | 10.0 | 22.0 | 28.5 | 34.2 |
| InternVL 2.5-78B | 92.0 | 86.0 | 40.2 | 43.5 | 41.0 |
| Qwen 2.5-VL-72B | 96.0 | 81.7 | 42.1 | 35.2 | 53.4 |

**Table 25:** 2D and 3D perception performance in cube net folding.

**Extended Perception Probing.** Beyond cube nets, we extend the perception probing to 2D/3D transformation tasks and tangram puzzles using GPT-4o and o3. For each task, we isolate individual perceptual attributes (e.g., color, shape, position, size for 2D/3D; counting, overlap, rotation for tangram) and test whether the model can correctly identify them. As shown in Table 26, both models perform well above chance on most basic perceptual subtasks (tangram count/position, 2D/3D color and shape), often near ceiling. However, some aspects remain challenging: 2D size identification (51.1% for GPT-4o, 55.6% for o3), 3D material recognition (55.6% and 56.3%), and fine-grained tangram rotation (53.0% and 67.5%). When contrasted with much lower accuracies on the full reasoning tasks, these perception scores indicate that many failures are not due to basic perception alone. For most tasks, models can identify the right pieces, colors, and positions; they struggle to perform and track multi-step spatial transformations and composition.

| Task | Subtask | Random | GPT-4o | o3 |
|------|---------|--------|--------|-----|
| | *Tangram Puzzles* | | | |
| | Count | 25 | 90.0 | 100.0 |
| | Overlap (yes/no) | 50 | 75.0 | 92.0 |
| | Position (which piece) | 25 | 71.0 | 96.0 |
| | Rotation | 25 | 53.0 | 67.5 |
| | Size+Count | 25 | 74.1 | 88.9 |
| | Intermediate placement | 25 | 71.0 | 96.0 |
| | *2D Transformations* | | | |
| | Color (same/different) | 50 | 89.5 | 94.7 |
| | Shape (same/different) | 50 | 94.8 | 93.5 |
| | Position (closest to origin) | 25 | 71.3 | 83.0 |
| | Size (largest/smallest) | 25 | 51.1 | 55.6 |
| | *3D Transformations* | | | |
| | Color (same/different) | 50 | 96.2 | 96.2 |
| | Shape (same/different) | 50 | 97.5 | 95.9 |
| | Position (closest to origin) | 25 | 75.0 | 75.0 |
| | Material (same/different) | 50 | 55.6 | 56.3 |

**Table 26:** Extended perception probing results (%) for GPT-4o and o3 across tangram, 2D, and 3D tasks. Each subtask isolates a specific perceptual attribute. Random baseline shown for reference.

**Question-only vs. Question+Steps** As shown in Table 27, adding explicit reasoning steps ("Q + Steps") has opposite effects on cube-net tasks for the two model groups: open-source models gain, while closed-source ones do not. The three open-source VL models jump a mean + 20 points on cube nets (driven by LLaVA's + 40 pts), whereas the five proprietary models average a small decline (-1 pt, with mixed signs). On tangram puzzles, however, the pattern converges: every model—open or closed—drops sharply once reasoning steps are included, with average losses of about -24 pts for closed-source and -19 pts for open-source models. Again, the trivial solution on tangram puzzles would be comparing the total areas of all available pieces and the grid area, which can easily lead to 75% performance. This result suggest that the models cannot leverage explicit text reasoning steps.

| Model | Cube Nets | | | Tangram Puzzles | | |
|-------|-----------|---------|------|-----------------|---------|------|
| | Q-only | Q+Steps | Δ | Q-only | Q+Steps | Δ |
| | *Closed-source Models* | | | | | |
| GPT-4o | 50.2 | 50.4 | **+0.2** | 62.4 | 34.7 | **-27.7** |
| Claude-3.5 Sonnet | 51.5 | 46.4 | **-5.1** | 71.1 | 41.9 | **-29.2** |
| Gemini-2.0 Flash | 47.4 | 51.5 | **+4.1** | 72.8 | 59.8 | **-13.0** |
| Gemini-2.0 Flash Thinking | 47.2 | 49.6 | **+2.4** | 42.9 | 35.3 | **-7.6** |
| o1 | 56.0 | 47.0 | **-7.0** | 73.5 | 29.6 | **-43.9** |
| | *Open-source Models* | | | | | |
| LLaVA-OneVision | 0.0 | 40.5 | **+40.5** | 30.3 | 14.6 | **-15.7** |
| InternVL 2.5-78B | 33.2 | 41.4 | **+8.2** | 69.5 | 51.7 | **-17.8** |
| Qwen 2.5-VL-72B | 29.0 | 41.6 | **+12.6** | 72.3 | 47.7 | **-24.6** |

**Table 27:** Model performance on question-only prompts versus prompts that include explicit reasoning steps (Q+Steps). Δ values are Q+Steps performance - Q-only performance.

**Intermediate Visual Simulation States vs. Performance**

Table 28 summarizes extended results on varying the slice of intermediate visual simulation presented to the model across different tasks. Across models, which slice of the simulation you show matters, and the "best slice" shifts with task type. For 2D transformations, most closed-source mod-

els and the stronger open-source one (InternVL) peak when they see only the last intermediate state, gaining 2–6 points over the full roll-out; showing every intermediate frame ("all") often drags accuracy down a few points. For 3D transformations, the pattern flips—accuracy is usually highest with "all" states ($\approx$ +2–4 points over "partial"), while the last-only view tends to erase that gain, especially for GPT-4o, Gemini Flash, and o1. For cube nets, no single view helps every model. Scores barely change with "all" frames, and last-only often hurts closed-source models (-8 points on average) yet uniquely rescues LLaVA (+11 points). For Tangram puzzles, seeing "all" steps is consistently best: every model but LLaVA jumps 7–24 points versus the partial view, whereas last-only falls back to—or below—the partial baseline. Overall, for more complex tasks, models struggle to leverage intermediate visual states effectively.

| Model | 2D Transformation | | | 3D Transformation | | | Cube Nets | | | Tangram Puzzles | | |
|---|---|---|---|---|---|---|---|---|---|---|---|---|
| | Partial | All | Last | Partial | All | Last | Partial | All | Last | Partial | All | Last |
| *Closed-source Models* | | | | | | | | | | | | |
| GPT-4o | 86.8 | 82.8 | 89.4 | 72.1 | 68.4 | 68.4 | 51.3 | 52.2 | 35.2 | 43.5 | 51.5 | 43.4 |
| Claude-3.5-Sonnet | 67.8 | 71.4 | 70.7 | 54.9 | 57.8 | 55.9 | 58.7 | 51.6 | 46.8 | 43.5 | 67.6 | 43.3 |
| Gemini-2.0-Flash | 75.4 | 75.2 | 79.3 | 61.0 | 59.3 | 57.8 | 40.5 | 35.6 | 41.5 | 63.8 | 65.5 | 58.2 |
| o1 | 89.3 | 87.7 | 93.4 | 70.1 | 71.6 | 65.2 | 54.4 | 53.4 | 45.4 | 34.8 | 53.2 | 46.0 |
| *Open-source Models* | | | | | | | | | | | | |
| LLaVA-OneVision | 28.3 | 32.2 | 31.8 | 25.5 | 30.6 | 29.4 | 40.2 | 34.2 | 45.6 | 44.9 | 40.2 | 39.8 |
| InternVL 2.5-78B | 48.3 | 54.5 | 56.6 | 32.3 | 36.5 | 40.2 | 34.7 | 37.3 | 37.8 | 54.3 | 48.2 | 41.8 |
| Qwen 2.5-VL-72B | 44.4 | 48.5 | 44.4 | 48.7 | 49.1 | 43.6 | 41.9 | 53.4 | 42.3 | 49.0 | 56.7 | 44.3 |

**Table 28:** Model performance with partial, all, and last intermediate visual simulations.

