# OpenReview forum: "Unfolding Spatial Cognition: Evaluating Multimodal Models on Visual Simulations"
_ICLR.cc/2026/Conference — ICLR 2026 Poster_

### Official Review · Reviewer_BDc7 · 2025-10-21

**Soundness:** 3
**Presentation:** 4
**Contribution:** 4
**Rating:** 8
**Confidence:** 4

**Summary:**

This paper introduces STARE, a novel multimodal benchmark evaluating a model's capability on tasks that requires visual simulation. STARE features 4k tasks spanning foundational geometric transformations, integrated spatial reasoning and real-world spatial reasoning. Through experiments, the authors show that state of the art MLLMs perform well in simple 2D transformation tasks, but struggle with complex tasks like 3D cube folding. Human performance is also detailedly evaluated: although humans can achieve near-perfect accuracy, the completion time is considerable (up to 28 seconds). In addition, when providing visual simulation intermediate hints, humans can speed up their completion time while models exhibit inconsistent performance gains.

**Strengths:**

1. This paper is trying to draw attention to a relatively new field for multimodal AI: visual simulation. Which is crucial since many studies indicate that this is the core of human intelligence, and there are many important applications requiring such ability, such as arranging furniture. In addition, the lack of such a dataset further emphasizes the importance of the developed dataset.
2. The tasks introduced in the papers are diverse and well designed. The descriptions about each task are detailed and easy to follow. The inclusion of visual simulation hints is also useful in both model and human evaluation.
3. The experiment is comprehensive, covering multiple state of the art models, with and without visual simulation hints. Particularly, human response time is an important aspect which is not covered by previous benchmarks, this is important especially in benchmarks that are extremely difficult for current MLLMs.

**Weaknesses:**

1.  The role of visual perception in this task is important and experiment in this aspect can be done better. Currently, only one task (cube folding) is further decomposed to basic perception questions, however, the difficulty of visual perception varies across different tasks, it is possible for another task that visual perception is the key bottleneck. This is crucial since it can inform model developers on how to improve their models. In addition, since the random chance is 50, 57.4 performance on 3D perception should indicate very poor performance, which is contradictory to the claim in line 422-425.
2. Since most tasks are purely synthetic, it is questionable whether the performance in these tasks can reflect the model's performance on more real-world tasks (like object assembly, mechanical diagram interpretation as mentioned in the abstract).

**Questions:**

1.  I would like to hear the author's clarification regarding the weakness 1 above.
2.  Have the authors tried reasoning models on table3’s experiment? It is expected that reasoning models can perform much better when text is provided.

---

> ### Author Response · Authors · 2025-11-20
>
> We sincerely thank the reviewer for the positive assessment and constructive feedback, especially highlighting that our work draws attention to **visual simulation** as an important and underexplored area for multimodal AI, introduces a **diverse and well-designed set of tasks** with clear descriptions and useful visual simulation hints, and provides a **comprehensive experimental study** over state-of-the-art models, including **human response time**, which is rarely measured in prior benchmarks.
>
>
> > Role of visual perception beyond cube folding
>
> We thank the reviewer for emphasizing the role of visual perception and for suggesting that other tasks might also be perception-limited.
>
>
> In Appendix E, we actually provided analysis of the full chain-of-thought traces for **all** synthetic STARE tasks (2D, 3D, Cube Net, Tangram) using an LLM judge and four error types stand out:
>
> * **A — Misperception:** misreading shapes, colors, adjacency, or face layout
> * **B — Flawed spatial simulation:** incorrect mental model of how shapes transform
> * **C — Heuristic over-use:** shallow shortcuts (e.g., area/heuristic counting)
> * **D — Logic inconsistency:** reasoning contradicts itself or the final answer
>
> We find that while **misperception (A)** appears across tasks, especially in complex 3D scenes, a large fraction of errors—particularly in 3D and Cube Net—comes from **flawed spatial simulation (B)**. To reflect this, we have softened our wording in lines 422–425.
>
> To directly address the reviewer’s suggestion, we introduce **dedicated perception tests** for 2D, 3D, and **Tangram**. These tests isolate **basic visual sub-skills** that are prerequisites for the full tasks, rather than full multi-step reasoning.
>
> For example, for **Tangram**, the full task requires the model to:
> (1) **see how many pieces** are available,
> (2) understand **their sizes and shapes**,
> (3) track **rotations and placements**, and
> (4) ensure the placement completes the whole tangram puzzle (e.g., **no overlap** when composing the pieces).
>
> We therefore decompose Tangram into perception subtasks that mirror exactly these requirements:
>
> * **Count:** “How many pieces are available?”
> * **Size + Count:** “How many cells make up piece X?”
> * **Rotation:** “What is the rotation angle of piece X?”
> * **Position / Intermediate placement:** “Where is piece X located after step k?”
> * **Overlap (yes/no):** “Does this step result in overlapping pieces?”
>
> If a model fails these subtasks, it cannot reasonably succeed at full Tangram reasoning; conversely, strong performance here indicates that **basic visual perception is not the main bottleneck**, and remaining failures must come from higher-level composition and search.
>
> We apply the same philosophy to **2D** and **3D** transformations: we strip away transformations and ask only about **color**, **shape**, **position**, **size**, and **material** under balanced multiple-choice formats.

---

> > ### Author Response · Authors · 2025-11-20
> >
> > Below we report accuracies (random baseline shown for reference):
> >
> > | Task             | Perception Type              | Random | GPT-4o |   o3   |
> > | ---------------- | ---------------------------- | :----: | :----: | :----: |
> > | **Tangram**      | Count                        |   25%  |  90.0% | 100.0% |
> > |                  | Overlap (yes/no)             |   50%  |  75.0% |  92.0% |
> > |                  | Position (which piece)       |   25%  |  71.0% |  96.0% |
> > |                  | Rotation                     |   25%  |  53.0% |  67.5% |
> > |                  | Size + Count                 |   25%  |  74.1% |  88.9% |
> > |                  | Intermediate placement       |   25%  |  71.0% |  96.0% |
> > | **2D Transform** | Color (same/different)       |   50%  |  89.5% |  94.7% |
> > |                  | Shape (same/different)       |   50%  |  94.8% |  93.5% |
> > |                  | Position (closest to origin) |   25%  |  71.3% |  83.0% |
> > |                  | Size (largest/smallest)      |   25%  |  51.1% |  55.6% |
> > | **3D Transform** | Color (same/different)       |   50%  |  96.2% |  96.2% |
> > |                  | Shape (same/different)       |   50%  |  97.5% |  95.9% |
> > |                  | Position (closest to origin) |   25%  |  75.0% |  75.0% |
> > |                  | Material (same/different)    |   50%  |  55.6% |  56.3% |
> >
> > These results show that:
> >
> > * On **most basic perceptual subtasks** (Tangram count/position, 2D/3D color and shape, 3D position), models are **well above chance**, often near ceiling.
> > * Some aspects (e.g., 2D size, 3D material, fine-grained Tangram rotation) remain challenging and reflects the bottleneck in visual perception.
> > * However, when contrasted with much lower accuracies on the **full STARE reasoning tasks** (especially for GPT-4o), these perception scores indicate that **many failures are not due to basic perception alone**. For most tasks, models can see the right pieces, colors, and positions; they struggle to perform and track **multi-step spatial transformations and composition**.
> >
> > Together with the error-category analysis, this supports our updated claim: although visual perception contributes to errors in some settings, **higher-level spatial simulation and reasoning are still bottlenecks** across STARE.

---

> > > ### Author Response · Authors · 2025-11-20
> > >
> > > > Connection to more real-world tasks
> > >
> > > We agree with the reviewer that many real-world applications of visual simulation (e.g., object assembly, mechanical diagram interpretation) are more complex than our synthetic tasks. These scenarios usually require **additional domain knowledge** and language understanding on top of spatial reasoning, which makes it difficult to cleanly isolate *why* a model fails (knowledge vs. perception vs. spatial simulation). One motivation for STARE is precisely to **factor out these confounds** and focus on the core spatial reasoning and visual simulation abilities, in a setting where average humans perform near ceiling.
> > >
> > > To still probe more realistic settings, we constructed a small **mechanical diagram interpretation** set. We manually collected 28 questions from publicly available mechanical aptitude tests that are widely used in hiring for technical and engineering roles. These questions focus on **gear and pulley mechanics**, asking models to reason about rotation directions, speed ratios, and basic mechanical effects from diagrams.
> > >
> > >
> > > | Model             | Accuracy |
> > > | ----------------- | -------- |
> > > | GPT-4o            | 39.29%   |
> > > | o3                | 67.86%   |
> > >
> > > Although this is a small pilot, the results show that **even strong models struggle on real mechanical diagrams**, with performance far from reliable. This pattern is consistent with our findings on STARE: models have significant difficulty with multi-step spatial reasoning, and this difficulty carries over to more “real-world” mechanical tasks.
> > >
> > >
> > > > Reasoning models on Table 3 (text vs image vs image+text)
> > >
> > > We thank the reviewer for this suggestion. In the main paper, Table 3 reports results for GPT-4o. Following the reviewer’s comment, we repeated the same experiment with o3 on the corresponding test sets:
> > >
> > > | Task (Table 3) | Text-only | Image-only | Image+Text |
> > > | -------------- | :-------: | :--------: | :--------: |
> > > | 2D Trans.      |    92.8%   |    95.2%   |  **98.5%** |
> > > | 3D Trans.      |   71.0%   |  **73.5%** |    69.6%   |
> > > | Cube Nets      |   66.1%   |    64.3%   |  **69.0%** |
> > > | Tangram        | **87.7%** |    78.2%   |     N/A    |
> > >
> > >
> > > * For **2D Trans.** and **Cube Nets**, adding text on top of images (Image+Text) gives the best performance.
> > > * For **Tangram**, the **text-only** setting is strongest, consistent with strong language priors about piece counts and simple configurations.
> > > * For **3D Trans.**, performance is higher with **images** (73.5%) than with **text only** (71.0%), and Image+Text does not recover the full image-only advantage.
> > >
> > > These results **verified Table 3’s trends with a reasoning model**, and that while text can help, especially on more language-friendly tasks, **strong performance on 3D and other spatial tasks still critically depends on visual perception and non-verbal reasoning**, which is the core focus of STARE.

---

### Official Review · Reviewer_Pqgq · 2025-10-25

**Soundness:** 2
**Presentation:** 2
**Contribution:** 3
**Rating:** 4
**Confidence:** 4

**Summary:**

This paper introduces STARE, a new benchmark designed to evaluate the spatial reasoning capabilities of Multimodal Large Language Models (MLLMs). The authors argue that existing benchmarks neglect a crucial aspect of human intelligence: the ability to perform multi-step, non-verbal visual simulations. STARE is structured with a hierarchy of tasks, ranging from foundational 2D/3D geometric transformations to more integrated tasks like cube-net folding and tangram puzzles, and finally to real-world scenarios like perspective and temporal reasoning. Through extensive experiments on a suite of state-of-the-art closed and open-source models, the paper demonstrates a significant performance gap between current MLLMs and humans.

**Strengths:**

1. Novel and Well-Motivated Benchmark: The paper introduces STARE, a new benchmark that addresses a critical area of multimodal AI: multi-step spatial reasoning. It provides a structured framework for diagnosing the spatial cognition capabilities of models.

2. Thorough Experimental Evaluation: The study is grounded in a comprehensive evaluation of a wide range of contemporary models, including a crucial human performance baseline. The analysis offers some valuable insights into model failure modes.

**Weaknesses:**

1. **Potential Bias in the "Simulation" Paradigm**: The paper's core claim is about evaluating "visual simulations." However, its primary method for this—providing intermediate visual steps—tests a model's ability to interpret a given sequence of images, not its ability to generate that sequence internally. A human performing mental rotation isn't shown snapshots; their mind creates the intermediate frames. This is a subtle but critical distinction. The benchmark is, therefore, a stronger test of sequential visual comprehension and context integration than of genuine, unprompted mental simulation. This could favor models that are good at processing image-text interleaved inputs over those with true non-verbal reasoning abilities.

2. **Mismatch Between Core Tasks and "Real-World" Scenarios**: A potential weakness lies in the selection of the "real-world" tasks. While the paper's core thesis is to evaluate multi-step visual simulation, the chosen temporal and perspective reasoning tasks do not appear to measure this capability in the same way as the other tasks.

3. **Limited Real-World Complexity**: The tasks categorized as "Real-world Spatial Reasoning" are highly constrained (selecting a missing frame or a rendered viewpoint). This raises questions about whether the observed model failures and successes would generalize to messier, embodied tasks like actual robot-based assembly or navigation.

**Questions:**

Could the benchmark be strengthened by including real-world tasks that are more procedural and thus align better with the paper's core thesis on multi-step simulation? If sticking with the current tasks, is it feasible to frame them in a way that allows for a more consistent and meaningful evaluation under the with/without intermediate simulation settings?

---

> ### Author Response · Authors · 2025-11-20
>
> We appreciate the reviewer for the constructive feedback and for recognizing our work as a **novel, well-motivated benchmark with thorough experimental evaluation**.
>
>
> > Evaluation of models generating visual simulations
>
> Our current “w/ visual simulation” setting indeed tests whether a model can **use** a correct sequence of intermediate images as a visual scratchpad, rather than whether it can internally generate that sequence on its own. That said, we see this setting as **a necessary upper bound** for true visual simulation. If a model cannot solve a task even when given the correct intermediate frames, it is unlikely to succeed when it must first imagine or generate those frames itself. **Our results show that, even with ground-truth intermediate images, performance is still far from human level, suggesting a core weakness in spatial reasoning rather than just in image generation.**
>
>
> To directly address the reviewer’s concern about models that can generate visual thoughts, we also followed the suggestion to evaluate a unified text–image model, Bagel-7B. Although Bagel-7B in principle supports interleaved generation, the released checkpoint and inference code only support either text-only or image-only generation. We therefore modified the decoding logic to force interleaved generation: after the model produces a text “thought”, we force the model to switch to image generation mode with previous text thought as context to generate image tokens, and following that we switch back to text generation mode with all previous generation as context (both image and text) to continue reasoning in text until it spits out the end of sequence token. Because the model often does not strictly follow the requested format or directly state the answer, we use o3 as a judge model to extract answer choices and compute accuracy.
>
> | Task (w/o VisSim) | Text-only Generation     | Interleaved Generation     |
> | ---------------------------- | --------------- | --------------- |
> | 2D                           |   44.90   |     46.00     |
> | 3D                           |     24.00    |    48.00      |
> | Cube Net                     |     50.76    |     76.54     |
> | Tangram                      |    48.48     |     64.79     |
> | Temporal                     |    22.45     |    16.00      |
> | Perspective                  |     6.0    |       4.0   |
> | **Overall**                  |    32.76     |       42.56   |
>
>
> Interleaved generation helps on some tasks (3D, Cube Net, Tangram), but performance remains low overall and even drops on Temporal and Perspective. Qualitatively, we find that Bagel-7B often: (1) ignores or only loosely follows the requested output format. (2) produces long descriptions without committing to a clear final answer. (3) repeats itself or fails to give a direct answer, especially on Perspective tasks (examples below).
>
> These results support two key points relevant to the reviewer’s comment:
> - Our VisSim setting really is an upper bound. Even when a unified model is allowed to “simulate” by generating its own images, accuracy remains limited and does not close the gap to the setting where the correct intermediate frames are provided.
> - The bottleneck is not just interleaved IO. Simply allowing interleaved image–text generation does not automatically produce meaningful, task-aligned visual simulations. The main difficulty lies in performing the correct spatial transformations and using visual evidence to reach the right answer, not just in processing image–text sequences.
>
> We will addd the above discussion in revision.

---

> ### Author Response · Authors · 2025-11-20
>
> Example BAGLE-output
>
> ```
> # perspective (Text-only Generation)
>
> The question image shows a 3D model of a room with a red arrow pointing to a specific area. The question text asks for a zoomed-in view of that area. The answer image should focus on the section indicated by the arrow, providing a detailed, close-up view of the room's interior, including furniture and decor. The zoomed-in view should maintain clarity and detail, showing the textures and objects in the area.
> ```
>
> ```
> # perspective (Interleaved Generation)
>
> The user wants to compare the original 3D reconstruction (Image 1) with the final 3D reconstruction (Image 2). The goal is to highlight the differences in detail, clarity, and accuracy between the two images.
>
> First, I observe that Image 1 is a 3D reconstruction with a red arrow pointing to a specific area, likely indicating a focus point or area of interest. Image 2 is a more refined 3D reconstruction with improved clarity and detail, showing the same space but with enhanced textures and lighting.
>
> To create the answer image, I would overlay the two images side by side for direct comparison. The red arrow from Image 1 should be retained in Image 2 to maintain the focus point. Additionally, I would ensure that the lighting and color tones are consistent between the two images to facilitate a clear visual comparison. This approach effectively showcases the improvements in the final 3D reconstruction.
> </think>
> [Image: ./results/imgs/bagel_7b_baseline-STARE_seed42-TIT-T20251117_Gb3dfc6ff_TIT/bagel_out_74511645_1.jpg] -- this image simply copies the answer choice image.
> <think>
> The user wants to compare the original 3D reconstruction (Image 1) with the final 3D reconstruction (Image 2). The goal is to highlight the differences in detail, clarity, and accuracy between the two images.
>
> First, I observe that Image 1 is a 3D reconstruction with a red arrow pointing to a specific area, likely indicating a focus point or area of interest. Image 2 is a more refined 3D reconstruction with improved clarity and detail, showing the same space but with enhanced textures and lighting.
>
> To create the answer image, I would overlay the two images side by side for direct comparison. The red arrow from Image 1 should be retained in Image 2 to maintain the focus point. Additionally, I would ensure that the lighting and color tones are consistent between the two images to facilitate a clear visual comparison. This approach effectively showcases the improvements in the final 3D reconstruction.
> </think>
> <think>
> The user wants to compare the original 3D reconstruction (Image 1) with the final 3D reconstruction (Image 2). The goal is to highlight the differences in detail, clarity, and accuracy between the two images.
>
> First, I observe that Image 1 is a 3D reconstruction with a red arrow pointing to a specific area, likely indicating a focus point or area of interest. Image 2 is a more refined 3D reconstruction with improved clarity and detail, showing the same space but with enhanced textures and lighting.
>
> To create the answer image, I would overlay the two images side by side for direct comparison. The red arrow from Image 1 should be retained in Image 2 to maintain the focus point. Additionally, I would ensure that the lighting and color tones are consistent between the two images to facilitate a clear visual comparison. This approach effectively showcases the improvements in the final 3D reconstruction.
> </think>
> ```

---

> ### Author Response · Authors · 2025-11-20
>
> > W2 & Q2: Framing temporal and perspective tasks under “with/without simulation”
>
> We thank the reviewer for raising the concern about how our temporal and perspective tasks fit the “visual simulation” framing.
>
> For these real-world tasks, human “mental simulation” is much more implicit. When humans solve them, they may imagine how the camera moves or how the view changes, but they do not see a clear sequence of mental frames that we could record or annotate one by one. Because of this, it is very hard to define or collect human “intermediate visualizations” in the same way as for our synthetic tasks.
>
> To still study the effect of visual simulation on these two tasks, we create three settings:
>
> * **w/o VSim:**
>
>   * Perspective: The model sees a top-down map and the four candidate egocentric views (A–D), then must choose the correct egocentric view.
>   * Temporal: The model sees a 4-frames where one of them is missing (indicated by “?”), plus 3 candidate images (A–C), and must pick the image that best fills the missing slot.
>
> * **w/ partial VSim:**
>
>   * Perspective: In addition to the top-down map and candidates, we show 1 intermediate frame sampled along the camera trajectory that moves from the top-down configuration toward the final egocentric view.
>   * Temporal: We add 1 extra context frame before and after the missing frame position respectively, so the model sees how the scene changes locally around the gap.
>
> * **w/ all VSim:**
>
>   * Perspective: We show 3 intermediate frames sampled along the same camera trajectory from top-down to egocentric, effectively “unrolling” the viewpoint change before presenting the same A–D candidates.
>   * Temporal: We add 2 extra context frames before and after the missing frame respectively, giving a short, denser local sequence around the missing frame.
>
> We then evaluate GPT-4o and o3 under these three settings:
>
> | Model                 | Task        | w/o VSim  | w/ partial VSim | w/ all VSim |
> | --------------------- | ----------- | --------- | --------------- | ----------- |
> | **GPT-4o**            | Temporal    | **39.0%** | 37.4%           | 37.5%       |
> |                       | Perspective | 38.7%     | 55.0%           | **76.7%**   |
> | **o3**                | Temporal    | **51.4%** | 50.7%           | 50.3%       |
> |                       | Perspective | 42.8%     | 55.2%           | **84.8%**   |
>
> For **perspective** reasoning, as we provide more intermediate views along the agent’s trajectory (from top-down to egocentric), accuracy increases sharply. This shows that perspective tasks *do* support a meaningful with/without simulation framing: models benefit from additional intermediate frames in a way analogous to our synthetic visual-simulation tasks.
>
> For **temporal reasoning**, adding partial or full context yields only modest or no gains, suggesting that current models struggle to infer a coherent motion trajectory from video, even when more evidence is available. performance.
>
> It is worth noting that humans already reach 99.0% (temporal) and 98.1% (perspective) in the w/o VSim setting, i.e., without any extra simulated frames. In contrast, even with all intermediate frames shown, models are still far below these numbers. The large gap between human w/o VSim and model w/ all VSim, and the strong dependence of models on extra frames, highlight that models are still far from human-like spatial reasoning and rely heavily on external visual scaffolding rather than robust internal and implicit mental simulation.

---

> > ### Author Response · Authors · 2025-11-20
> >
> > > Limited real-world complexity and more procedural tasks
> >
> > We thank the reviewer for suggesting more procedural, robot-style tasks that better match multi-step simulation.
> >
> > Following this suggestion, we ran a **pilot experiment on robot planning** using the ALFRED benchmark [1]. In our setup, the model sees:
> >
> > * A **current state image** and a **goal state image**
> > * Several **candidate plans** (sequences of high-level actions) in text, and must choose which plan can reach the goal.
> >
> > For each sample, we also have the **intermediate state images** along the execution of the correct plan (average 5.73 steps; min 2, max 12). We treat these as visual simulations and evaluate models in several settings:
> >
> > * **w/o VSim:** only current image, goal image, and plan text.
> > * **w/ partial VSim (25%, 50%, 75%):** we show a subset of intermediate state images.
> > * **w/ all VSim:** we show all intermediate state images as a step-by-step visual cues.
> >
> > We construct distractor plans by either sampling from other trajectories in the same environment or perturbing the correct plan (swapping left/right, shuffling step order, etc.). We report the performance of GPT-4o and o3 on 200 multiple-choice samples below.
> >
> > | Model      | w/o VSim | w/ VSim 25% | w/ VSim 50% | w/ VSim 75% | w/ **all** VSim |
> > | ---------- | -------- | ----------- | ----------- | ----------- | --------------- |
> > | **GPT-4o** | 39.5%    | 60.5%       | 66.5%       | 68.0%       | **79.5%**       |
> > | **o3**     | 73.5%    | 71.0%       | 78.5%       | 87.0%       | **93.0%**       |
> >
> > This ALFRED-style robot planning task is more procedural and closer to embodied settings, and we do see a **strong benefit from visual simulations** as we add intermediate state images. However, in this setup a model can often succeed by **verbal reasoning alone**: it can describe the goal image in its reasoning (e.g., “a mug in the sink”) and then match this to a plan like “pick up the mug, then place it in the sink,” making it hard to isolate **non-verbal spatial reasoning** from language priors. We therefore treat ALFRED-style planning as a **complementary procedural test** that supports our findings, while keeping STARE focused on tightly controlled, non-verbal spatial reasoning, which better matches the core goal of our benchmark.
> >
> >
> > [1] Mohit Shridhar, Jesse Thomason, Daniel Gordon, Yonatan Bisk, Winson Han, Roozbeh Mottaghi, Luke Zettlemoyer, and Dieter Fox. ALFRED: A Benchmark for Interpreting Grounded Instructions for Everyday Tasks. In The IEEE Conference on Computer Vision and Pattern Recognition (CVPR), 2020

---

> > > ### Comment · Reviewer_Pqgq · 2025-11-26
> > >
> > > Thanks for the detailed response. I have no further issues and have raised my rating.

---

> > > > ### Author Response · Authors · 2025-11-28
> > > >
> > > > Thank you for raising the score. We sincerely appreciate your acknowledgment of our rebuttal.

---

### Official Review · Reviewer_Egcv · 2025-10-25

**Soundness:** 4
**Presentation:** 3
**Contribution:** 4
**Rating:** 8
**Confidence:** 5

**Summary:**

The paper introduces a benchmark known as STARE (Spatial Transformations and Reasoning Evaluation) to rigorously evaluate MLLMs on 2D and 3D geometric transformation tasks ideally solved through multi-step visual simulations. The tasks span three major varieties : (i) Foundational Geometric Transformations, (ii) Integrated spatial reasoning and (iii) Real-world spatial reasoning. Through ~4000 instances of these tasks, they demonstrate that models perform relatively well on reasoning over 2D transformations, but demonstrate random chance responses with 3D cube net folding and tangram puzzles. The authors show that when models receive intermediate visual steps, their performance improves for GPT-4o, Gemini-2.0 Flash Thinking and o1 while Gemini-2.0 Flash and Claude worsen on cube net folding. The paper also includes a detailed analysis of (i) how models understand individual transformation in 2D and 3D, showing a 3% - 8% increase, (ii) how model performance erodes with task complexity, showing a ~20% reduction in 2D tasks and ~12% reduction in 3D tasks, (iii) how model failures originate, demonstrating failure reduction on complexity removal, (iv) how do models reason spatially through textual descriptions, showing that text mainly helps in 2D tasks and (v) how models utilize visual simulations and how well do model use disconnected final visual simulations.

**Strengths:**

The paper presents the problem of composite reasoning through geometric transformation tasks clearly and concisely. I find these particular strengths of the paper interesting:

1) Extensive task definition - The variety of tasks covering: (i) integrated spatial reasoning over 2D and 3D, (ii) Foundational geometric transformations, and (iii) real-world spatial reasoning encompasses a majority of reasoning behaviour abilities for MLLMs.
2) Behavioural consistency with prior works - The paper demonstrates consistency of random memory associative behaviour with prior works in this domain, also extending to comprehensive real-world reasoning targeted towards helpful tasks such as cube folding and tangram shapes.
3) Comprehensive error analysis and behaviour evaluation - The research questions posed and analysed cover the major behaviours responsible for reasoning. They cover the performance reduction for an increase in task complexity, failure modes for models in tasks and the effectiveness of textual descriptions in spatial reasoning.

**Weaknesses:**

I have only one major weakness/ criticism to identify for the authors, although this has been highlighted in the limitations:

1) Comprehensive answering for models can be incorporated into the work, i.e. moving beyond just multiple-choice and binary question answering, towards one-word or two-word answering to also analyse the probabilistic tendency to actually give the true answer as a response and intermediate reasoning, and not just choose an option.

**Questions:**

I have one suggestion for authors:

1) The paper can include some more relevant work on geometric reasoning benchmarks. "Forgotten Polygons: Multimodal Large Language Models are Shape-Blind", "GeoGramBench: Benchmarking the Geometric Program Reasoning in Modern LLMs", and "VisuLogic: A Benchmark for Evaluating Visual Reasoning in Multi-modal Large Language Models" might be some works relevant to the current paper.

---

> ### Author Response · Authors · 2025-11-20
>
> We thank the reviewer for the positive assessment, especially noting our **extensive task definitions**, **coverage of real-world spatial reasoning**, and **detailed error and behavior analysis**.
>
> > More open-ended evaluation
>
> Our current design uses multiple-choice **image** options to (1) focus on **non-verbal spatial reasoning** rather than verbal reasoning, and (2) keep evaluation **simple and reproducible** across models. Even in this setting, we explicitly prompt models to **think step by step**, so they can freely produce intermediate reasoning before picking an answer. We also go beyond checking only the final choice: in Appendix E, we provide **fine-grained error analysis** where we examine the entire reasoning chain on all synthetic STARE tasks, which speaks to the reviewer’s request to analyze the reasoning process and when models actually arrive at the true answer.
>
> In addition, since cube net and tangram are the only tasks that do not require choosing from images (they are yes/no), we use them to test an **open-ended** setting. We ask GPT-4o to freely answer the question without restricting the answer format or providing options (no “A. Yes, B. No”), and then use o3 to map these free-form answers to yes/no:
>
> | Model  | Task     | MC (restricted) | **Open-Ended** |
> | ------ | -------- | --------------- | -------------- |
> | GPT-4o | Cube Net | 50.3%           | 50.9%          |
> |        | Tangram  | 52.5%           | 48.3%          |
>
> The similar performance across formats suggests that the **main difficulty comes from the underlying spatial reasoning challenge**, not from the multiple-choice format or from “just picking an option,” even when the model can reason and answer freely.
>
> We agree that extending to open-ended, short-form answers and more detailed analysis of those responses is a valuable next step, and we will highlight this as an important direction for future extensions of STARE.
>
>
>
> > More Related works
>
> We sincerely thank the reviewer for sharing the related works. We have updated Table 6 and Related Work section in Appendix accordingly.

---

> > ### Comment · Reviewer_Egcv · 2025-11-21
> >
> > Thank you for providing the additional results on open-ended answering. I find the justification that the actual root cause of the low performance (nearing random) on these tasks is due to the complexity of the tasks themselves, and not the answering format acceptable.

---

> > > ### Author Response · Authors · 2025-11-24
> > >
> > > Thanks for your response. Please let us know if you have any other questions, we would be more than happy to address them.

---

### Official Review · Reviewer_uKSW · 2025-11-01

**Soundness:** 4
**Presentation:** 4
**Contribution:** 4
**Rating:** 8
**Confidence:** 4

**Summary:**

This paper introduces the STARE benchmark, which evaluates multimodal large language models on vision-reasoning tasks involving spatial transformations and reasoning. It covers various contexts, reasoning types, and difficulty levels, showing that models excel at simple 2D transformations but struggle on more challenging tasks such as 3D transformations, real-world perspective taking, and temporal reasoning.

**Strengths:**

The paper is well-motivated and well-written, with a clear structure and sufficient detail about the execution of both dataset construction and experiments, which makes it easy to replicate and build upon. The benchmark covers the spatial reasoning tasks the authors target, with multiple difficulty levels and checks for different types of reasoning in models. The evaluations are robust, including settings with and without visual simulations and step-by-step reasoning. The findings are presented clearly and are straightforward, making the takeaways easy to understand. Overall, the paper offers a valuable contribution in both the idea and its execution.

**Weaknesses:**

I didn't notice major issues with this paper, but the few minor points I saw are as follows:

There is a problem with Figure 1: the question image does not match the step images.

The text mentions "Fig. 10" on line 196, but I assume the correct reference is the figure references 9 and 10. And Figure 10 has missing pieces.

Line 323: "Notably, o3 seems to better at leveraging visual simulations" has a grammatical error.

**Questions:**

I don't have any questions for the authors.

---

> ### Author Response · Authors · 2025-11-20
>
> We thank the reviewer for the very positive assessment, especially the comments that our work is *“well-motivated and well-written,”* *“easy to replicate and build upon,”* and that it *“covers the spatial reasoning tasks  with multiple difficulty levels and checks for different types of reasoning in models,”* with *“robust”* evaluations and *“clear”* findings that make the takeaways *“easy to understand.”*
>
>
> > Fig. 1 mismatch
>
> Thank you for catching this issue. We have updated Figure 1 to resolve the mismatch.
>
> > Missing figure reference in Line 196
>
> Thank you for pointing this out. We have updated Line 196 to explicitly reference both Figure 9 and 10 in the Appendix. The missing intermediate transformation in the second row is intentional and corresponds to the 2D visual analogy setting **without** intermediate visualization.
>
> > Line 323, grammatical error
>
> We have corrected the grammatical error at Line 323. Thank you for noting this.

---

> > ### Comment · Reviewer_uKSW · 2025-11-25
> >
> > I don't have any remaining concerns about this paper.
> >
> > Best of luck

---

### Author Response · Authors · 2025-12-04
**Rebuttal Summary to AC**

We thank the Program Committee, Senior AC, AC, and reviewers uKSW, Egcv, Pqgq, and BDc7 for their detailed and constructive feedback.

Reviewers consistently emphasized the value of STARE:
- Reviewer **uKSW** called it “*well-motivated and well-written, with a clear structure and sufficient detail … which makes it easy to replicate and build upon*” and noted that “*the benchmark covers the spatial reasoning tasks the authors target, with multiple difficulty levels and checks for different types of reasoning in models*.”
- Reviewer **Pqgq** highlighted STARE as “*a new benchmark that addresses a critical area of multimodal AI: multi-step spatial reasoning*” and “*a structured framework for diagnosing the spatial cognition capabilities of models*.” And notably, Reviewer Pqgq, **raised their rating from 4 to 6** after the discussion and revisions.
- Reviewer **BDc7** praised that the work “*draw[s] attention to a relatively new field for multimodal AI: visual simulation*” and that “*the lack of such a dataset further emphasizes the importance of the developed dataset*.”
- Reviewer **uKSW** ultimately stated: “*I don’t have any remaining concerns about this paper.*”

During the discussion period, we made several focused additions and clarifications:

1. **Additional experiments and ablations.**
   We added **open-ended answering** experiments (e.g., for cube nets and tangrams), and the resulting analysis—acknowledged by Reviewer Egcv as clarifying that low performance is due to task complexity rather than answer format—strengthens our interpretation of the results. We also introduced a **robot-planning pilot** on the **ALFRED** benchmark with and without intermediate visual simulations, directly responding to requests for more procedural, real-world-style tasks. For perspective and temporal reasoning, we formalized three conditions—**w/o VSim, partial VSim, and all VSim**—and evaluated models such as GPT-4o and o3 under these settings to more cleanly study the effect of intermediate visual simulations.

2. **Perception vs. spatial simulation.**
   To address concerns about whether failures stem from perception or reasoning, we added **dedicated perception tests for 2D, 3D, and Tangram**, isolating basic visual sub-skills (e.g., counting pieces, recognizing shapes/rotations and placements). Models reach near-ceiling on many of these subtasks yet still struggle on full STARE tasks, supporting our claim that **multi-step spatial simulation**, rather than low-level perception, is often the core bottleneck. We also refined our error taxonomy and softened the wording where misperception plays a non-trivial role.

3. **Clarifications and broader positioning.**
   We fixed minor presentation issues noted by reviewers (e.g., figure mismatch and wording glitches), **expanded the related-work section and comparison table** to explicitly include additional geometric reasoning benchmarks (such as *Forgotten Polygons*, *GeoGramBench*, and *VisuLogic*), and clarified STARE’s scope as a **tightly controlled, non-verbal spatial simulation benchmark** that factors out heavy language priors while keeping humans near ceiling.

Overall, the discussion phase helped us substantially improve the experimental support, diagnostic clarity, and positioning of STARE, and we are grateful to the committee and reviewers for guiding these revisions.

---

### Meta-Review · Area_Chair_GHPD · 2026-01-04

**Summary:**

This paper introduces a novel benchmark, focusing on evaluating MLLM visual spatial transformation and reasoning. Reviewers find this paper is clearly written and well motivated. The benchmark is carefully designed, covering a broad range of spatial and geometric reasoning tasks at different difficulty levels. In the initial version, there are a few concerns and questions raised; for example, the benchmark primarily evaluates models' ability to interpret provided intermediate visual steps rather than perform internal visual simulation. Additionally, the "real-world" tasks remain highly constrained and synthetic, resulting in generalization concerns for more complex, real-world scenarios. While in the rebuttal, the authors provided a detailed response, and reviewers are satisfied with the response. Reviewer BDc7 has no reply to the rebuttal, but gives a positive score initially.

**Reviewer Concerns:**

Apart from Reviewer BDc7, who might not have had the chance to reply to the rebuttal, other reviewers all state that their concerns are addressed. Particularly, Reviewer Pqgq raised the rating from 4 to 6 after discussion. All reviewers have a consensus that this is a solid and interesting paper.

**Reviewer Scores:**

Reviewer Pqgq mentioned that he/she has raised the score. All other reviewers stay positive.

---

### Decision · Program_Chairs · 2026-01-26

Accept (Poster)